# Adaptive Preconditioners Trigger Loss Spikes in Adam

Zhiwei Bai [* 1 2]   Zhangchen Zhou [* 1 2]   Jiajie Zhao [1 2]   Xiaolong Li [1 2]   Zhiyu Li [3 4]   Feiyu Xiong [3 4]
Hongkang Yang [3 4]   Yaoyu Zhang [1 2]   Zhi-Qin John Xu [1 2 5]

## Abstract

Loss spikes commonly emerge during neural network training with the Adam optimizer across diverse architectures and scales, yet their underlying mechanism remains elusive. While previous explanations attribute these phenomena to sharper loss landscapes at lower loss, we show that landscape geometry alone is insufficient to explain the phenomenon. In this work, we pinpoint the root cause in the internal dynamics of Adam's second moment estimator. We identify a critical "decoupling" mechanism where the adaptive preconditioner $v_t$ fails to track the instantaneous squared gradients $g_t^2$, causing the adaptive mechanism to effectively fail. This decoupling allows the preconditioner to decay autonomously despite rising gradients, which pushes the maximum eigenvalue of the preconditioned Hessian beyond the stability threshold $2/\eta$ for sustained periods, manifesting as dramatic loss spikes. Through a quadratic approximation analysis, we theoretically and experimentally characterize five distinct stages of spike evolution and propose a predictor for anticipating spikes based on gradient-directional curvature. We empirically find that the proposed loss spike mechanism, although derived from simplified models, generalizes well to practical scenarios ranging from small neural networks to large-scale Transformers.

---

[*]Equal contribution   [1]Institute of Natural Sciences, MOE-LSC, Shanghai Jiao Tong University [2]School of Mathematical Sciences, Shanghai Jiao Tong University [3]MemTensor (Shanghai) Technology Co., Ltd.   [4]Institute for Advanced Algorithms Research, Shanghai [5]Shanghai Seres Information Technology Co., Ltd, Shanghai 200040, China.   Correspondence to: Yaoyu Zhang <zhyy.sjtu@sjtu.edu.cn>, Zhi-Qin John Xu <xuzhiqin@sjtu.edu.cn>.

*Proceedings of the 43$^{rd}$ International Conference on Machine Learning*, Seoul, South Korea. PMLR 306, 2026. Copyright 2026 by the author(s).

## 1. Introduction

Neural network optimization remains complex and unpredictable despite significant advances in training methodologies. One particularly intriguing phenomenon is the "loss spike"—a sudden, sharp surge in the loss function that subsequently subsides. As illustrated in Fig. 1, these spikes differ markedly from normal fluctuations, resembling systematic instabilities rather than random noise. Though observed across diverse architectures and datasets, their underlying mechanisms remain poorly understood. This poses a critical question: should practitioners intervene to eliminate these spikes, or might they actually benefit optimization? Answering this requires deeper understanding of when, how, and why loss spikes occur.

Previous research has attempted to explain loss spikes through loss landscape geometry (Ma et al., 2022a; Li et al., 2025). The lower-loss-as-sharper (LLAS) hypothesis (Li et al., 2025) suggests that lower-loss regions correspond to sharper curvature, potentially causing instability. However, this explanation fails to account for the behavior of adaptive optimizers like Adam (Kingma & Ba, 2014), which consistently exhibit spikes even in simple scenarios with well-understood geometry. For instance, as shown in Fig. 2(a), Adam produces loss spikes on a quadratic function even with learning rates well below theoretical stability thresholds, while gradient descent (GD) converges smoothly. Since quadratic functions have constant curvature, landscape geometry alone cannot explain this behavior. Furthermore, while previous work has identified the Edge of Stability (EoS) phenomenon in GD (Cohen et al., 2021; Wu et al., 2018; Xing et al., 2018; Ahn et al., 2022; Lyu et al., 2022; Arora et al., 2022; Wang et al., 2022; Cohen et al., 2023), where loss decreases non-monotonically while the largest Hessian eigenvalue hovers around $2/\eta$ ($\eta$ is the learning rate), loss spikes represent more *dramatic instabilities* than typical EoS behavior. **The precise relationship between these instabilities and observed spikes remains unclear—instability may manifest as oscillations or spikes (Ma et al., 2022b), but the specific mechanism governing spike occurrence is not well understood.**

In this work, we present a mechanistic analysis of loss spikes in Adam optimization. Our key finding is that these spikes

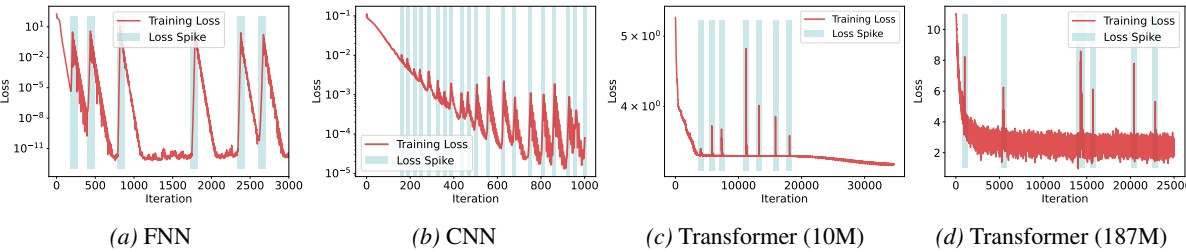

*Figure 1.* Loss spikes across architectures: (a) FNNs for function approximation. (b) CNNs on CIFAR10. (c-d) Transformers on language tasks. See experimental details in App. G.

arise not primarily from complex loss landscape geometry, but from the internal dynamics of Adam's second moment estimator. We demonstrate both theoretically and experimentally that Adam's stability is governed by a preconditioned Hessian. We identify a critical "**decoupling**" mechanism where the adaptive preconditioner $v_t$ fails to track the instantaneous squared gradients $g_t^2$, causing the adaptive mechanism to effectively fail. Although the derived mechanism and indicators are from simple model analysis, we find that the proposed loss spike mechanics generalize well to realistic, high-dimensional settings. In addition, we find that directly reducing $\beta_2$ effectively mitigates loss spikes.

Our main contributions are:

(i) We show that landscape geometry alone is insufficient to explain loss spikes—Adam's adaptive preconditioners can independently cause spikes in practical training. This mechanism differs from the previous lower-loss-as-sharper (LLAS) landscape hypothesis (Li et al., 2025) (see Sec. 3, Sec. 4.1, and Sec. 5).

(ii) Although the AEoS literature (Cohen et al., 2023) mentions similar preconditioned Hessian perspective, the precise relationship with severe spikes remains unclear. We identify a critical "decoupling" mechanism where $v_t$ fails to track $g_t^2$, allowing the preconditioner to decay autonomously despite rising gradients. This pushes the maximum eigenvalue of the preconditioned Hessian beyond $2/\eta$ for sustained periods, manifesting as dramatic loss spikes. Conversely, momentary overshoots manifest as typical EoS oscillations rather than spikes (see Sec. 4.1, Sec. 4.2, and Sec. 6).

(iii) We propose $\lambda_{\mathrm{grad}}(\hat{\boldsymbol{H}}_t)$, a predictor for anticipating spikes based on curvature in the gradient direction. We empirically show this predictor is more accurate than established $\lambda_{\max}$ in forecasting spike onset, and validate practical strategies for mitigating spikes (see Sec. 4.3 and Sec. 6).

## 2. Related Works

**Edge of Stability (EoS).** Various works (Cohen et al., 2021; Wu et al., 2018; Xing et al., 2018; Ahn et al., 2022; Lyu et al., 2022; Arora et al., 2022; Jastrzebski et al., 2020;

Jastrzębski et al., 2019; Lewkowycz et al., 2020) have investigated the *Edge of Stability* (EoS), a phenomenon where gradient descent progressively increases the sharpness of the loss landscape—a process known as *progressive sharpening*—until the maximum Hessian eigenvalue stabilizes near the threshold $2/\eta$, while the loss continues to decrease non-monotonically. Ma et al. (2022a) proposed a subquadratic structure near local minima, where sharpness increases when the loss decreases along the gradient direction, providing a theoretical account of this behavior. Other studies (Damian et al., 2023; Wang et al., 2022) show that when $\lambda_{\max} > 2/\eta$, self-stabilization mechanisms can reduce sharpness and restore stability. More recently, Cohen et al. (2023) extended the EoS framework to adaptive optimizers, introducing the concept of *Adaptive Edge of Stability* (AEoS). Furthermore, Cohen et al. (2025) also developed the concept of central flow to study the average trajectory of oscillatory dynamics during EoS. **While EoS has been widely explored, its direct association with loss spikes has yet to be thoroughly investigated.**

**Convergence Analysis of Adam.** Numerous works have analyzed the convergence behavior of adaptive gradient methods (Chen et al., 2019; Li & Orabona, 2019; Xie et al., 2020; Défossez et al., 2022; Da Silva & Gazeau, 2020; Shi et al., 2021; Zou et al., 2019; Zhou et al., 2024). In particular, Reddi et al. (2018) demonstrated that Adam may fail to converge even in simple convex settings, prompting a series of variants (Liu et al., 2019; Taniguchi et al., 2024). Zhang et al. (2022) showed that in the case of learning rate decay Adam can converge to a neighborhood of critical points when $\beta_2$ is large, and $\beta_1 < \sqrt{\beta_2}$. Additionally, recent continuous-time ODE approximations and uniform a prior bounds have advanced the global stability and convergence rate analysis of Adam (Dereich et al., 2025c;b;a; 2026).

**Loss Spike Analysis.** Chowdhery et al. (2023) reported that restarting training from an earlier checkpoint and skipping the spiking data batch can mitigate spikes in large models. Molybog et al. (2023) found that the gradient and second-moment estimates of shallow layer parameters can decay to near-zero and then spike upon encountering a large gradient. Li et al. (2025) argued that spikes occur in sharp regions

of the loss landscape with a lower-loss-as-sharper (LLAS) structure. Ma et al. (2022b) qualitatively demonstrated that Adam's hyperparameters impact the occurrence of spikes or oscillations. **Although previous studies have uncovered parts of the puzzle surrounding spikes, this work provides a more detailed and comprehensive understanding of the spike formation.**

## 3. Distinct Loss Spike Mechanism in Adam

**Adam Algorithm.** The Adam algorithm is widely used in training Transformer models and is widely observed to be more prone to cause loss spikes. Adam maintains exponential moving averages of gradients (first moment) and squared gradients (second moment) to speed up training:

$$\boldsymbol{m}_t = \beta_1 \boldsymbol{m}_{t-1} + (1-\beta_1)\boldsymbol{g}_t,$$
$$\boldsymbol{v}_t = \beta_2 \boldsymbol{v}_{t-1} + (1-\beta_2)\boldsymbol{g}_t^2, \tag{1}$$

where $\boldsymbol{g}_t := \nabla L(\boldsymbol{\theta}_t)$ is the gradient, and $\beta_1, \beta_2 \in [0,1)$ are hyperparameters controlling the exponential decay rates (default values: $\beta_1 = 0.9, \beta_2 = 0.999$). To counteract the initialization bias toward zero, these moments are corrected: $\hat{\boldsymbol{m}}_t = \frac{\boldsymbol{m}_t}{1-\beta_1^t}$, $\hat{\boldsymbol{v}}_t = \frac{\boldsymbol{v}_t}{1-\beta_2^t}$. The parameter update rule is:

$$\boldsymbol{\theta}_{t+1} = \boldsymbol{\theta}_t - \eta\frac{\hat{\boldsymbol{m}}_t}{\sqrt{\hat{\boldsymbol{v}}_t} + \varepsilon}, \tag{2}$$

where $\eta > 0$ is the learning rate and $\varepsilon > 0$ is a small constant (default $10^{-8}$ in PyTorch).

**Differences in Spike Behavior Between GD and Adam.** Adaptive methods like Adam exhibit fundamentally different behavior compared to standard GD. A notable distinction is that Adam can encounter convergence difficulties even with simple quadratic functions and very small learning rates. For the quadratic function $f(\theta) = \frac{1}{2}\theta^2$, it is well established that GD converges when the learning rate $\eta < 2/\lambda_{\max} = 2$ (depicted by the black dashed line in Fig. 2(a)). However, Adam displays more intricate dynamics. As illustrated in Fig. 2(a), Adam with a learning rate $\eta \ll 2$ (using hyperparameters $\beta_1 = 0.9, \beta_2 = 0.99, \varepsilon = 10^{-8}$) still fails to converge. This non-convergence manifests in the distinctive colored curves in Fig. 2(a), where the training loss initially decreases steadily before abruptly spiking to a substantially higher magnitude.

Fig. 2(b) further examines the relationship between Adam's second moment $\sqrt{\hat{v}_t}$ at spike occurrence and learning rate. From Fig. 2(b), we observe that smaller learning rates correspond to smaller $\sqrt{\hat{v}_t}$ values when spikes occur, with the relationship appearing linear in log-log scale with a slope near 1. For one-dimensional quadratic optimization, $\eta/\sqrt{\hat{v}_t}$ can be interpreted as the effective learning rate and it increases as training progresses because $\sqrt{\hat{v}_t}$ diminishes

alongside the gradient $g_t$ according to Eq. (1). Experimentally, Fig. 2(c) confirms that this ratio increases until reaching a nearly consistent threshold value 38 (see Prop. 4.2 for a theoretical explanation), at which point the loss spike invariably occurs. While straightforward, this analysis provides valuable intuition for the emergence of spikes. However, it is important to note that in high-dimensional optimization scenarios, $\sqrt{\hat{v}_t}$ becomes a vector rather than a scalar, rendering the notion of an effective learning rate inapplicable. In the following section, we will quantitatively characterize Adam's spike behavior in more general settings.

## 4. Loss Spike Analysis of Adam

**Quadratic Approximation.** To understand the mechanism of loss spikes, we use classic linear stability analysis, which begins with a quadratic approximation. Consider optimizing a loss function $L(\boldsymbol{\theta})$ with respect to parameters $\boldsymbol{\theta} \in \mathbb{R}^M$. Around any current training step point $\theta_0$ (which we use as a local reference for the instantaneous update window), we approximate the loss using a second-order Taylor expansion:

$$L(\boldsymbol{\theta}_0 + \delta\boldsymbol{\theta}) \approx \tilde{L}(\delta\boldsymbol{\theta}) := L(\boldsymbol{\theta}_0) + \nabla L(\boldsymbol{\theta}_0)^\top \delta\boldsymbol{\theta} + \frac{1}{2}\delta\boldsymbol{\theta}^\top \boldsymbol{H}\delta\boldsymbol{\theta}, \tag{3}$$

where $\tilde{L}$ is quadratic in $\delta\boldsymbol{\theta}$, $\nabla L(\boldsymbol{\theta}_0)$ is the gradient, and $\boldsymbol{H} := \boldsymbol{H}(\boldsymbol{\theta}_0) = \nabla^2 L(\boldsymbol{\theta}_0)$ is the local Hessian at $\boldsymbol{\theta}_0$.

**Stability Analysis.** For the local quadratic model $\tilde{L}(\delta\boldsymbol{\theta})$ optimized with GD at learning rate $\eta$, the deviation $\delta\boldsymbol{\theta}_t$ evolves as:

$$\delta\boldsymbol{\theta}_{t+1} \approx \delta\boldsymbol{\theta}_t - \eta\nabla\tilde{L}(\delta\boldsymbol{\theta}_t) = \delta\boldsymbol{\theta}_t - \eta(\nabla L(\boldsymbol{\theta}_0) + \boldsymbol{H}\delta\boldsymbol{\theta}_t)$$
$$= (\boldsymbol{I} - \eta\boldsymbol{H})\delta\boldsymbol{\theta}_t - \eta\nabla L(\boldsymbol{\theta}_0). \tag{4}$$

Optimization becomes unstable along the maximum eigendirection when $\lambda_{\max}(\boldsymbol{H}) > 2/\eta$.

**Practical Stability Condition.** In neural network optimization, the loss landscape—and consequently the Hessian—evolves continuously as parameters update. However, the local Hessian stability condition indeed ensures stable loss decrease at each iteration, as formalized below.

**Proposition 4.1** (see App. D Prop. D.1 for proof). *Let $L : \mathbb{R}^M \to \mathbb{R}$ be twice continuously differentiable. For any iterate $\boldsymbol{\theta}_t$, define the gradient $\boldsymbol{g}_t := \nabla L(\boldsymbol{\theta}_t)$ and, for fixed learning rate $\eta > 0$, define the local directional maximum Hessian $\bar{\lambda}_t := \sup_{s \in [0,1]} \lambda_{\max}(\nabla^2 L(\boldsymbol{\theta}_t - s\eta\boldsymbol{g}_t))$, the maximum eigenvalue of the Hessian along the line segment from $\boldsymbol{\theta}_t$ to $\boldsymbol{\theta}_{t+1} = \boldsymbol{\theta}_t - \eta\boldsymbol{g}_t$. If $\eta < \frac{2}{\bar{\lambda}_t}$, the GD step $\boldsymbol{\theta}_{t+1} = \boldsymbol{\theta}_t - \eta\boldsymbol{g}_t$ satisfies:*

$$L(\boldsymbol{\theta}_{t+1}) \leq L(\boldsymbol{\theta}_t) - \eta\left(1 - \frac{\eta\bar{\lambda}_t}{2}\right)\|\boldsymbol{g}_t\|^2.$$

*In particular, whenever $\eta \in (0, 2/\bar{\lambda}_t)$ and $\boldsymbol{g}_t \neq 0$, we have strict decrease $L(\boldsymbol{\theta}_{t+1}) < L(\boldsymbol{\theta}_t)$.*

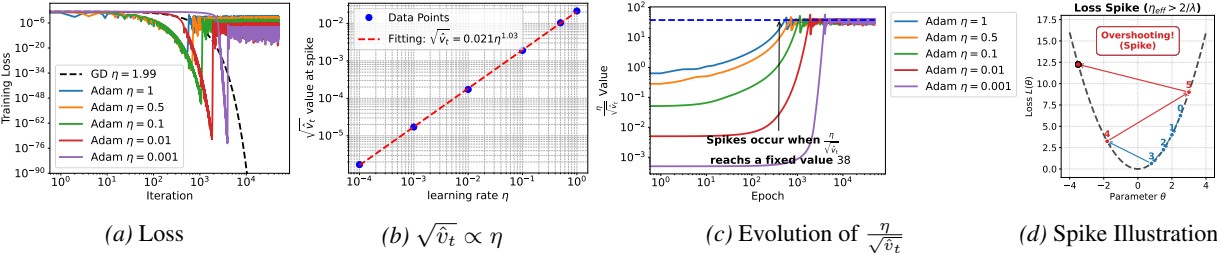

*(a)* Loss          *(b)* $\sqrt{\widehat{v}_t} \propto \eta$          *(c)* Evolution of $\frac{\eta}{\sqrt{\widehat{v}_t}}$          *(d)* Spike Illustration

*Figure 2.* Optimization of $L(\theta) = \frac{1}{2}\theta^2$. (a) Loss trajectories during Adam and GD training across various learning rates. Curves of different colors represent Adam's training loss, which initially decreases steadily before abruptly spiking to significantly higher values. (b) The relationship between learning rate and $\sqrt{\widehat{v}_t}$ value at spike occurrence follows a power law, appearing as a straight line with a slope of approximately 1 in log-log scale. (c) Under different learning rates, the ratio $\eta/\sqrt{\widehat{v}_t}$ consistently reaches a nearly identical threshold value immediately before the loss begins to spike. (d) Illustration of a spike that arises when the effective learning rate exceeds $2/\lambda$.

In practice, since learning rates are typically small, we can monitor **the step-wise stability condition** $\lambda_{\max}(\boldsymbol{H}_t) \leq 2/\eta$ **as a proxy**. When this condition is *persistently* violated, a loss spike is likely to occur.

### 4.1. Adam's Preconditioned Hessian and Stability

**Stability Analysis of Adaptive Mechanism.** To analyze Adam's stability conditions, we first examine the adaptive mechanism by setting $\beta_1 = 0$, ignoring momentum effects. Following the Taylor expansion approach from Eq. (3), we have:

$$\delta\boldsymbol{\theta}_{t+1} \approx \delta\boldsymbol{\theta}_t - \eta\frac{\nabla\tilde{L}(\delta\boldsymbol{\theta}_t)}{\sqrt{\widehat{\boldsymbol{v}}_t} + \varepsilon}$$
$$= \left(\boldsymbol{I} - \eta\mathrm{diag}\left(\frac{1}{\sqrt{\widehat{\boldsymbol{v}}_t} + \varepsilon}\right)\boldsymbol{H}\right)\delta\boldsymbol{\theta}_t - \eta\frac{\nabla L(\boldsymbol{\theta}_0)}{\sqrt{\widehat{\boldsymbol{v}}_t} + \varepsilon}. \quad (5)$$

Stability requires the spectral radius $\rho\left(\boldsymbol{I} - \eta\hat{\boldsymbol{H}}\right) < 1$, where $\hat{\boldsymbol{H}} = \mathrm{diag}((\sqrt{\widehat{\boldsymbol{v}}_t} + \varepsilon)^{-1})\boldsymbol{H}$ is the "adaptive preconditioned Hessian". Although asymmetric, $\hat{\boldsymbol{H}}$ is diagonalizable with strictly real eigenvalues[1] (see App. D Lem. D.1 for proof), yielding the stability condition $\lambda_{\max}(\hat{\boldsymbol{H}}) < 2/\eta$.

**Stability Analysis of Momentum Mechanism.** With momentum ($\beta_1 > 0$), we analyze the update rule $\boldsymbol{\theta}_{t+1} = \boldsymbol{\theta}_t - \eta\boldsymbol{m}_t$. Following the same Taylor expansion approach: $\delta\boldsymbol{\theta}_{t+1} \approx \delta\boldsymbol{\theta}_t - \eta(\beta_1\boldsymbol{m}_{t-1} + (1-\beta_1)(\nabla L(\boldsymbol{\theta}_0) + \boldsymbol{H}\delta\boldsymbol{\theta}_t))$. Substituting $\eta\boldsymbol{m}_{t-1} = \delta\boldsymbol{\theta}_{t-1} - \delta\boldsymbol{\theta}_t$ gives:

$$\delta\boldsymbol{\theta}_{t+1} \approx [(1+\beta_1)\boldsymbol{I} - \eta(1-\beta_1)\boldsymbol{H}]\,\delta\boldsymbol{\theta}_t$$
$$- \beta_1\delta\boldsymbol{\theta}_{t-1} - \eta(1-\beta_1)\nabla L(\boldsymbol{\theta}_0). \quad (6)$$

**Proposition 4.2** (see App. D Prop. D.2 for proof). *Consider the three-term recursive iteration Eq. (6) with learning rate*

---

[1]Let $\boldsymbol{D}_t = \mathrm{diag}\left(1/(\sqrt{\widehat{\boldsymbol{v}}_t} + \varepsilon)\right)$, which is positive definite. We can express the preconditioned Hessian as $\hat{\boldsymbol{H}} = \boldsymbol{D}_t\boldsymbol{H} = \boldsymbol{D}_t^{1/2}(\boldsymbol{D}_t^{1/2}\boldsymbol{H}\boldsymbol{D}_t^{1/2})\boldsymbol{D}_t^{-1/2}$. Since $\boldsymbol{D}_t^{1/2}\boldsymbol{H}\boldsymbol{D}_t^{1/2}$ is symmetric, $\hat{\boldsymbol{H}}$ is similar to a symmetric matrix, guaranteeing that it has real eigenvalues and is fully diagonalizable ($\hat{\boldsymbol{H}} = \boldsymbol{P}\boldsymbol{\Lambda}\boldsymbol{P}^{-1}$).

$\eta > 0$ *and momentum parameter* $\beta_1 \in [0, 1)$. *The linearized system at* $\boldsymbol{\theta}_0$ *is asymptotically stable in a specific eigendirection if and only if the corresponding eigenvalue* $\lambda_i$ *satisfies:*

$$0 < \lambda_i < \frac{2}{\eta} \cdot \frac{1 + \beta_1}{1 - \beta_1}.$$

*Consequently, to prevent instabilities that manifest as loss spikes in positive-curvature directions, the maximum positive eigenvalue must satisfy* $\lambda_{\max}\left(\frac{1-\beta_1}{1+\beta_1}\boldsymbol{H}\right) < \frac{2}{\eta}$.

*Remark* 4.3 (**Analysis of Stable and Unstable Directions.**). When the upper stability bound is violated ($\lambda_i > \frac{2}{\eta}\frac{1+\beta_1}{1-\beta_1}$) in positive-curvature directions, the corresponding parameter components undergo explosive exponential growth. Although the loss may simultaneously decrease along other stable or negative-curvature directions, a persistent instability in these sharp positive directions will eventually dominate the overall dynamics, macroscopically manifesting as a sharp loss spike. This competing dynamic highlights a fundamental trade-off: global landscape metrics or generic eigenvalue tracking may fail to capture the localized onset of inflation. Consequently, monitoring the gradient-directional curvature becomes essential for detecting impending loss spikes, a mechanism we formally develop and validate in Sec. 4.3.

**Comprehensive Stability Analysis of Adam: A Dual Preconditioning Perspective.** The core idea of the stability framework lies in recognizing Adam not merely as an algorithmic update rule, but as a dual-preconditioning operator on the Hessian of the loss landscape. Specifically, Adam introduces two structurally independent preconditioning mechanisms: (1) a *temporal preconditioning* governed by momentum ($\beta_1$), which scales the effective curvature by $\frac{1-\beta_1}{1+\beta_1}$ (as shown in Prop. 4.2); and (2) a *spatial metric preconditioning* governed by the adaptive term ($\beta_2$), which transforms the coordinate system via $\mathrm{diag}\left(1/(\sqrt{\widehat{\boldsymbol{v}}_t} + \varepsilon)\right)$. Because these two mechanisms operate on decoupled dimensions (temporal inertia versus spatial scaling), their effects can be composed multiplicatively. Integrating both mecha-

nisms and the momentum bias correction $\hat{m}_t = \frac{m_t}{1-\beta_1^t}$, the comprehensive "Adam preconditioned Hessian[2]" becomes:

$$\hat{H}_t = \frac{1}{1-\beta_1^t} \frac{1-\beta_1}{1+\beta_1} \text{diag}\left(\frac{1}{\sqrt{\hat{v}_t}+\varepsilon}\right) H_t. \quad (7)$$

In Sec. 4.3, we experimentally validate that this preconditioned step-wise instability criterion $\lambda_{\max}(\hat{H}_t) > 2/\eta$ accurately predicts loss spikes in one-dimensional case.

*Remark* 4.4 (**Justification of Integrating Momentum and Adaptive Mechanisms.**). Strictly speaking, compositing the spatial and temporal preconditioners into a single matrix $\hat{H}_t$ to evaluate step-wise stability mathematically relies on a *frozen-time approximation*. Incorporating Adam's full update rule yields a Linear Time-Varying (LTV) system. However, this theoretical heuristic is highly justified by the intrinsic timescale separation in Adam's design: typical hyperparameters (e.g., $\beta_2 = 0.999$) render the spatial metric $\hat{v}_t$ a "slow" variable compared to the "fast" temporal gradient dynamics. Because $\hat{v}_t$ is quasi-static within a short local optimization window (especially during the decoupling phase analyzed in Sec. 4.2), the spatial and temporal preconditioning effects do not dynamically interfere. This structural independence allows us to safely treat the system as locally Linear Time-Invariant (LTI) to derive the unified threshold $\lambda_{\max}(\hat{H}_t) < 2/\eta$, a mechanism overwhelmingly corroborated by our empirical validations.

*Remark* 4.5 (**Clarification on Step-wise vs. Global Approximation.**). It is crucial to emphasize that our quadratic model is employed for *step-wise, localized* stability analysis, rather than predicting the entire macroscopic trajectory of a loss spike. We do not assume a globally fixed Hessian $H(\theta_0)$ governs the large-displacement dynamics across the loss landscape. Instead, we continuously evaluate the local preconditioned Hessian $\hat{H}_t$ at each instantaneous iteration $t$. As formalized in Prop. 4.1, a continuous decrease in loss fundamentally requires the step-wise spectral radius to be less than $2/\eta$. While the developed loss spike is a highly non-linear macroscopic event, its *trigger*—the localized overshooting that breaches the current basin—is rigorously captured by the instantaneous violation of this local step-wise stability threshold ($\lambda_{\max}(\hat{H}_t) > 2/\eta$).

## 4.2. Decoupling of Preconditioner and Gradient Precipitates Loss Spikes

In this section, we demonstrate how Adam's internal dynamics—specifically the interaction between $v_t$ and $g_t^2$—can drive the optimizer to breach the stability threshold

for sustained periods. We analyze these dynamics through controlled experiments on the canonical quadratic objective.

**Decoupling Causes Sustained Instability (Spikes).** Fig. 3(a–b) shows results using the standard setting $\beta_1 = 0.9$ and $\beta_2 = 0.99$. Initially, the loss decreases gradually. A spike emerges at epoch 782, precisely when the effective curvature $\lambda_{\max}(\hat{H}_t)$ exceeds the stability threshold $2/\eta$. The mechanism unfolds as follows: As training progresses, the decay of $v_t$ naturally drives $\lambda_{\max}(\hat{H}_t)$ upward. Once it crosses $2/\eta$, the system enters an unstable regime where gradients begin to grow. Ideally, the adaptive mechanism should counter this by increasing $v_t$ to reduce the effective step size. However, we observe a critical **mismatch phase** (blue span in Fig. 3(b)) where the adaptive mechanism fails. Although the first moment $m_t$ tracks the rising gradient $g_t$, the second moment $v_t$ decouples from the instantaneous squared gradient $g_t^2$. Despite rising gradients, $v_t$ continues to decay autonomously, dominated by its history term: $v_t \approx \beta_2 v_{t-1}$. This is empirically confirmed by the red dashed line in Fig. 3(b), which fits the decay as $\sqrt{v_t} \propto \alpha^t$ with $\alpha \approx \sqrt{\beta_2} \approx 0.995$. Due to this decoupling, $v_t$ is "blind" to the rising instability, allowing $\lambda_{\max}(\hat{H}_t)$ to continue rising well above $2/\eta$. This sustained violation accumulates energy until epoch 845, when the gradient finally grows large enough to dominate the momentum term in $v_t$. Only then does $v_t$ increase, pulling $\lambda_{\max}(\hat{H}_t)$ back below $2/\eta$ and allowing the loss to recover.

**Coupling Prevents Spikes (Oscillations).** In contrast, Fig. 3(c-d) illustrates a "coupled" example using $\beta_1 = 0.6$ and $\beta_2 = 0.5$. Here, $\sqrt{v_t}$ remains tightly coupled with the gradient norm throughout training. As $\lambda_{\max}(\hat{H}_t)$ approaches $2/\eta$, the highly responsive $v_t$ (due to low $\beta_2$) immediately reacts to any increase in gradients. This prevents the effective curvature from significantly exceeding the threshold. Consequently, instead of a catastrophic spike, the system settles into typical Edge of Stability (EoS) oscillations, where $\lambda_{\max}(\hat{H}_t)$ hovers around $2/\eta$.

*Remark* 4.6. Although increasing $\beta_1$ reduces the preconditioned Hessian curvature, this effect is fundamentally limited: as $v_t$ can decay toward zero, the preconditioned curvature can grow unbounded and eventually dominate. Therefore, controlling spikes critically depends on regulating the behavior of $v_t$, rather than $\beta_1$. We empirically observe that $v_t - g_t^2$ decoupling is exacerbated by larger $\beta_2$ as higher $\beta_2$ increases the inertia of the second-moment estimator. In the no-inertia limit ($\beta_1, \beta_2 \to 0$), Adam reduces to SignGD, which is memoryless and thus immune to this decoupling effect —resulting in only oscillations rather than severe spikes.

*Remark* 4.7. (**Intuition for Decoupling**) The decoupling of $v_t$ from $g_t^2$ is a direct consequence of small parameter updates. Consider the effective update $\theta_{t+1} = \theta_t - \eta \frac{g_t}{\sqrt{v_t}}$.

---

[2]This preconditioner jointly incorporates the independent effects of $\beta_1$ and $\beta_2$, unifying the stability threshold at $2/\eta$. While the formulation differs slightly from that in Cohen et al. (2023), the two definitions are essentially equivalent under this dual-preconditioning view.

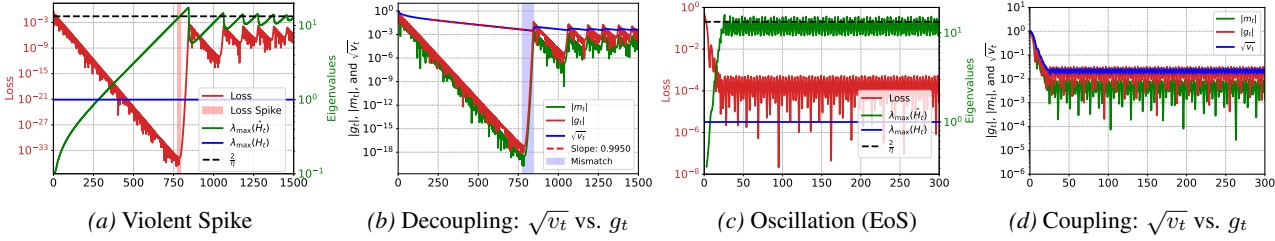

*(a)* Violent Spike  *(b)* Decoupling: $\sqrt{v_t}$ vs. $g_t$  *(c)* Oscillation (EoS)  *(d)* Coupling: $\sqrt{v_t}$ vs. $g_t$

*Figure 3.* Behaviors of Adam on $L(\theta) = \frac{1}{2}\theta^2$. **(a, b) Spike Behavior:** The red span marks the loss spike. The blue span highlights the *mismatch phase*, where the preconditioner $\sqrt{v_t}$ continues to decay autonomously (fitted by red dashed line $\alpha^t$) despite the rising gradient $g_t$. This lag causes sustained instability. **(c, d) Oscillation Behavior:** The preconditioner $\sqrt{v_t}$ remains tightly coupled with $g_t$, resulting in immediate corrections and stable oscillations at the Edge of Stability.

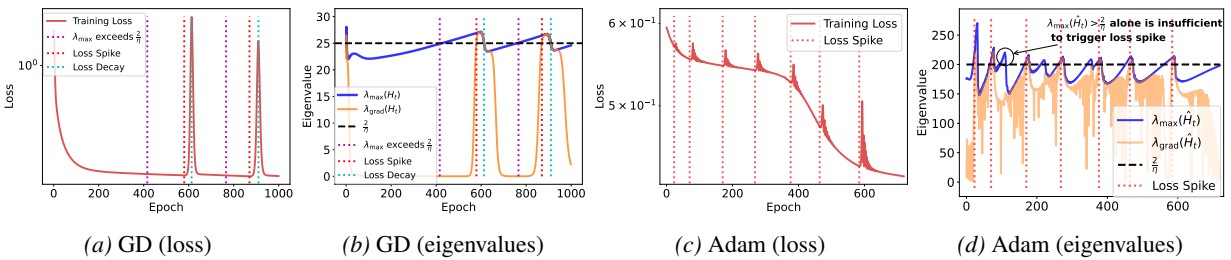

*(a)* GD (loss)  *(b)* GD (eigenvalues)  *(c)* Adam (loss)  *(d)* Adam (eigenvalues)

*Figure 4.* Experimental validation of the gradient-directional loss spike predictor. A two-layer fully connected neural network (width 20) is trained on 200 randomly sampled data points to fit $f(x) = \sin(x) + \sin(4x)$. (a–b) GD with learning rate $\eta = 0.08$. (c–d) Adam with learning rate $\eta = 0.01$, $\beta_1 = 0.9$, $\beta_2 = 0.999$.

When the update magnitude $\frac{g_t}{\sqrt{v_t}}$ is small, the current gradient contribution in $v_t = \beta_2 v_{t-1} + (1 - \beta_2)g_t^2$ becomes negligible compared to the history term. Consequently, the dynamics degenerate to $v_t \approx \beta_2 v_{t-1}$, causing $v_t$ to decouple from the current geometry and decay autonomously.

### 4.3. Precise Loss Spike Prediction via Gradient-Directional Curvature

In high-dimensional optimization, when $\lambda_{\max}(H_t) > 2/\eta$, instability occurs primarily along the corresponding unstable eigendirection, while other directions may remain stable. As shown in our 2D experiments with GD (Figs. D4 and D5), even when $\lambda_{\max}(H_t)$ exceeds $2/\eta$, loss can continue decreasing until oscillations along the unstable direction grow sufficiently large to cause loss increase. Thus, exceeding $\lambda_{\max}(H_t) > 2/\eta$ does not immediately trigger a spike.

To precisely predict loss spikes, we analyze the loss change between consecutive steps using a second-order Taylor expansion: $L(\theta_{t+1}) \approx L(\theta_t) + \nabla L(\theta_t)^\top (\theta_{t+1} - \theta_t) + \frac{1}{2}(\theta_{t+1} - \theta_t)^\top H_t(\theta_{t+1} - \theta_t)$. Substituting the GD update $\theta_{t+1} - \theta_t = -\eta\nabla L(\theta_t)$: $L(\theta_{t+1}) - L(\theta_t) \approx -\eta\|\nabla L(\theta_t)\|^2 + \frac{1}{2}\eta^2\nabla L(\theta_t)^\top H_t\nabla L(\theta_t)$. A loss increase (necessary for a loss spike) occurs when this expression is positive, yielding the condition (see Theorem D.1 for a general result and rigorous proof):

$$\lambda_{\mathrm{grad}}(H_t) := \frac{\nabla L(\theta_t)^\top H_t\nabla L(\theta_t)}{\|\nabla L(\theta_t)\|^2} > \frac{2}{\eta}. \quad (8)$$

Here, $\lambda_{\mathrm{grad}}(H_t) \leq \lambda_{\max}(H_t)$ represents the curvature along the gradient. For Adam, we define the analogous predictor as $\lambda_{\mathrm{grad}}(\hat{H}_t) := \frac{\nabla L(\theta_t)^\top \hat{H}_t\nabla L(\theta_t)}{\|\nabla L(\theta_t)\|^2}$, where $\hat{H}_t$ is the preconditioned Hessian from Eq. (7).

**Experimental Verification.** We validate our predictor using a two-layer network trained to fit $f(x) = \sin(x) + \sin(4x)$, tracking both $\lambda_{\max}(H_t)$ and $\lambda_{\mathrm{grad}}(H_t)$ during training. For GD (Fig. 4(a–b)), two loss spikes occur. At epoch 416, although $\lambda_{\max}(H_t)$ exceeds $2/\eta$, loss continues decreasing. The spike occurs only when $\lambda_{\mathrm{grad}}(H_t)$ also exceeds $2/\eta$. For Adam (Fig. 4(c–d)), 7 distinct spikes occur, while $\lambda_{\max}(\hat{H}_t)$ exceeds $2/\eta$ at 10 time steps. Crucially, spikes occur only when $\lambda_{\mathrm{grad}}(\hat{H}_t) > 2/\eta$, confirming that $\lambda_{\max}(\hat{H}_t)$ alone is insufficient for spike prediction. See verifications of Transformers in Figs. D14 and D22.

## 5. Five-Stage Characterization for Loss Spike Mechanics in Adam

Building on our theoretical and empirical findings, we summarize a five-stage progression that characterizes how loss spikes form and resolve during Adam optimization (Fig. 5).

**Stage 1: Stable Loss Decrease.** Training loss decreases steadily with no abnormalities observed.

**Stage 2: Preconditioner Decay.** As training progresses, gradients in some layers diminish as effective representa-

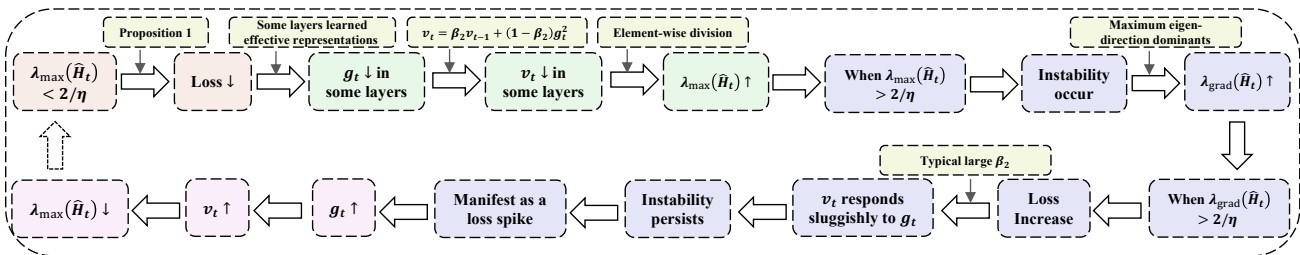

*Figure 5.* Five-stage progression for loss spike mechanics in Adam.

tions are learned. The corresponding second moment estimates $v_t$ also decrease. Due to the element-wise division in Eq. (7), this causes $\lambda_{\max}(\hat{\boldsymbol{H}}_t)$ to gradually increase.

**Stage 3: Spike Onset.** Instability begins when $\lambda_{\max}(\hat{\boldsymbol{H}}_t)$ exceeds the stability threshold $2/\eta$. Initially localized, the instability intensifies as the gradient aligns with max eigen-direction. A loss increase occurs only when the gradient curvature $\lambda_{\mathrm{grad}}(\hat{\boldsymbol{H}}_t)$ also exceeds $2/\eta$. With typical large values $\beta_2 \in [0.95, 0.9999]$, $\boldsymbol{v}_t$ responds sluggishly to gradient information, causing $\lambda_{\mathrm{grad}}(\hat{\boldsymbol{H}}_t)$ to persistently exceed $2/\eta$ and thus manifesting as a dramatic loss spike.

**Stage 4: Preconditioner Growth.** As the spike intensifies, gradients grow larger. When gradients become sufficiently large to influence $\boldsymbol{v}_t$, the decay of $\boldsymbol{v}_t$ halts and reverses. This growth in $\boldsymbol{v}_t$ reduces $\lambda_{\max}(\hat{\boldsymbol{H}}_t)$, helping restore stability.

**Stage 5: Loss Decrease.** When $\lambda_{\max}(\hat{\boldsymbol{H}}_t)$ falls below $2/\eta$, the optimizer regains stability. Loss resumes decreasing, completing the spike cycle and returning to Stage 1.

We also provide a rigorous mathematical five-stage characterization for quadratic optimization:

**Theorem 5.1** (**Five Stages of Adam for Quadratic Optimization** (see App. D Thm. D.2 and Fig. D1 for details and proof)). *Consider the 1D loss* $L(\theta) = \frac{1}{2}\theta^2$, *optimized using Adam with* $\beta_1 = 0$, $\beta_2 \in (0,1)$, *and* $\eta > 0$. *The update rules are:* $\theta_{t+1} = \left(1 - \frac{\eta}{\sqrt{v_t}}\right)\theta_t$, $v_{t+1} = \beta_2 v_t + (1 - \beta_2)\theta_t^2$. *Assume* $v_0 = \theta_0^2$ *and* $|\theta_0| > \frac{\eta}{2}$. *Then there exist integers* $t_0 < t_1 < t_2 < t_3 < t_4 < t_5 < \infty$ *such that the iterates* $(\theta_t, v_t)$ *exhibit the five stages described above in intervals* $[t_i, t_{i+1})$, *respectively.*

Furthermore, we show that common learning rate decay strategies are insufficient to avoid this unstable behavior for sufficiently large $\beta_2$, suggesting its inevitability:

**Theorem 5.2** (**Decaying Learning Rate Scheduler** (see App. D Thm. D.3 for proof)). *Consider the same setup as Thm. 5.1 with decaying learning rate* $\eta_t = \eta_0(t+1)^{-\alpha}$ *where* $\alpha \in (0,1)$. *Assume* $v_0 = \theta_0^2$ *and* $|\theta_0| > 2\eta_0 > 0$. *Assume* $\beta_2$ *is sufficiently close to* 1. *Then the stability condition* $\left|1 - \frac{\eta_t}{\sqrt{v_t}}\right| < 1$ *cannot hold for all* $t \in \mathbb{N}^+$.

# 6. Empirical Validation of Loss Spike Mechanics in Adam

Although we derived the mechanism and indicators from simple model analysis, we find that the proposed loss spike mechanics generalize well to realistic, high-dimensional settings. To track theoretical indicators in practical networks, we compute $\lambda_{\max}$ and $\lambda_{\mathrm{grad}}$ via efficient Hessian-vector products, bypassing full Hessian computation. Experiment details are provided in App. G, with additional validation experiments (including CNN models) in App. F.

## 6.1. FNN for Function Approximation

We trained a two-layer fully connected network on a 50-dimensional function approximation task using Adam with $\beta_1 = 0.9, \beta_2 = 0.999$. The optimization dynamics mirror our quadratic analysis: both loss and gradient norm decrease rapidly before experiencing a sharp spike (Fig. 6(a)).

**Eigenvalue Evolution and Spike Timing:** Fig. 6(b) shows that $\lambda_{\max}(\boldsymbol{H}_t)$ stabilizes quickly while $\lambda_{\max}(\hat{\boldsymbol{H}}_t)$ continues increasing due to decreasing $\boldsymbol{v}_t$ (Fig. 6(c)). Crucially, although $\lambda_{\max}(\hat{\boldsymbol{H}}_t)$ surpasses the stability threshold $2/\eta$ at epoch 179, the spike occurs precisely at epoch 184 when $\lambda_{\mathrm{grad}}(\hat{\boldsymbol{H}}_t)$ exceeds $2/\eta$, confirming the effectiveness.

**Second Moment $v_t$ Dynamics:** Fig. 6(c) shows the evolution of second-moment norms $\sqrt{\hat{\boldsymbol{v}}_t}$ for each parameter block. Before the spike, the gradient norm $\|\boldsymbol{g}_t\| \approx 10^{-2}$ becomes much smaller than $\|\sqrt{\hat{\boldsymbol{v}}_t}\|$, causing $\boldsymbol{v}_t$ to decay automatically at rate $\beta_2$. During the spike, gradient norms increase while $\hat{\boldsymbol{v}}_t$ continues decreasing due to its sluggish response. Once gradients become sufficiently large, $\boldsymbol{v}_t$ rises rapidly, driving $\lambda_{\max}(\hat{\boldsymbol{H}}_t)$ below $2/\eta$ and allowing loss descent to resume at epoch 206.

**Validation of Quadratic Analysis.** The cosine similarity between maximum eigenvectors of $\boldsymbol{H}_t$ across consecutive steps approaches 1 early in training (Fig. 6(d)), validating our quadratic analysis. Fig. 6(e) confirms that spikes occur when gradients align with the maximum curvature direction by projecting the trajectory onto maximum and minimum eigenvectors. To suppress the spike, a straightforward method involves increasing $\varepsilon$ in Eq. (2). As demonstrated

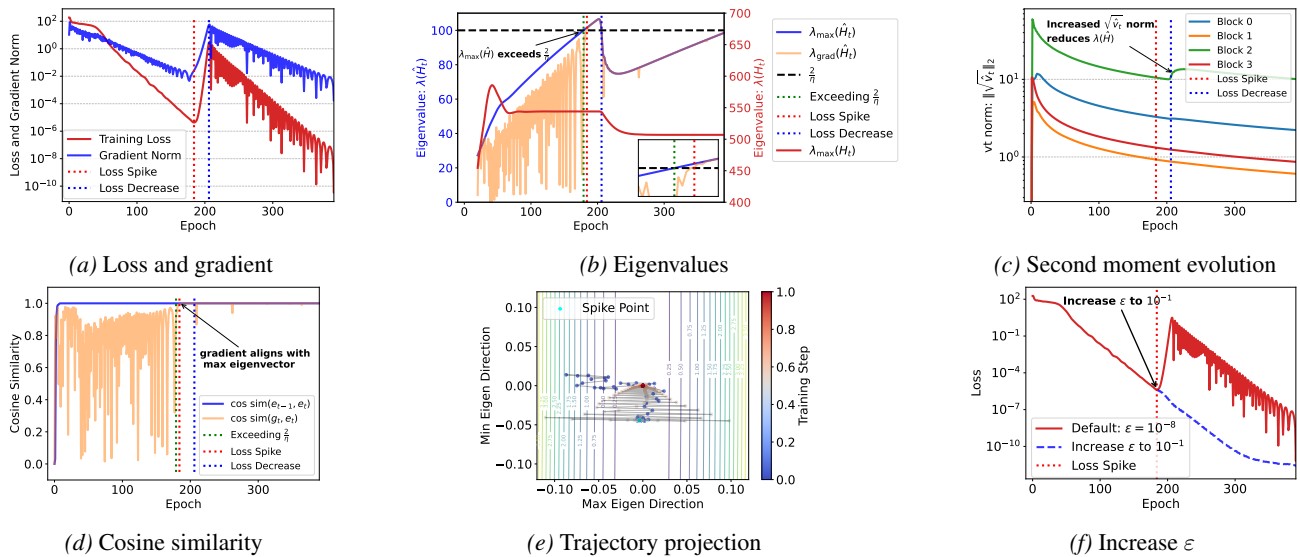

*(a)* Loss and gradient       *(b)* Eigenvalues       *(c)* Second moment evolution

*(d)* Cosine similarity       *(e)* Trajectory projection       *(f)* Increase $\varepsilon$

*Figure 6.* (a) Training loss and gradient norm over time. (b) Evolution of critical eigenvalues: original Hessian maximum eigenvalue $\lambda_{\max}(\boldsymbol{H}_t)$, preconditioned Hessian maximum eigenvalue $\lambda_{\max}(\hat{\boldsymbol{H}}_t)$ and gradient-directional eigenvalue $\lambda_{\mathrm{grad}}(\hat{\boldsymbol{H}}_t)$ relative to $2/\eta$. (c) $L_2$-norm of second moment $||\sqrt{\hat{\boldsymbol{v}}_t}||_2$ of different parameter blocks during training. (d) Cosine similarity between maximum eigenvectors in two consecutive epochs (blue) and between gradient and current maximum eigenvector (orange). (e) Training trajectory projected onto maximum and minimum Hessian eigenvectors at epoch 390. The colorbar for training steps is normalized to the range $[0, 1]$, where 0 corresponds to epoch 28 and 1 corresponds to epoch 390. (f) Increase the default $\varepsilon$ in Eq. (2) to 0.1 at epoch 184.

in Fig. 6(f), increasing $\varepsilon$ to 0.1 at spike onset effectively eliminates the instability.

### 6.2. Transformer Models for Language Tasks

We trained an 8-layer Transformer (approximately 10M parameters) on a synthetic reasoning dataset of 900k sequences (batch size 2048) under the next-token prediction. This dataset is used to investigate the reasoning and memory preferences of Transformer models (Zhang et al., 2025b; 2024; 2025a). Fig. 7(a) shows seven distinct loss spikes. Prior to each spike, the norm of the second-moment estimate $\hat{v}_t$ for the embedding and $\boldsymbol{W}_V$ parameters across attention layers decays automatically at rate 0.999003 ($\approx \beta_2 = 0.999$), followed by a sudden increase in $\|\hat{\boldsymbol{v}}_t\|$ and a sharp drop in loss. Fig. 7(b) describes a typical case where $\lambda_{\mathrm{grad}}(\hat{\boldsymbol{H}}_t)$ exceeds $2/\eta$ causing a spike. However, stochastic batching introduces significant noise, making precise spike prediction challenging. To address this, we define a "sustained spike predictor" as: $\lambda_{\mathrm{grad}}(\hat{\boldsymbol{H}}_t)(\text{sustained}) = \min(\lambda_{\mathrm{grad}}(\hat{\boldsymbol{H}}_{t-1}), \lambda_{\mathrm{grad}}(\hat{\boldsymbol{H}}_t), \lambda_{\mathrm{grad}}(\hat{\boldsymbol{H}}_{t+1}))$. This refined predictor (Fig. 7(b) ) demonstrates perfect correspondence with all seven loss spike occurrences. Sustained periods above threshold trigger loss spikes, which is consistent with the findings in Fig. 3. In addition, we find that directly reducing $\beta_2$ is effective to mitigate loss spikes (Fig. 7(d)).

**Large-Scale Language Model Validation:** We trained a 187M parameter LLaMA-structured transformer on 100B tokens from SlimPajama to validate our mechanics in realistic

large-scale settings. With the default $\beta_2 = 0.999$, training exhibits multiple loss spikes (Fig. 8(a)). Fig. 8(b) examines a representative spike occurring between iterations 14,250-14,400. We observe that the gradient-directional eigenvalue $\lambda_{\mathrm{grad}}(\hat{\boldsymbol{H}}_t)$ exceeds the stability threshold $2/\eta$, signaling the spike onset. Consistent with our proposed mechanism (Sec. 5), gradient norms in certain layers diminish before this spike (Fig. 8(c)). As expected, reducing $\beta_2$ consistently decreases spike frequency during training (Fig. 8(a)), confirming the key role of second-moment in spike formation.

## 7. Conclusion and Discussion

In this work, we provide a mechanistic analysis of loss spikes in Adam. By identifying a critical decoupling between the second moment and current gradients, we reveal the mechanism underlying the persistence of these instabilities. Our theory suggests a simple remedy—reducing $\beta_2$—which we experimentally validate. Encouragingly, many recent large-scale language model studies (Touvron et al., 2023; Dubey et al., 2024; Orvieto & Gower, 2025) have already adopted lower $\beta_2$ values (e.g., 0.95 or lower), further underscoring the practical relevance of our analysis.

**Limitations.** (1) Our rigorous theoretical analysis is established for the 1D quadratic case, and the transfer to high-dimensional non-convex settings rests on empirical validation rather than formal proof. In practical large-scale scenarios, loss landscape geometry and adaptive precondi-

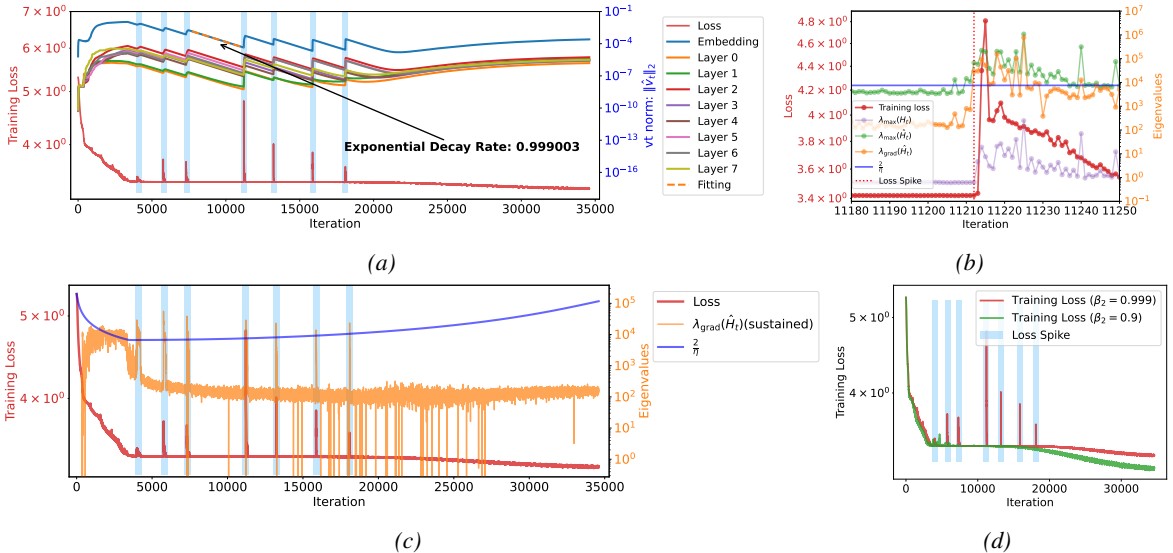

*(a)*                                                      *(b)*

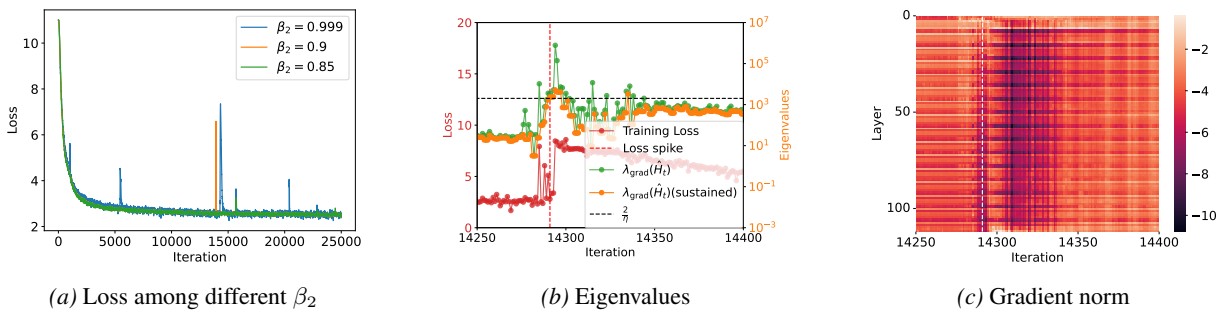

*(c)*                                                      *(d)*

*Figure 7.* (a) Evolution of training loss and second moment $\|\hat{v}_t\|$, with seven spikes highlighted. (b) Eigenvalue analysis near a typical spike. (c) Sustained gradient-directional eigenvalue $\lambda_{\mathrm{grad}}(\hat{H}_t)$(sustained) (orange) versus stability threshold $2/\eta$. The raw $\lambda_{\mathrm{grad}}(\hat{H}_t)$ is shown in Fig. D10. The $2/\eta$ line is plotted against the secondary y-axis on the right for for comparison with the eigenvalues.(d) Reduce the hyperparameter $\beta_2$ in Adam to $0.9$ and retrain.

*(a)* Loss among different $\beta_2$       *(b)* Eigenvalues       *(c)* Gradient norm

*Figure 8.* (a) Training loss evolution for a 187M parameter LLaMA transformer with different $\beta_2$ values. Loss curves show time-weighted EMA smoothing; raw loss appears in Fig. D12. (b) Gradient-directional eigenvalues $\lambda_{\mathrm{grad}}(\hat{H}_t)$ and sustained version $\lambda_{\mathrm{grad}}(\hat{H}_t)$(sustained) during a representative spike (iterations 14,250-14,400) with $\beta_2 = 0.999$. (c) Layer-wise gradient norms (log-scale colorbar) during the spike period. Sudden small gradients are often observed before spikes. See similar findings in Molybog et al. (2023). Sudden diminishing gradients often imply decoupling, where $v_t$ cannot track the gradient quickly via $\boldsymbol{v}_t = \beta_2 \boldsymbol{v}_{t-1} + (1 - \beta_2)\boldsymbol{g}_t^2$.

tioner dynamics likely interact jointly to produce spikes, and cleanly disentangling these contributions remains an open challenge. (2) Additionally, most of our eigenvalue-tracking experiments use full-batch gradients for tractability; the stochastic mini-batch setting introduces noise that complicates the precise timing of spike prediction, as we partially address with the sustained predictor in Sec.6.2. (3) Finally, scaling the Hessian-vector-product-based eigenvalue analysis to models significantly beyond 200M parameters remains computationally demanding, and developing more efficient approximation methods is an important direction for future work.

In addition, loss spikes represent more than optimization phenomena; they may signify transitions between distinct attractor basins in the landscape. Our supplementary ex-

periments in App. E identify four spike types (**neutral**, **benign**, **malignant**, and **catastrophic**) in Transformer training, highlighting the importance of context-specific decisions on whether to suppress or preserve them. Precisely distinguishing between these spike types remains an open challenge.

Beyond hyperparameter adjustments to Adam, alternative spike mitigation techniques include sandwich normalization (Ding et al., 2021; Yin et al., 2025), $\sigma$-Reparam (Zhai et al., 2023), and scaled-decouple distribution (Wang et al., 2025). While some studies (Lyu et al., 2022; Mueller et al., 2023) attribute normalization's effectiveness to sharpness reduction, a deeper understanding of how to leverage or control spikes remains a promising direction for future research.

## Acknowledgements

This work was supported by the National Key R&D Program of China (Grant No. 2022YFA1008200), the National Natural Science Foundation of China (Grant Nos. 92270001, 12371511, 12422119, and 12571567), the Natural Science Foundation of Shanghai (Grant No. 25ZR1402280), and the 2025 Key Technology R&D Program "New Generation Information Technology" Project of the Shanghai Municipal Science and Technology Commission (Grant No. 25511103100).

## Impact Statement

This research provides a comprehensive mechanistic understanding of loss spikes in Adam optimization, a phenomenon that has long challenged the stability of neural network training. By identifying adaptive preconditioners as a primary trigger for these instabilities, our work offers contributions to the training efficiency and reliability of machine learning systems.

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

## A. The Use of Large Language Models(LLMs)

We acknowledge the use of large language models in the preparation of this manuscript. Specifically, we employed LLMs (including but not limited to GPT-4, Claude, and similar models) solely for language polishing and writing enhancement purposes. The LLMs were used to: (i) Improve sentence structure and clarity; (ii) Enhance grammatical accuracy and flow; (iii) Refine technical writing style and consistency; and (iv) Polish language expression while preserving original meaning.

## B. Ethics and Reproducibility Statement

**Ethics Statement.** This work involves theoretical analysis and empirical studies of Adam optimization algorithm using standard neural network architectures and publicly available datasets. All experiments were conducted following established ethical guidelines for machine learning research. No human subjects, sensitive data, or potentially harmful applications were involved in this study.

**Reproducibility Statement.** To ensure reproducibility, we provide detailed experimental configurations in App. G and supplementary experiments in App. F. Our theoretical analysis includes complete mathematical derivations and proofs in App. D. All hyperparameters, network architectures, and training procedures are fully specified. The synthetic datasets and training procedures can be reproduced following the provided specifications. Code and additional implementation details are made available in the supplementary materials.

## C. Limitation and Future Work

Our detailed analysis of loss spikes in Adam optimization reveals that adaptive preconditioners can themselves trigger these phenomena and we verify this mechanism in certain neural network architectures. However, we acknowledge that in more complex scenarios, both the intrinsic geometry of the loss landscape and the applied preconditioners likely interact to jointly produce loss spikes. Disentangling these individual contributions and accurately attributing different spike mechanisms in large-scale models remains a significant challenge for future research.

While we have developed efficient Hessian-vector products to compute gradient-directional eigenvalues without full Hessian computation, computational cost remains a key constraint for scaling this analysis to larger models. Developing efficient algorithms to approximate maximum Hessian eigenvalues and gradient-directional eigenvalues represents a critical direction for future work.

Furthermore, as discussed in App. E, the precise categorization of loss spikes into our proposed taxonomy (**neutral**, **benign**, **malignant**, and **catastrophic** types) presents ongoing challenges. Developing robust, computationally efficient criteria to distinguish between these categories would significantly enhance our ability to detect and appropriately respond to different spike types during training.

## D. Proofs of Theoretical Results

*Proposition* D.1. Let $L : \mathbb{R}^M \to \mathbb{R}$ be twice continuously differentiable. For any iterate $\boldsymbol{\theta}_t$ define the gradient $\boldsymbol{g}_t := \nabla L(\boldsymbol{\theta}_t)$ and, for a fixed learning rate $\eta > 0$, define the local directional maximum Hessian $\bar{\lambda}_t := \sup_{s \in [0,1]} \lambda_{\max}\big(\nabla^2 L(\boldsymbol{\theta}_t - s\eta \boldsymbol{g}_t)\big)$, the maximum eigenvalue of the Hessian along the line segment from $\boldsymbol{\theta}_t$ to $\boldsymbol{\theta}_{t+1} = \boldsymbol{\theta}_t - \eta \boldsymbol{g}_t$. If $\eta < \frac{2}{\bar{\lambda}_t}$, then we have the descent estimate:

$$L(\boldsymbol{\theta}_{t+1}) \leq L(\boldsymbol{\theta}_t) - \eta\Big(1 - \frac{\eta \bar{\lambda}_t}{2}\Big)\|\boldsymbol{g}_t\|^2.$$

In particular, whenever $\eta \in \big(0, 2/\bar{\lambda}_t\big)$ and $\boldsymbol{g}_t \neq 0$ we have strict decrease $L(\boldsymbol{\theta}_{t+1}) < L(\boldsymbol{\theta}_t)$.

*Proof.* Apply the one-dimensional Taylor expansion of the scalar function $\phi(s) := L(\boldsymbol{\theta}_t - s\eta \boldsymbol{g}_t)$ around $s = 0$ up to second order with the remainder written using the Hessian at some point along the segment. Equivalently, use the multivariate Taylor expansion along the direction $-\eta \boldsymbol{g}_t$:

$$L(\boldsymbol{\theta}_t - \eta \boldsymbol{g}_t) = L(\boldsymbol{\theta}_t) - \eta \boldsymbol{g}_t^\top \boldsymbol{g}_t + \frac{\eta^2}{2}\, \boldsymbol{g}_t^\top \Big(\nabla^2 L(\boldsymbol{\theta}_t - s^* \eta \boldsymbol{g}_t)\Big)\boldsymbol{g}_t$$

for some $s^* \in (0, 1)$. Since the symmetric matrix $\nabla^2 L(\boldsymbol{\theta}_t - s^*\eta\boldsymbol{g}_t)$ has largest eigenvalue at most $\bar{\lambda}_t$, we get

$$\boldsymbol{g}_t^\top \left(\nabla^2 L(\boldsymbol{\theta}_t - s^*\eta\boldsymbol{g}_t)\right)\boldsymbol{g}_t \leq \bar{\lambda}_t \|\boldsymbol{g}_t\|^2.$$

Hence

$$L(\boldsymbol{\theta}_{t+1}) \leq L(\boldsymbol{\theta}_t) - \eta\|\boldsymbol{g}_t\|^2 + \frac{\eta^2}{2}\bar{\lambda}_t\|\boldsymbol{g}_t\|^2 = L(\boldsymbol{\theta}_t) - \eta\left(1 - \frac{\eta\bar{\lambda}_t}{2}\right)\|\boldsymbol{g}_t\|^2.$$

If $\eta < 2/\bar{\lambda}_t$, then $1 - \frac{\eta\bar{\lambda}_t}{2} > 0$, so the right-hand side is strictly less than $L(\boldsymbol{\theta}_t)$ whenever $\boldsymbol{g}_t \neq 0$. $\qquad\square$

*Lemma* D.1. Let $\boldsymbol{H}$ be a real symmetric matrix and $\hat{\boldsymbol{H}} = \text{diag}\left(\frac{1}{\sqrt{\hat{\boldsymbol{v}}_t}+\varepsilon}\right)\boldsymbol{H}$. Then $\hat{\boldsymbol{H}}$ is diagonalizable in the field of real numbers.

*Proof.* While $\text{diag}\left(\frac{1}{\sqrt{\hat{\boldsymbol{v}}_t}+\varepsilon}\right)\boldsymbol{H}$ is generally asymmetric, we can demonstrate that it is similar to a symmetric matrix and therefore has real eigenvalues. Let $\boldsymbol{D}_t = \text{diag}\left(\frac{1}{\sqrt{\hat{\boldsymbol{v}}_t}+\varepsilon}\right)$, which is positive definite. We can express:

$$\boldsymbol{D}_t\boldsymbol{H} = \boldsymbol{D}_t^{1/2} \cdot (\boldsymbol{D}_t^{1/2}\boldsymbol{H}\boldsymbol{D}_t^{1/2}) \cdot \boldsymbol{D}_t^{-1/2}$$

Since $\boldsymbol{D}_t^{1/2}\boldsymbol{H}\boldsymbol{D}_t^{1/2}$ is symmetric, $\boldsymbol{D}_t\boldsymbol{H}$ is similar to a symmetric matrix. This confirms that $\boldsymbol{D}_t\boldsymbol{H}$ has real eigenvalues and is diagonalizable. $\qquad\square$

*Proposition* D.2. Consider the three-term recursive iteration

$$\delta\boldsymbol{\theta}_{t+1} = \left[(1 + \beta_1)\boldsymbol{I} - \eta(1 - \beta_1)\boldsymbol{H}(\boldsymbol{\theta}_0)\right]\delta\boldsymbol{\theta}_t - \beta_1\delta\boldsymbol{\theta}_{t-1} - \eta(1 - \beta_1)\nabla L(\boldsymbol{\theta}_0),$$

with learning rate $\eta > 0$ and momentum parameter $\beta_1 \in [0, 1)$. The linearized system at $\boldsymbol{\theta}_0$ is asymptotically stable in a specific eigendirection if and only if the corresponding eigenvalue $\lambda_i$ satisfies:

$$0 < \lambda_i < \frac{2}{\eta} \cdot \frac{1 + \beta_1}{1 - \beta_1}.$$

Consequently, to prevent instabilities that manifest as loss spikes in positive-curvature directions, the maximum positive eigenvalue must satisfy $\lambda_{\max}\left(\frac{1-\beta_1}{1+\beta_1}\boldsymbol{H}(\boldsymbol{\theta}_0)\right) < \frac{2}{\eta}$.

*Proof.* We analyze the stability of the vector recurrence by decomposing it along the eigenspace of the Hessian matrix. Since the Hessian $\boldsymbol{H} := \boldsymbol{H}(\boldsymbol{\theta}_0)$ is diagonalizable, it admits an eigen-decomposition $\boldsymbol{H} = \boldsymbol{P}\boldsymbol{\Lambda}\boldsymbol{P}^{-1}$, where $\boldsymbol{P}$ is an invertible matrix and $\boldsymbol{\Lambda} = \text{diag}(\lambda_1, \ldots, \lambda_d)$ contains the eigenvalues of $\boldsymbol{H}$.

Define the change of variables $\delta\boldsymbol{\theta}_t = \boldsymbol{P}\boldsymbol{z}_t$. Substituting into the recurrence yields

$$\boldsymbol{z}_{t+1} = \left[(1 + \beta_1)\boldsymbol{I} - \eta(1 - \beta_1)\boldsymbol{\Lambda}\right]\boldsymbol{z}_t - \beta_1\boldsymbol{z}_{t-1} - \eta(1 - \beta_1)\boldsymbol{P}^{-1}\nabla L(\boldsymbol{\theta}_0).$$

Since this is a decoupled system in the eigenbasis, for any eigendirection with curvature $\lambda_i$, the $i$-th component $z_t^{(i)}$ satisfies a scalar second-order linear nonhomogeneous recurrence:

$$z_{t+1}^{(i)} = \alpha_i z_t^{(i)} - \beta_1 z_{t-1}^{(i)} + c_i,$$

where

$$\alpha_i := (1 + \beta_1) - \eta(1 - \beta_1)\lambda_i, \quad c_i := -\eta(1 - \beta_1)g^{(i)}, \quad g^{(i)} := \left[\boldsymbol{P}^{-1}\nabla L(\boldsymbol{\theta}_0)\right]_i.$$

The general solution to this nonhomogeneous recurrence is the sum of the homogeneous solution and a particular solution. The homogeneous part is governed by the characteristic equation:

$$r^2 - \alpha_i r + \beta_1 = 0.$$

It is well known (e.g., see Elaydi, *An Introduction to Difference Equations* (Elaydi, 2005)) that the homogeneous solution is asymptotically stable (both characteristic roots lie strictly inside the unit circle in the complex plane) if and only if the following three conditions are met:

$$\begin{cases} 1 - \alpha_i + \beta_1 > 0, \\ 1 + \alpha_i + \beta_1 > 0, \\ \qquad |\beta_1| < 1. \end{cases}$$

Since $\beta_1 \in [0, 1)$ by assumption, the third condition always holds. Substituting $\alpha_i$ into the first condition yields:

$$1 - (1 + \beta_1) + \eta(1 - \beta_1)\lambda_i + \beta_1 > 0 \quad \Longleftrightarrow \quad \eta(1 - \beta_1)\lambda_i > 0 \quad \Longleftrightarrow \quad \lambda_i > 0.$$

This mathematical requirement for $\lambda_i > 0$ reflects the well-established optimization behavior in non-convex landscapes: directions with negative curvature (saddle points) are inherently unstable, causing the optimizer to diverge along those axes to escape the saddle, which effectively *decreases* the loss.

Substituting $\alpha_i$ into the second condition yields:

$$1 + (1 + \beta_1) - \eta(1 - \beta_1)\lambda_i + \beta_1 > 0 \quad \Longleftrightarrow \quad \lambda_i < \frac{2}{\eta} \cdot \frac{1 + \beta_1}{1 - \beta_1}.$$

Unlike saddle-point escape, violating this upper bound in positive-curvature directions causes **oscillatory overshooting**, which forces the parameter out of the local valley and causes a violent *increase* in loss (a loss spike). Therefore, to bound this specific spike-inducing instability, the maximum positive eigenvalue must satisfy:

$$\lambda_{\max}\left( \frac{1 - \beta_1}{1 + \beta_1} \boldsymbol{H} \right) < \frac{2}{\eta}.$$

This completes the proof. $\qquad\square$

*Theorem* D.1 (**Exact Necessary and Sufficient Condition for Loss Spike Onset**). Let $L : \mathbb{R}^M \to \mathbb{R}$ be twice continuously differentiable. At iterate $\boldsymbol{\theta}_t$, denote the gradient $\boldsymbol{g}_t := \nabla L(\boldsymbol{\theta}_t) \neq 0$, and consider a GD update

$$\boldsymbol{\theta}_{t+1} = \boldsymbol{\theta}_t - \eta \boldsymbol{g}_t, \qquad \eta > 0.$$

Define the weighted averaged Hessian along the update direction by

$$\bar{\boldsymbol{H}}_t := 2 \int_0^1 (1 - s) \, \nabla^2 L(\boldsymbol{\theta}_t - s\eta\boldsymbol{g}_t) \, ds,$$

and the corresponding directional curvature by

$$\lambda_{\text{grad}}(\bar{\boldsymbol{H}}_t) := \frac{\boldsymbol{g}_t^\top \bar{\boldsymbol{H}}_t \boldsymbol{g}_t}{\|\boldsymbol{g}_t\|^2}.$$

Then the update exhibits a loss increase (necessary for a loss spike) if and only if $\lambda_{\text{grad}}(\bar{\boldsymbol{H}}_t) > \frac{2}{\eta}$, i.e.,

$$L(\boldsymbol{\theta}_{t+1}) > L(\boldsymbol{\theta}_t) \quad \Longleftrightarrow \quad \lambda_{\text{grad}}(\bar{\boldsymbol{H}}_t) > \frac{2}{\eta}.$$

*Proof.* Consider the univariate function

$$\phi(s) := L(\boldsymbol{\theta}_t - s\eta\boldsymbol{g}_t), \qquad s \in [0, 1].$$

By the chain rule,

$$\phi'(s) = -\eta \, \boldsymbol{g}_t^\top \nabla L(\boldsymbol{\theta}_t - s\eta\boldsymbol{g}_t), \qquad \phi'(0) = -\eta\|\boldsymbol{g}_t\|^2.$$

Differentiating once more yields

$$\phi''(s) = \eta^2 \, \boldsymbol{g}_t^\top \nabla^2 L(\boldsymbol{\theta}_t - s\eta\boldsymbol{g}_t) \, \boldsymbol{g}_t.$$

Since $L$ is twice continuously differentiable, $\phi$ is $C^2$ on $[0, 1]$. The second-order Taylor theorem with *integral remainder* gives the exact identity

$$\phi(1) = \phi(0) + \phi'(0) + \int_0^1 (1 - s)\, \phi''(s)\, ds.$$

Substituting the expressions for $\phi(0)$, $\phi'(0)$, and $\phi''(s)$ yields

$$L(\boldsymbol{\theta}_{t+1}) - L(\boldsymbol{\theta}_t) = -\eta\|\boldsymbol{g}_t\|^2 + \eta^2 \int_0^1 (1 - s)\, \boldsymbol{g}_t^\top \nabla^2 L(\boldsymbol{\theta}_t - s\eta\boldsymbol{g}_t)\, \boldsymbol{g}_t\, ds.$$

Introduce the weighted averaged Hessian

$$\bar{\boldsymbol{H}}_t := \frac{\displaystyle\int_0^1 (1 - s)\, \nabla^2 L(\boldsymbol{\theta}_t - s\eta\boldsymbol{g}_t)\, ds}{\displaystyle\int_0^1 (1 - s)\, ds} = 2\int_0^1 (1 - s)\, \nabla^2 L(\boldsymbol{\theta}_t - s\eta\boldsymbol{g}_t)\, ds.$$

Then the previous equality becomes

$$L(\boldsymbol{\theta}_{t+1}) - L(\boldsymbol{\theta}_t) = -\eta\|\boldsymbol{g}_t\|^2 + \frac{\eta^2}{2}\, \boldsymbol{g}_t^\top \bar{\boldsymbol{H}}_t \boldsymbol{g}_t.$$

Dividing both sides by $\|\boldsymbol{g}_t\|^2 > 0$ shows that the sign of the loss change is exactly the sign of

$$-\eta + \frac{\eta^2}{2}\lambda_{\mathrm{grad}}(\bar{\boldsymbol{H}}_t),$$

where

$$\lambda_{\mathrm{grad}}(\bar{\boldsymbol{H}}_t) := \frac{\boldsymbol{g}_t^\top \bar{\boldsymbol{H}}_t \boldsymbol{g}_t}{\|\boldsymbol{g}_t\|^2}.$$

Therefore,

$$L(\boldsymbol{\theta}_{t+1}) > L(\boldsymbol{\theta}_t) \quad \Longleftrightarrow \quad -\eta + \frac{\eta^2}{2}\lambda_{\mathrm{grad}}(\bar{\boldsymbol{H}}_t) > 0 \quad \Longleftrightarrow \quad \lambda_{\mathrm{grad}}(\bar{\boldsymbol{H}}_t) > \frac{2}{\eta}.$$

This proves the claimed necessary and sufficient condition for a loss spike onset. $\qquad\square$

**Practical proxy for loss spike onset.** The exact loss–spike condition in Theorem D.1 depends on the directional curvature $\lambda_{\mathrm{grad}}(\bar{\boldsymbol{H}}_t)$, where $\bar{\boldsymbol{H}}_t$ is the weighted line–segment average of the true Hessian. Computing $\bar{\boldsymbol{H}}_t$ is intractable in modern deep networks, as it requires access to second-order information along the entire update path. In practice, since learning rates are typically small, we can monitor the step-wise curvature as a proxy:

$$\lambda_{\mathrm{grad}}(\boldsymbol{H}_t) := \frac{\boldsymbol{g}_t^\top \boldsymbol{H}_t \boldsymbol{g}_t}{\|\boldsymbol{g}_t\|^2}.$$

Our central theoretical insight is that Adam can be understood as applying a preconditioning transformation to the Hessian, as expressed in our Equation 7:

$$\hat{\boldsymbol{H}}_t = \frac{1}{1 - \beta_1^t}\frac{1 - \beta_1}{1 + \beta_1}\mathrm{diag}\left(\frac{1}{\sqrt{\hat{\boldsymbol{v}}_t} + \varepsilon}\right)\boldsymbol{H}_t.$$

Therefore, a natural extension for Adam is to replace $\boldsymbol{H}_t$ with the preconditioned Hessian $\hat{\boldsymbol{H}}_t$. This yields our predictor:

$$\lambda_{\mathrm{grad}}(\hat{\boldsymbol{H}}_t) := \frac{\nabla L(\boldsymbol{\theta}_t)^\top \hat{\boldsymbol{H}}_t \nabla L(\boldsymbol{\theta}_t)}{\|\nabla L(\boldsymbol{\theta}_t)\|^2} > \frac{2}{\eta}.$$

Empirically, we observe that this curvature proxy aligns closely with the onset of loss spikes across architectures and datasets, suggesting that it provide a robust approximation to the underlying directional curvature governing spike formation.

*Theorem* D.2 (**Five Stages of Adam for Optimizing Quadratic Loss**). Consider the 1-d quadratic loss $L(\theta) = \frac{1}{2}\theta^2$, optimized using Adam with hyper-parameters $\beta_1 = 0$, $\beta_2 \in (0,1)$, and learning rate $\eta > 0$. The update rules are:

$$\theta_{t+1} = \left(1 - \frac{\eta}{\sqrt{v_t}}\right)\theta_t, \quad v_{t+1} = \beta_2 v_t + (1 - \beta_2)\theta_t^2.$$

Assume the initialization satisfies $v_0 = \theta_0^2$ and $|\theta_0| > \frac{\eta}{2}$. Assume $\frac{1}{\ln(1/\beta_2)} > \frac{1}{\ln(\frac{2|\theta_0|}{\eta})} + \frac{1}{\ln 2}$. Then there exist integers $t_0 < t_1 < t_2 < t_3 < t_4 < t_5 < \infty$ such that the iterates $(\theta_t, v_t)$ exhibit the five stages described above in intervals $[t_i, t_{i+1})$, respectively. Specifically,

(i) **Stable Loss Decrease**. Define $t_0 = 0$, then for all $t_0 \leq t < t_1$, where

$$t_1 := \frac{2\ln\left(\frac{|\theta_0|}{\eta} + \frac{1}{2}\right)}{\ln\frac{1}{\beta_2}},$$

the sequence $|\theta_t|$ decreases exponentially, and $v_t \in [\beta_2^t \theta_0^2, \theta_0^2]$. In particular, there exists $s \in (0,1)$ such that

$$|\theta_t| \leq s^t |\theta_0|, \quad \text{and} \quad |\theta_{t_1}| \leq \delta := s^{t_1}|\theta_0|.$$

(ii) **Preconditioners Decay.** For $t_1 \leq t < t_2$, where

$$t_2 := \inf\left\{t > t_1 \mid \sqrt{v_t} < \frac{\eta}{2}\right\},$$

the momentum $v_t$ decays exponentially as

$$v_t \leq (v_{t_1+1} - \delta^2)\beta_2^{t-t_1-1} + \delta^2.$$

(iii) **Spike Onset.** Define

$$t_3 := \inf\{t > t_2 \mid v_{t+1} > v_t\}.$$

For $t_2 \leq t < t_3$, the preconditioner $v_t$ continues to decay, and the update multiplier $\left|1 - \frac{\eta}{\sqrt{v_t}}\right|$ grows, causing $|\theta_t|$ to increase exponentially.

(iv) **Preconditioners Growth.** Define

$$t_4 := \inf\{t > t_3 \mid \sqrt{v_t} > \frac{\eta}{2}\}.$$

For $t_3 \leq t < t_4$, the growing gradient magnitude forces the preconditioner $v_t$ to increase. Consequently, the update multiplier $\left|1 - \frac{\eta}{\sqrt{v_t}}\right|$ shrinks steadily, preparing the transition from explosive growth to contraction.

(v) **Loss Decrease.** Define

$$t_5 := \inf\left\{t > t_4 : \sqrt{v_t} < \frac{\eta}{2}\right\}.$$

If no such $t$ exists, we simply take $t_5 > t_4$ to be any larger index. For $t_4 \leq t < t_5$, the preconditioner has grown sufficiently so that $\frac{\eta}{\sqrt{v_t}} < 1$. In this regime, the update multiplier satisfies $\left|1 - \frac{\eta}{\sqrt{v_t}}\right| < 1$, ensuring that $|\theta_t|$ contracts and the loss $L(\theta_t) = \frac{1}{2}\theta_t^2$ decreases once again.

*Proof.* We proceed in stages and make all inequalities explicit. The corresponding schematic diagrams of the five stages are shown in Fig. D1.

**Stage 1 (Stable loss decrease).** For the given initialization $v_0 = \theta_0^2$ and $0 < \beta_2 < 1$ we have the trivial lower bound (single-step recurrence gives a simple monotone inequality)

$$v_t \geq \beta_2^t v_0 = \beta_2^t \theta_0^2, \qquad \forall t \geq 0.$$

Also note $v_t \geq 0$ for all $t$.

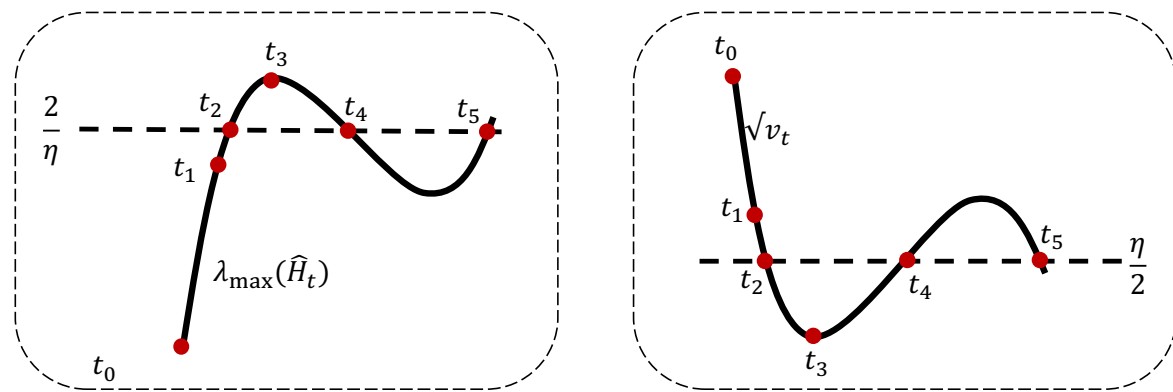

*Figure D1.* The five stages are illustrated schematically.

**Construction of $t_1$ and $\delta$.** Define

$$t_1 := \frac{2\ln\left(\frac{|\theta_0|}{\eta} + \frac{1}{2}\right)}{\ln(1/\beta_2)}.$$

Because $0 < \beta_2 < 1$, $\ln(1/\beta_2) > 0$ and $t_1$ is well defined. Set

$$s := \max\left\{\frac{1}{2}\frac{\eta}{|\theta_0|}, \ \left|1 - \frac{\eta}{|\theta_0|}\right|\right\}.$$

By the hypothesis $|\theta_0| > \eta/2$ we have $s \in (0, 1)$. Define

$$\delta := s^{\lfloor t_1 \rfloor}|\theta_0|.$$

Here, $\lfloor \cdot \rfloor$ is the floor function. The choice of $t_1$ ensures the following inequality chain for all integers $t$ with $t_0 \le t < t_1$. Using the lower bound $v_t > \beta_2^t \theta_0^2$ and the definition of $t_1$, one obtains

$$\sqrt{v_t} \ge \beta_2^{t/2}|\theta_0| \ge \beta_2^{t_1/2}|\theta_0| \quad \text{and by the definition of } t_1, \quad \beta_2^{t_1/2}|\theta_0| = \frac{|\theta_0|}{\frac{|\theta_0|}{\eta} + \frac{1}{2}},$$

so in particular $\sqrt{v_t} > \frac{|\theta_0|}{\frac{|\theta_0|}{\eta} + \frac{1}{2}}$ and hence

$$1 - \frac{\eta}{\sqrt{v_t}} > -\frac{1}{2}\frac{\eta}{|\theta_0|}.$$

Therefore

$$-1 < -\frac{1}{2}\frac{\eta}{|\theta_0|} < 1 - \frac{\eta}{\sqrt{v_t}} < 1, \forall 0 \le t < t_1.$$

This indicates that $|\theta_t|$ is monotonically decreasing for $0 < t < t_1$. Thus, $\sqrt{v_t} \le |\theta_0|$ for all $0 < t < t_1$. This completes the upper bound of $1 - \frac{\eta}{\sqrt{v_t}}$ as follows:

$$-\frac{1}{2}\frac{\eta}{|\theta_0|} < 1 - \frac{\eta}{\sqrt{v_t}} < 1 - \frac{\eta}{|\theta_0|}, \forall 0 \le t < t_1.$$

By definition of $s$, we get

$$\left|1 - \frac{\eta}{\sqrt{v_t}}\right| \le s < 1.$$

Therefore for $0 < t < t_1$,

$$|\theta_t| \le s^t|\theta_0|.$$

In particular $|\theta_{\lfloor t_1 \rfloor}| \le \delta$, establishing the intended bound at the end of Stage 1. This proves Stage 1.

**Stage 2 (Preconditioner decay).** Define

$$t_2 := \inf\left\{ t \in \mathbb{N}^+ : \ 1 - \frac{\eta}{\sqrt{v_t}} < -1 \right\}.$$

For integers $t_1 \le t \le t_2$, we have $|\theta_t| \le |\theta_{t_1}| \le \delta$. The recurrence for $v$ implies

$$v_{t+1} = \beta_2 v_t + (1 - \beta_2)\theta_t^2 \le \beta_2 v_t + (1 - \beta_2)\delta^2.$$

This is an affine linear inequality in $v_t$. Iterating this inequality forward from $t = t_1 + 1$ yields, for any integer $t_1 + 1 \le t \le t_2$,

$$v_t \le (v_{t_1+1} - \delta^2)\beta_2^{t-t_1-1} + \delta^2, \tag{9}$$

which shows $v_t$ decays geometrically toward $\delta^2$ with factor $\beta_2$ so long as $|\theta_t| \le \delta$. Because $|\theta_t| \le \delta$ on the time window following Stage 1 by construction, we have established the Stage 2 statement.

Note also the obvious lower bound obtained by ignoring the additive $(1 - \beta_2)\theta_t^2$ term:

$$v_t \ge v_{t_1+1}\beta_2^{t-t_1-1},$$

so $v_t$ is squeezed between two geometric forms until $|\theta_t|$ leaves the small region.

**Existence and finiteness of $t_2$:** Suppose by contradiction that $t_2 = +\infty$. Then $1 - \frac{\eta}{\sqrt{v_t}} \ge -1$, which simplifies to $v_t \ge \frac{\eta^2}{4}, \forall t \in \mathbb{N}^+$. In Eq. (9) let $t \to +\infty$, it follows that $\limsup_{t\to\infty} v_t \le \delta^2$. So $\delta^2 \ge \frac{\eta^2}{4}$, which indicates that $\delta \ge \frac{\eta}{2}$. Since $\delta := s^{\lfloor t_1 \rfloor}|\theta_0|$, we have $\lfloor t_1 \rfloor \le \frac{\ln(\frac{2|\theta_0|}{\eta})}{\ln(1/s)}$. By definition, $s \ge \frac{1}{2}$, so $\lfloor t_1 \rfloor \le \frac{\ln(\frac{2|\theta_0|}{\eta})}{\ln 2}$. By definition of $t_1$, it follows that

$$\frac{\ln(\frac{2|\theta_0|}{\eta})}{\ln(1/\beta_2)} - 1 \le \frac{2\ln(\frac{|\theta_0|}{\eta} + \frac{1}{2})}{\ln(1/\beta_2)} - 1 \le \lfloor t_1 \rfloor \le \frac{\ln(\frac{2|\theta_0|}{\eta})}{\ln 2}.$$

Therefore we have

$$\frac{1}{\ln(1/\beta_2)} \le \frac{1}{\ln(\frac{2|\theta_0|}{\eta})} + \frac{1}{\ln 2},$$

which contradicts the assumption. So $t_2$ is finite.

**Stage 3 (Spike onset).** By definition of $t_2$, at $t = t_2$ we have $\sqrt{v_{t_2}} < \eta/2$. Consequently

$$\left| 1 - \frac{\eta}{\sqrt{v_{t_2}}} \right| > 1,$$

so passing from $t_2$ to $t_2 + 1$ yields

$$|\theta_{t_2+1}| = \left| 1 - \frac{\eta}{\sqrt{v_{t_2}}} \right| |\theta_{t_2}| > |\theta_{t_2}|.$$

Thus $|\theta_t|$ grows for $t$ just after $t_1$.

**Finiteness of $t_3$.** To capture when the second-moment estimate $v_t$ ceases to decay, define

$$t_3 := \inf\{\, t > t_2 : v_{t+1} > v_t \,\}.$$

If no such $t$ exists we set $t_3 = +\infty$. Suppose, for contradiction, that $t_3 = \infty$. Then $v_{t+1} \le v_t$ for all $t \ge t_2$, so $v_t$ is monotonically decreasing and bounded below by 0. Thus the limit

$$v_\infty := \lim_{t\to\infty} v_t$$

exists. Since $v_t \le v_{t_2}$ for all $t \ge t_2$, we obtain

$$\frac{\eta}{\sqrt{v_t}} \ge \frac{\eta}{\sqrt{v_{t_2}}} > 2,$$

hence there exists a constant $q := \frac{\eta}{\sqrt{v_{t_2}}} - 1 > 1$ such that

$$\left| 1 - \frac{\eta}{\sqrt{v_t}} \right| \geq q > 1, \qquad \forall t \geq t_2.$$

By recursion,

$$|\theta_{t_2+k}| \geq q^k |\theta_{t_2}| \to \infty \quad \text{as } k \to \infty.$$

However, the recurrence for $v_t$ is

$$v_{t+1} = \beta_2 v_t + (1 - \beta_2)\theta_t^2.$$

Since $|\theta_t| \to \infty$ and $1 - \beta_2 > 0$, the term $(1 - \beta_2)\theta_t^2 \to \infty$, forcing $v_{t+1} \to \infty$. This contradicts the assumption that $v_t$ is monotonically decreasing with a finite limit $v_\infty$. Therefore, $t_3 < \infty$. The larger $\beta_2$ is, the more slowly $v_t$ responds to $g_t$, and the later the index $t_3$ of the monotonic change will occur.

**Exponential growth in loss for $t_2 \leq t < t_3$.** For any $t_2 \leq t < t_3$, we have $v_{t+1} \leq v_t \leq v_{t_2}$. Hence

$$\frac{\eta}{\sqrt{v_t}} \geq \frac{\eta}{\sqrt{v_{t_2}}} > 2,$$

and so

$$\left| 1 - \frac{\eta}{\sqrt{v_t}} \right| \geq q > 1,$$

where $q = \frac{\eta}{\sqrt{v_{t_2}}} - 1$. By induction,

$$|\theta_t| \geq q^{t-t_2} |\theta_{t_2}|, \qquad \forall t_2 \leq t < t_3.$$

Thus $|\theta_t|$ grows at least exponentially on the interval $[t_2, t_3)$, and the loss

$$l(\theta_t) = \tfrac{1}{2}\theta_t^2$$

increases dramatically, capturing the onset of the spike.

**Stage 4 (Preconditioner growth).** Define

$$t_4 := \inf\{\, t > t_3 \mid \sqrt{v_t} > \tfrac{\eta}{2} \,\}.$$

**Finiteness of $t_4$.** We first show that $t_4 < +\infty$. Suppose, for contradiction, that $t_4 = +\infty$. By the definition of $t_3$, we have $v_{t_3+1} > v_{t_3}$. Since

$$v_{t_3+1} = \beta_2 v_{t_3} + (1 - \beta_2)\theta_{t_3}^2,$$

this inequality implies $\theta_{t_3}^2 > v_{t_3}$. On the other hand,

$$\theta_{t_3+1} = \left( 1 - \frac{\eta}{\sqrt{v_{t_3}}} \right)\theta_{t_3}.$$

If $\theta_{t_3} > 0$, then

$$\theta_{t_3+1} < (1 - \frac{\eta}{\theta_{t_3}})\theta_{t_3} = \theta_{t_3} - \eta,$$

so either $\theta_{t_3} > \frac{\eta}{2}$ or $\theta_{t_3+1} < -\frac{\eta}{2}$. Thus, in either case, there exists some $t \in \{t_3, t_3 + 1\}$ such that

$$|\theta_t| > \tfrac{\eta}{2}.$$

Now assume $t_4 = +\infty$. Then by definition we must have $\sqrt{v_t} \leq \frac{\eta}{2}$ for all $t > t_3$. Hence

$$\left| 1 - \frac{\eta}{\sqrt{v_t}} \right| \geq 1,$$

implying that $|\theta_t|$ is monotonically non-decreasing. Since at least one of $|\theta_{t_3}|$ or $|\theta_{t_3+1}|$ already exceeds $\frac{\eta}{2}$, it follows that

$$|\theta_t| \geq a := \max\{|\theta_{t_3}|, |\theta_{t_3+1}|\} > \tfrac{\eta}{2}, \qquad \forall t > t_3.$$

Thus $|\theta_t|$ converges to a limit (possibly $+\infty$) with

$$\lim_{t\to\infty} |\theta_t| \geq a > \tfrac{\eta}{2}.$$

But then, since

$$v_{t+1} = \beta_2 v_t + (1-\beta_2)\theta_t^2,$$

we must have

$$\lim_{t\to\infty} v_t = a^2,$$

so that

$$\lim_{t\to\infty} \sqrt{v_t} = a > \tfrac{\eta}{2}.$$

This contradicts the assumption that $\sqrt{v_t} \leq \tfrac{\eta}{2}$ for all $t > t_3$. Therefore $t_4$ must be finite.

During the interval $t_3 < t \leq t_4$, the preconditioner $\sqrt{v_t}$ evolves from being strictly below $\tfrac{\eta}{2}$ to exceeding it. We refer to this regime as the "preconditioner growth stage".

**Stage 5 (Loss decrease).** Define

$$t_5 := \inf\left\{ t > t_4 : \ 1 - \frac{\eta}{\sqrt{v_t}} < -1 \right\}.$$

If no such $t$ exists, we simply set $t_5 > t_4$ to be any larger index for convenience. At time $t_4$, the preconditioner satisfies $\sqrt{v_{t_4}} > \tfrac{\eta}{2}$. Hence, for $t \geq t_4$,

$$\left| 1 - \frac{\eta}{\sqrt{v_t}} \right| < 1.$$

This ensures that, during the interval $t_4 \leq t \leq t_5$, the multiplicative factor falls strictly within $(-1, 1)$, so $|\theta_t|$ no longer grows but instead contracts. Consequently, the loss $L(\theta_t) = \tfrac{1}{2}\theta_t^2$ decreases over this period.

Thus the trajectory transitions from exponential growth (Stage 3) and preconditioner growth (Stage 4) into a contraction regime. In this way, the cycle closes and the dynamics return to behavior of the same type as in Stage 1.

This completes the proof of the five-stage behavior for the quadratic optimization. $\qquad\square$

*Theorem* D.3 (**Analysis of decaying learning rate scheduler**). Consider the same setup as Thm. 5.1 with decaying learning rate $\eta_t = \eta_0(t+1)^{-\alpha}$ where $\alpha \in (0,1)$. Assume the initialization satisfies $v_0 = \theta_0^2$ and $|\theta_0| > 2\eta_0 > 0$. Assume $\beta_2$ is sufficiently close to 1. Then the stability condition $|1 - \frac{\eta_t}{\sqrt{v_t}}| < 1$ cannot hold for all $t \in \mathbb{N}^+$.

*Proof.* Assume by contradiction that $|1 - \frac{\eta_t}{\sqrt{v_t}}| < 1$ holds for all $t \in \mathbb{N}^+$.

**Stage 1 (Loss Decay Stage).** For all $t$, $\beta_2^t v_0 \leq v_t \leq \theta_0^2$. Define $t_0 = \frac{\log 2}{\log \frac{1}{\beta_2}}$. Then for all $t \leq t_0$, $v_t \geq \frac{1}{2}v_0$. Since $|\theta_0| > 2\eta_0$,

we have $\frac{\eta_t}{\sqrt{v_t}} < \frac{\eta_0}{\sqrt{\frac{1}{2}v_0}} < \frac{2\eta_0}{|\theta_0|} < 1$ for all $0 \leq t \leq t_0$. Therefore , $\prod_{k=0}^{t_0}(1 - \frac{\eta_k}{\sqrt{v_k}}) = e^{\sum_{k=0}^{t_0}\log(1-\frac{\eta_k}{\sqrt{v_k}})} \leq e^{-\sum_{k=0}^{t_0}\frac{\eta_k}{\sqrt{v_k}}} \leq$
$e^{-\frac{1}{|\theta_0|}\sum_{k=0}^{t_0}\eta_k} \leq e^{-\frac{\eta_0}{(1-\alpha)|\theta_0|}\left((t_0+2)^{1-\alpha}-1\right)} \leq e^{-\frac{\eta_0}{(1-\alpha)|\theta_0|}\left(t_0^{1-\alpha}-1\right)}$. Therefore $|\theta_t| \leq |\theta_0|e^{-\frac{\eta_0}{(1-\alpha)|\theta_0|}\left(t_0^{1-\alpha}-1\right)}$. By assumption, $s := t_0^{1-\alpha}$ is sufficiently large. Therefore $|\theta_{t_0}| := \delta$ is sufficiently small, whereas $\frac{1}{2}|\theta_0|^2 \leq v_{t_0} \leq |\theta_0^2|$.

**Stage 2 (Decay of the Adaptive Preconditioners).** With the same argument of Theorem D.2(ii), we have

$$v_t \leq (v_{t_0+1} - \delta^2)\beta_2^{t-t_0-1} + \delta^2,$$

Solving $\eta_T = 3\delta$, we have $T = (\frac{\eta_0}{3\delta})^\alpha - 1$. Then $v_T \leq (v_{t_0+1} - \delta^2)\beta_2^{T-t_0-1} + \delta^2$. Therefore

$$\frac{\eta_T}{\sqrt{v_T}} \geq \frac{3\delta}{\sqrt{(v_{t_0+1} - \delta^2)\beta_2^{T-t_0-1} + \delta^2}} = \frac{3}{\sqrt{(v_{t_0+1} - \delta^2)\frac{\beta_2^{T-t_0-1}}{\delta^2} + 1}}.$$

By calculation,

$$\frac{\beta_2^{T-t_0-1}}{\delta^2} = e^{\left(\left(\frac{\eta_0}{3\delta}\right)^\alpha - \frac{\log 2}{\log \frac{1}{\beta_2}} - 2\right)\log \beta_2 - 2\log \delta}.$$

When $\beta_2 \to 1$, $\log \beta_2 \to 0$, $\delta \to 0$, but $\delta$ is of the form $e^{(\frac{c_1}{\log \beta_2})^{c_2}}$ with $c_1, c_2 > 0$. Intuitively, $\delta << \log \beta_2$. From $e^{(\frac{c_1}{\log \beta_2})^{c_2}}$ with $c_1, c_2 > 0$, one may verify that

$$\left( (\frac{\eta_0}{3\delta})^\alpha - \frac{\log 2}{\log \frac{1}{\beta_2}} - 2 \right) \log \beta_2 - 2 \log \delta \to -\infty.$$

So $\frac{\beta_2^{T-t_0-1}}{\delta^2} \to 0$. Thus, $\frac{\eta_T}{\sqrt{v_T}} > 2$ when $\beta_2$ is sufficiently close to 1. This breaks the stability condition. $\square$

# E. Discussion: The Pros and Cons of Loss Spikes

**Connection to Generalization Transitions.** Loss spikes represent more than mere optimization phenomena; they may signify transitions between distinct attractor basins in the optimization landscape. To systematically investigate the relationship between loss spikes and generalization, we conducted controlled experiments using a Transformer model. The model was trained to identify specific anchors within sequences, using a dataset of 2,000 samples (1,800 training, 200 test). We employed full-batch Adam optimization for training (detailed experimental setups and dataset specifications are provided in App. F). By analyzing the differential impacts on training and test losses before and after spike occurrences, we identified four distinct categories of loss spikes:

**(i) Neutral Spikes** (Fig. D2(a)): Both training and test losses resume their normal declining trajectory following the spike, suggesting minimal impact on the overall optimization process.

**(ii) Benign Spikes** (Fig. D2(b)): Prior to the spike, training loss reaches very low values while test loss remains elevated, indicating overfitting. After the spike, test loss decreases rapidly, suggesting improved generalization performance.

**(iii) Malignant Spikes** (Fig. D2(c)): Before the spike, both training and test losses achieve low values. After the spike, while training loss continues to decrease normally, test loss plateaus, indicating deteriorated generalization.

**(iv) Catastrophic Spikes** (Fig. D2(d)): Both training and test losses are low before the spike but neither recovers afterward, signifying a complete breakdown of the optimization process. These findings demonstrate that loss spikes can have context-dependent effects on generalization—sometimes enhancing model performance while in other cases degrading performance.

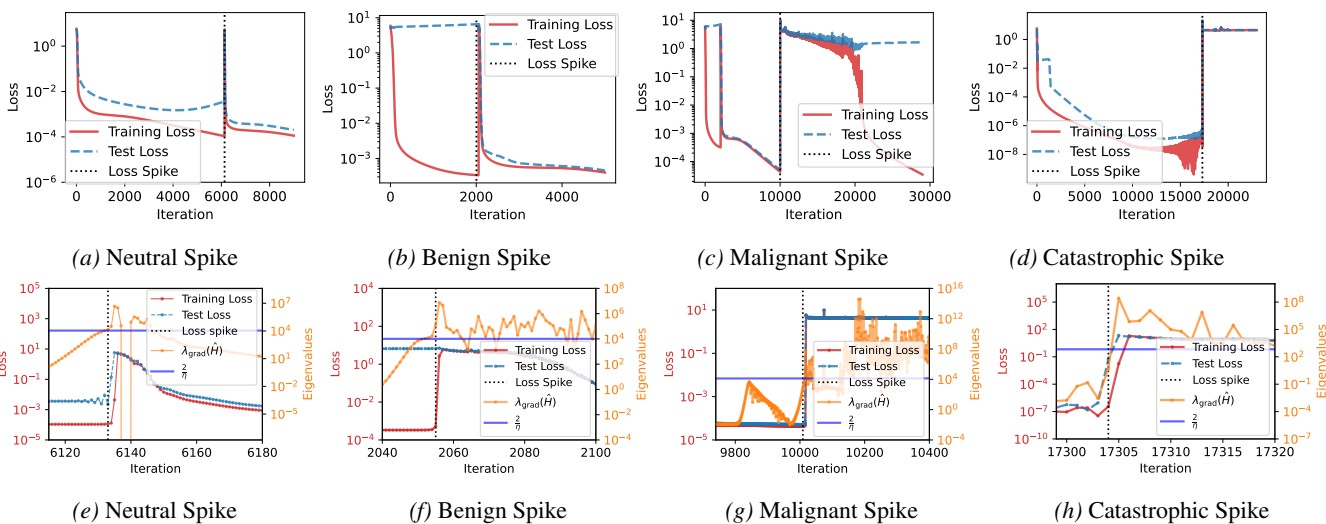

*Figure D2.* The Transformer model was trained to identify specific anchors within sequences. (a–d) Evolution of the training and test losses over the course of training. (e-h) Evolution of the eigenvalues in the gradient direction $\lambda_{\mathrm{grad}}(\hat{H}_t)$ near the spike.

As shown in Fig. D2(e–h), all four types of spikes correspond to our proposed indicator, $\lambda_{\mathrm{grad}}(\hat{H}_t)$, exceeding the classical stability threshold $2/\eta$. Despite this commonality, their effects on generalization differ significantly. While our study uncovers the underlying mechanism that triggers these spikes, determining the precise conditions under which a spike becomes benign or malignant remains an open question for future research.

# F. Supplementary Experiments

**Optimization of Quadratic Function with Varying Hyper-parameters.** For the optimization of a one-dimensional quadratic function, Fig. D3 illustrates the precise location of the spike under various hyperparameter configurations, where $\lambda_{\max}(\hat{\boldsymbol{H}}_t)$ exceeds the stability threshold $\frac{2}{\eta}$.

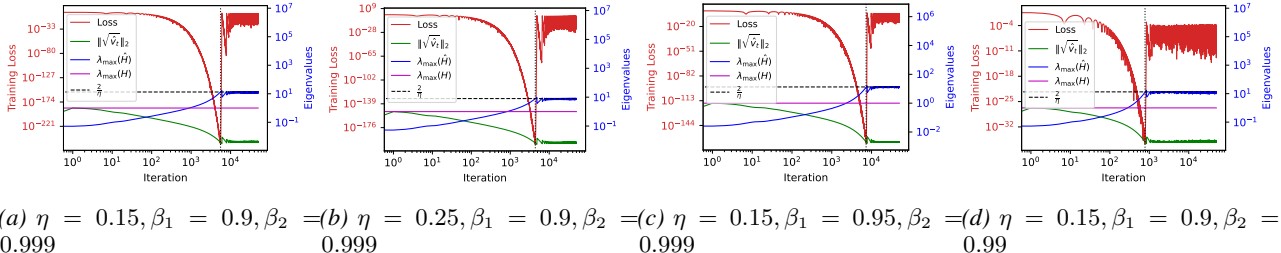

*(a)* $\eta = 0.15, \beta_1 = 0.9, \beta_2 = 0.999$
*(b)* $\eta = 0.25, \beta_1 = 0.9, \beta_2 = 0.999$
*(c)* $\eta = 0.15, \beta_1 = 0.95, \beta_2 = 0.999$
*(d)* $\eta = 0.15, \beta_1 = 0.9, \beta_2 = 0.99$

*Figure D3.* Optimization of $f(\theta) = \frac{1}{2}\theta^2$ using the Adam algorithm with different hyperparameter settings. The solid red line denotes the training loss. The dashed black line indicates the stability threshold $\frac{2}{\eta}$. The blue, purple, and green solid lines represent $\lambda_{\max}(\boldsymbol{H}_t)$, $\lambda_{\max}(\hat{\boldsymbol{H}}_t)$, and the bias-corrected $\|\sqrt{\hat{\boldsymbol{v}}_t}\|_2$, respectively, at each training step.

### Delay Mechanism in GD

To verify that in high-dimensional cases, when $\lambda_{\max} > \frac{2}{\eta}$, the maximum eigenvalue direction oscillates while other eigenvalue directions steadily decrease (resulting in overall loss reduction), we conducted experiments on one and two-dimensional quadratic functions with varying learning rates.

For a one-dimensional quadratic function, the loss landscape curvature remains constant. In this setting, the learning rate initially produces linear improvement over time, followed by gradual decay. When the instability condition is met—as illustrated in Fig. D4(a)—the loss increases immediately.

In contrast, for the two-dimensional case, instability primarily emerges along the dominant eigendirection, while other directions continue to descend stably. As shown in Fig. D4(b), this leads to a delayed onset of the loss spike.

To further validate this mechanism, we visualize the training trajectories in Fig. D5(a–b). In GD (GD), the component along the maximum eigenvalue direction is learned rapidly at first, resulting in a small magnitude. However, once the instability condition is triggered, this component requires significant time to grow and eventually dominate the dynamics.

### Gradient-direction Curvature vs. Update-direction Curvature for Loss Spike Prediction

For Adam, where the Hessian is preconditioned, we define the predictor as

$$\lambda_{\text{grad}}(\hat{\boldsymbol{H}}) := \frac{\nabla L(\boldsymbol{\theta}_t)^{\top} \hat{\boldsymbol{H}} \nabla L(\boldsymbol{\theta}_t)}{\|\nabla L(\boldsymbol{\theta}_t)\|^2},$$

where $\hat{\boldsymbol{H}}$ denotes the preconditioned Hessian in Eq. (7).

We also define

$$\lambda_{\text{update}}(\hat{\boldsymbol{H}}) := \frac{\boldsymbol{u}_t^{\top} \hat{\boldsymbol{H}} \boldsymbol{u}_t}{\|\boldsymbol{u}_t\|^2},$$

where $\boldsymbol{u}_t = \frac{\hat{\boldsymbol{m}}_t}{\sqrt{\hat{\boldsymbol{v}}_t}+\varepsilon}$ is the update vector.

To validate our quadratic approximation-based predictor, we tracked the eigenvalue evolution of the preconditioned Hessian throughout training. Fig. D6(b) reveals that while $\lambda_{\max}(\boldsymbol{H}_t)$ quickly stabilizes, $\lambda_{\max}(\hat{\boldsymbol{H}}_t)$ continues to increase steadily. Notably, $\lambda_{\max}(\hat{\boldsymbol{H}}_t)$ surpasses the stability threshold $\frac{2}{\eta}$ at epoch 179, yet no immediate spike occurs. At epoch 184, precisely when $\lambda_{\text{grad}}(\hat{\boldsymbol{H}}_t)$ exceeds $\frac{2}{\eta}$, we observe the loss spike depicted in Fig. D6(a). Subsequently, the eigenvalue $\lambda_{\text{update}}(\hat{\boldsymbol{H}}_t)$ in the parameter update direction also exceeds $\frac{2}{\eta}$.

This demonstrates that the eigenvalue in the gradient direction more accurately predicts the onset of the actual spike. The update direction requires time to respond to changes in the gradient. When $\lambda_{\text{update}}$ exceeds $2/\eta$, the loss spike has already occurred.

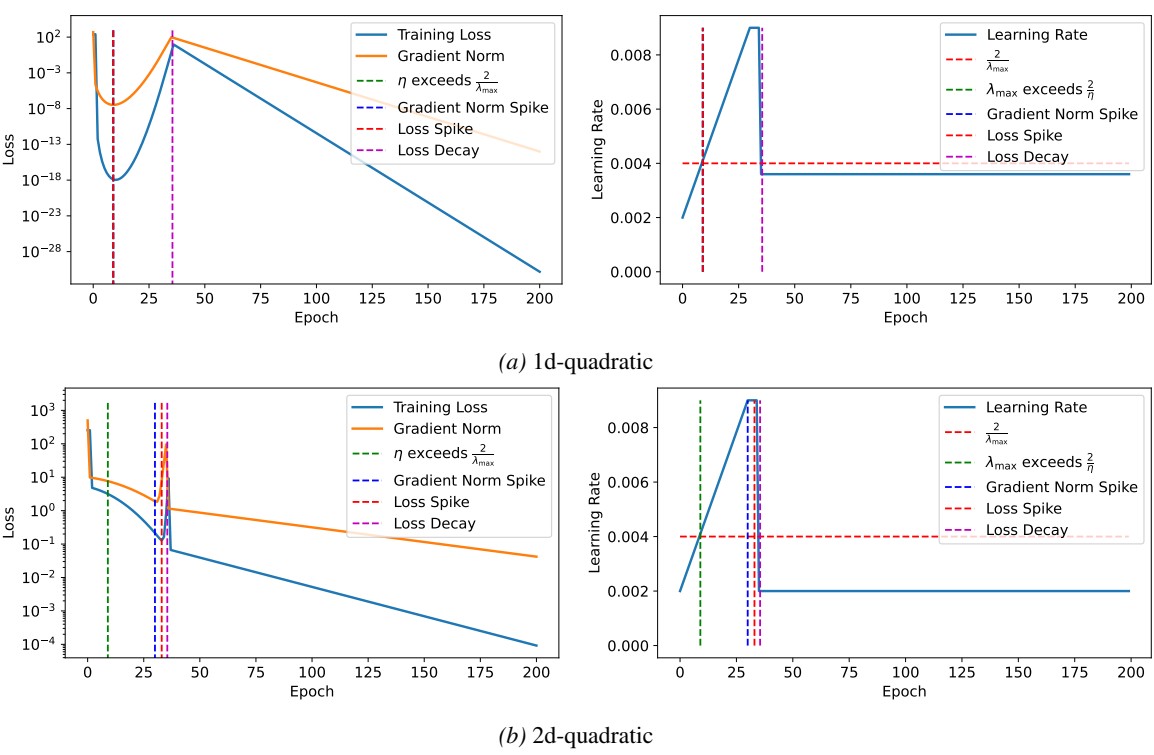

*(a)* 1d-quadratic

*(b)* 2d-quadratic

*Figure D4.* Delay mechanism in GD: Comparison of loss dynamics for 1D and 2D quadratic functions. The learning rate varies over the course of training.

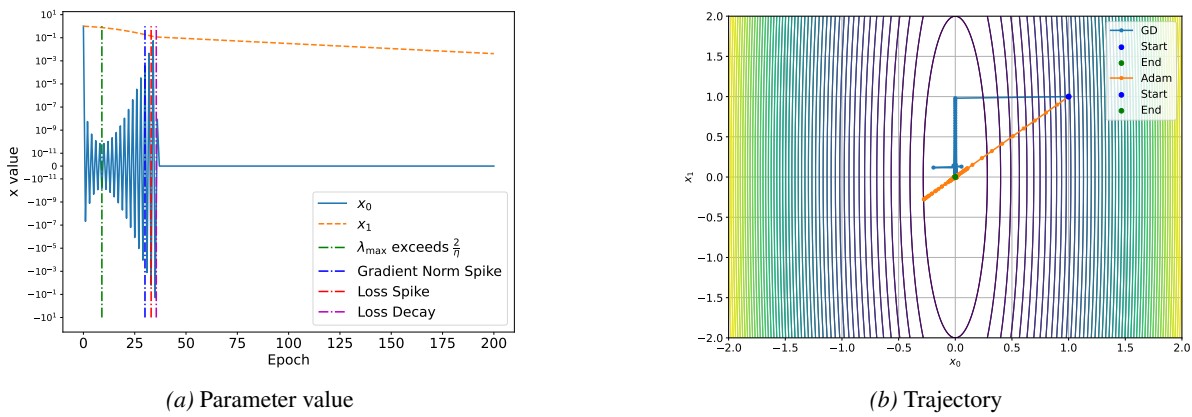

*(a)* Parameter value

*(b)* Trajectory

*Figure D5.* Training dynamics for the 2D quadratic function under GD. (a) Evolution of the solution components along different eigendirections. (b) Optimization trajectory in parameter space.

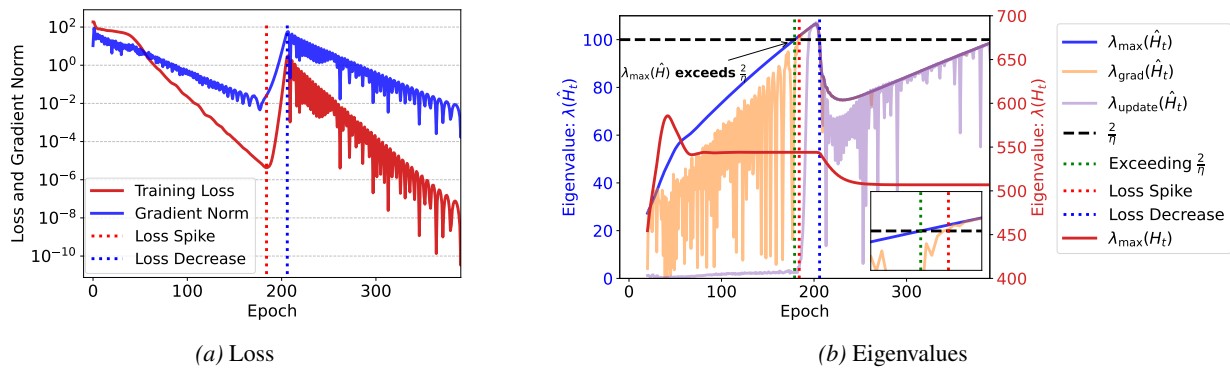

*(a)* Loss        *(b)* Eigenvalues

*Figure D6.* (a) Training loss and gradient norm over time. (b) Evolution of critical eigenvalues: original Hessian maximum eigenvalue $\lambda_{\max}(\boldsymbol{H}_t)$, preconditioned Hessian maximum eigenvalue $\lambda_{\max}(\hat{\boldsymbol{H}}_t)$, gradient-directional eigenvalue $\lambda_{\mathrm{grad}}(\hat{\boldsymbol{H}}_t)$ and update-directional eigenvalue $\lambda_{\mathrm{update}}(\hat{\boldsymbol{H}}_t)$ relative to $2/\eta$.

## MNIST Experiment

We trained an MLP on MNIST using Adam hyperparameters $\beta_1 = 0.9, \beta_2 = 0.999$. As shown in Fig. D7(a), the optimization performs with an initial loss decrease followed by three distinct spikes. Analysis of the preconditioned Hessian's eigenvalues (Fig. D7(b)) shows $\lambda_{\max}(\boldsymbol{H}_t)$ remaining below the stability threshold $2/\eta$, while $\lambda_{\max}(\hat{\boldsymbol{H}}_t)$ increases until exceeding it. Loss spikes occur precisely when $\lambda_{\mathrm{grad}}(\hat{\boldsymbol{H}}_t)$ surpasses $2/\eta$. Figs. D7(c-d) show the evolution of squared gradients and second-order moments $\sqrt{\hat{\boldsymbol{v}}_t}$ across parameter blocks. Before spikes, $\|\boldsymbol{g}_t\|$ is much smaller than $\|\sqrt{\hat{\boldsymbol{v}}_t}\|$, with $\hat{\boldsymbol{v}}_t$ decaying exponentially at rate $\approx \beta_2$. During spikes, while $\hat{\boldsymbol{v}}_t$ continues decreasing, the gradient norm increases until substantially impacting $\boldsymbol{v}_t$. Subsequently, $\hat{\boldsymbol{v}}_t$ rises, causing $\lambda_{\mathrm{grad}}(\hat{\boldsymbol{H}}_t)$ to fall below $2/\eta$ and allowing loss descent to resume.

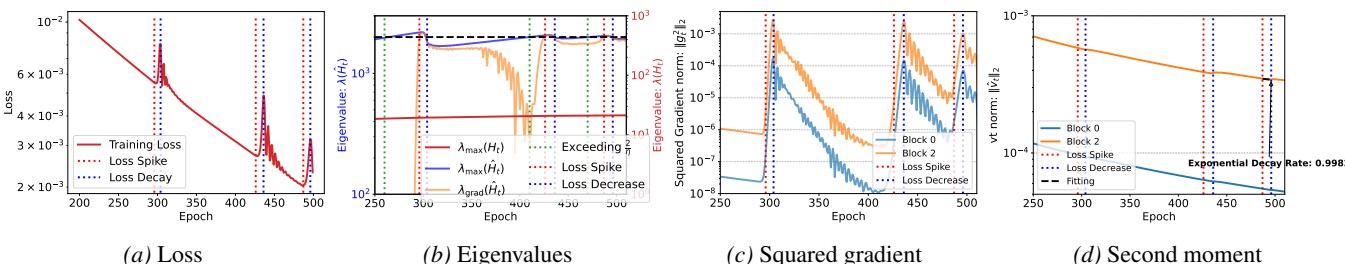

*(a)* Loss     *(b)* Eigenvalues     *(c)* Squared gradient     *(d)* Second moment

*Figure D7.* Training an MLP on 1000 randomly selected MNIST images to illustrate the detailed spikes. (a) Training loss over time. (b) Evolution of eigenvalues: original Hessian maximum eigenvalue $\lambda_{\max}(\boldsymbol{H}_t)$, preconditioned Hessian maximum eigenvalue $\lambda_{\max}(\hat{\boldsymbol{H}}_t)$, and gradient-directional eigenvalue $\lambda_{\mathrm{grad}}(\hat{\boldsymbol{H}}_t)$ relative to $2/\eta$ (black dashed line). (c) Gradient norm evolution across parameter blocks. (d) $L_2$-norm of second moment estimate $\|\hat{\boldsymbol{v}}_t\|$ of different parameter blocks.

## CIFAR-10 Experiments

We trained a convolutional neural network on CIFAR10 using Adam hyperparameters $\beta_1 = 0.9, \beta_2 = 0.999$. As shown in Fig. D8(a), the optimization follows a pattern similar to FNN, with an initial loss decrease followed by three distinct spikes. Analysis of the preconditioned Hessian's eigenvalues (Fig. D8(b)) shows $\lambda_{\max}(\boldsymbol{H}_t)$ remaining below the stability threshold $2/\eta$, while $\lambda_{\max}(\hat{\boldsymbol{H}}_t)$ increases until exceeding it. Loss spikes occur precisely when $\lambda_{\mathrm{grad}}(\hat{\boldsymbol{H}}_t)$ surpasses $2/\eta$. Figs. D8(c-d) show the evolution of squared gradients and second-order moments $\sqrt{\hat{\boldsymbol{v}}_t}$ across parameter blocks. Before spikes, $\|\boldsymbol{g}_t\|$ is much smaller than $\|\sqrt{\hat{\boldsymbol{v}}_t}\|$, with $\hat{\boldsymbol{v}}_t$ decaying exponentially at rate $\approx \beta_2$. During spikes, while $\hat{\boldsymbol{v}}_t$ continues decreasing, the gradient norm increases until substantially impacting $\boldsymbol{v}_t$. Subsequently, $\hat{\boldsymbol{v}}_t$ rises, causing $\lambda_{\mathrm{grad}}(\hat{\boldsymbol{H}}_t)$ to fall below $2/\eta$ and allowing loss descent to resume.

### Transformer Models for Sequence Learning

For the experiment illustrated in Fig. 7, Fig. D10 presents the complete evolution of all eigenvalues, along with detailed views of each spike in Fig. 7(c-e) and Fig. D11(a-d).

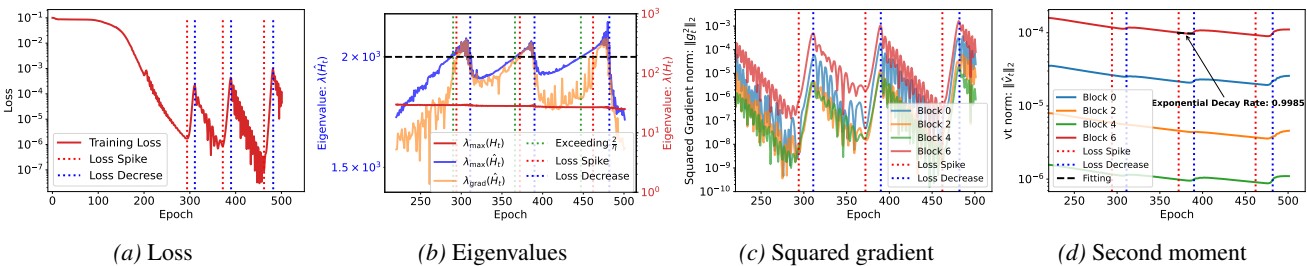

*(a)* Loss      *(b)* Eigenvalues      *(c)* Squared gradient      *(d)* Second moment

*Figure D8.* Training a CNN on 50 randomly selected CIFAR-10 images to illustrate the detailed spikes (see similar result for larger datasets in App. F Fig. D9). (a) Training loss over time. (b) Evolution of eigenvalues: original Hessian maximum eigenvalue $\lambda_{\max}(\boldsymbol{H}_t)$, preconditioned Hessian maximum eigenvalue $\lambda_{\max}(\hat{\boldsymbol{H}}_t)$, and gradient-directional eigenvalue $\lambda_{\text{grad}}(\hat{\boldsymbol{H}}_t)$ relative to $2/\eta$ (black dashed line). (c) Gradient norm evolution across parameter blocks. (d) $L_2$-norm of second moment estimate $\|\hat{\boldsymbol{v}}_t\|$ of different parameter blocks.

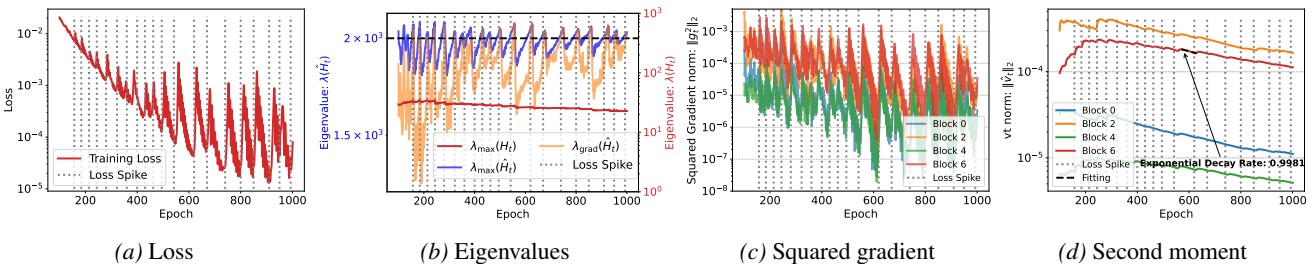

*(a)* Loss      *(b)* Eigenvalues      *(c)* Squared gradient      *(d)* Second moment

*Figure D9.* Loss spike in CNNs on CIFAR10 for randomly sampled 1000 images. (a) Temporal evolution of training loss. (b) Progression of critical eigenvalue metrics: original Hessian maximum eigenvalue $\lambda_{\max}(\boldsymbol{H}_t)$, preconditioned Hessian maximum eigenvalue $\lambda_{\max}(\hat{\boldsymbol{H}}_t)$, and gradient-directional eigenvalue $\lambda_{\text{grad}}(\hat{\boldsymbol{H}}_t)$ relative to the stability threshold $\frac{2}{\eta}$ (black dashed line). (c) Temporal evolution of gradient norm of different parameter blocks. (d) $L_2$-norm of second moment $\|\hat{\boldsymbol{v}}_t\|$ of different parameter blocks.

As depicted in Fig. D11(a-d), we found that transient periods where $\lambda_{\max}(\hat{\boldsymbol{H}}_t)$ and $\lambda_{\text{grad}}(\hat{\boldsymbol{H}}_t)$ exceed $2/\eta$ are insufficient to induce a spike. Loss spikes only materialize when $\lambda_{\text{grad}}(\hat{\boldsymbol{H}}_t)$ remains above the threshold for a sustained duration. This observation aligns with stability analysis principles, which suggest that loss increases exponentially only after persistent instability, with isolated threshold violations being insufficient to trigger rapid loss elevation. Based on this insight, we formulated a "sustained spike predictor" defined as:

$$\lambda_{\text{grad}}(\hat{\boldsymbol{H}}_t)(\text{sustained}) = \min(\lambda_{\text{grad}}(\hat{\boldsymbol{H}}_{t-1}), \lambda_{\text{grad}}(\hat{\boldsymbol{H}}_t), \lambda_{\text{grad}}(\hat{\boldsymbol{H}}_{t+1})).$$

This refined predictor demonstrates perfect correspondence with loss spike occurrences, as shown by the orange line in Fig. D10(b).

### Controlling Adaptive Preconditioners to Eliminate Spikes

We discovered that the epsilon parameter ($\varepsilon$) in Adam plays a critical role in modulating loss spike behavior. Specifically, using a larger $\varepsilon$ significantly reduces spike severity by effectively imposing an upper bound on the preconditioned eigenvalues. Additionally, we experimented with component-wise clipping of $\boldsymbol{v}_t$, where elements falling below a specified threshold are clipped to that threshold value.

As shown in Fig. D13(a), locally increasing $\varepsilon$ during training can effectively suppress loss spikes. Fig. D13(b) further demonstrates that increasing $\varepsilon$ or applying $\boldsymbol{v}_t$ clipping from the beginning of training can also mitigate spike behavior, although this may come at the cost of slower convergence.

### Optimization of Quadratic Function with different adaptive optimizer

We also investigate the spiking behavior of three additional adaptive optimizers on quadratic functions, as illustrated in Figs. D15, D16, and D17. For the adafactor, we use one dimensional tensor form.

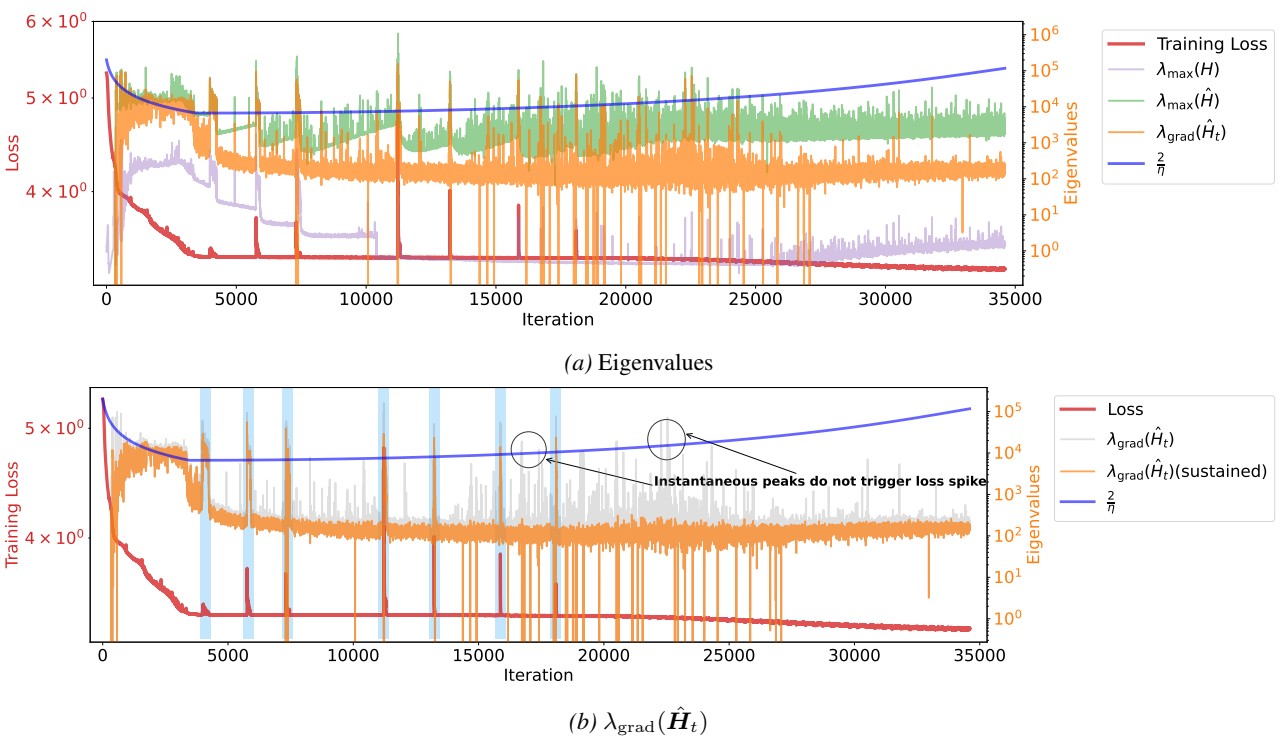

*(a)* Eigenvalues

*(b)* $\lambda_{\mathrm{grad}}(\hat{\boldsymbol{H}}_t)$

*Figure D10.* (a) Evolution of critical eigenvalues: original Hessian maximum eigenvalue $\lambda_{\mathrm{max}}(\boldsymbol{H}_t)$, preconditioned Hessian maximum eigenvalue $\lambda_{\mathrm{max}}(\hat{\boldsymbol{H}}_t)$ and gradient-directional eigenvalue $\lambda_{\mathrm{grad}}(\hat{\boldsymbol{H}}_t)$ relative to $2/\eta$. (b) Gradient-directional eigenvalues $\lambda_{\mathrm{grad}}(\hat{\boldsymbol{H}}_t)$ (gray) and sustained predictor $\lambda_{\mathrm{grad}}(\hat{\boldsymbol{H}}_t)$(sustained) (orange) vs. $2/\eta$.

**Adagrad**

$$\boldsymbol{v}_t = \boldsymbol{v}_{t-1} + \boldsymbol{g}_t^2$$
$$\boldsymbol{\theta}_{t+1} = \boldsymbol{\theta}_t - \eta \frac{\boldsymbol{g}_t}{\sqrt{\boldsymbol{v}_t} + \varepsilon} \tag{10}$$

**RMSProp**

$$\boldsymbol{v}_t = \beta_2 \boldsymbol{v}_{t-1} + (1 - \beta_2)\boldsymbol{g}_t^2$$
$$\boldsymbol{\theta}_{t+1} = \boldsymbol{\theta}_t - \eta \frac{\boldsymbol{g}_t}{\sqrt{\boldsymbol{v}_t} + \varepsilon} \tag{11}$$

**Adafactor**

$$\boldsymbol{v}_t = \beta_2 \boldsymbol{v}_{t-1} + (1 - \beta_2)\boldsymbol{g}_t^2$$
$$\boldsymbol{u}_t = \frac{\boldsymbol{g}_t}{\sqrt{\boldsymbol{v}_t} + \varepsilon_1}$$
$$\hat{\boldsymbol{u}}_t = \frac{\boldsymbol{u}_t}{\max\left(1, \frac{\mathrm{RMS}(\boldsymbol{u}_t)}{d}\right)}$$
$$\rho_t = \max(\varepsilon_2, \mathrm{RMS}(\boldsymbol{\theta}_t)) \tag{12}$$
$$\alpha_t = \eta_t \cdot \rho_t$$
$$\boldsymbol{\theta}_{t+1} = \boldsymbol{\theta}_t - \alpha_t \hat{\boldsymbol{u}}_t$$

# G. Experimental Setup

All experiments were conducted on 1 NVIDIA RTX 4080 GPU. The runtime varied across tasks, ranging from a few minutes for smaller models to several days for large-scale training.

Computing the full Hessian matrix for large-scale neural networks is computationally prohibitive due to its quadratic memory

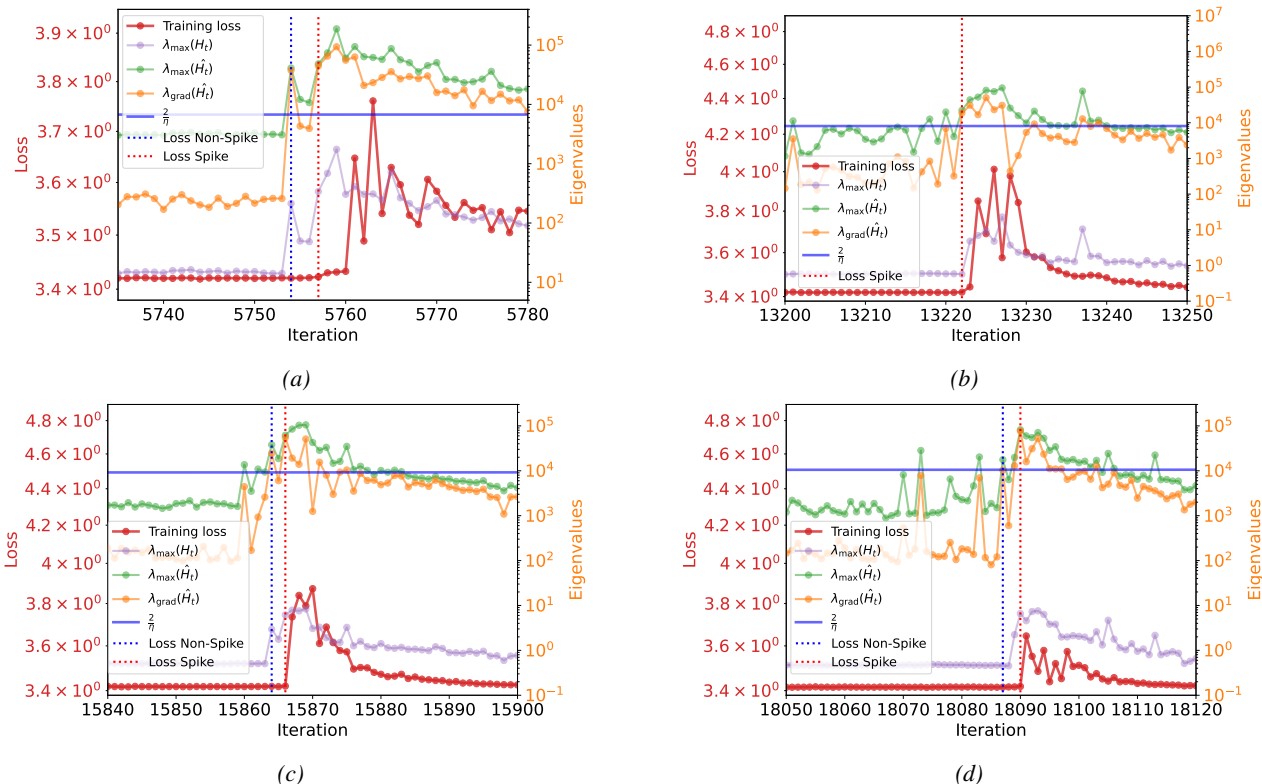

*Figure D11.* Detailed inspection of loss spike intervals showing the maximum eigenvalues of the original Hessian $\lambda_{\max}(\boldsymbol{H}_t)$, preconditioned Hessian $\lambda_{\max}(\hat{\boldsymbol{H}}_t)$, and $\lambda_{\mathrm{grad}}(\hat{\boldsymbol{H}}_t)$.

complexity. To address this challenge, we employ an efficient power iteration method combined with Hessian-vector products that leverages automatic differentiation, circumventing the explicit construction of the complete Hessian matrix.

**Setup for Fig. 6 and Fig. 1(a).** We trained two-layer fully connected neural network applied to a high-dimensional function approximation task. The target function is defined as $f^*(\boldsymbol{x}) = \boldsymbol{w}^{*\top}\boldsymbol{x} + \boldsymbol{x}^\top \mathrm{diag}(\boldsymbol{v}^*)\boldsymbol{x}$, where $\boldsymbol{w}^*, \boldsymbol{v}^* \in \mathbb{R}^{50}$ are the ground-truth parameters and $\boldsymbol{x} \in \mathbb{R}^{50}$ denotes the input features. A total of $n = 200$ data points are sampled, with inputs drawn from a standard Gaussian distribution. Gaussian noise with standard deviation $\varepsilon = 0.1$ is added to the outputs. The network has a hidden layer width of $m = 1000$, placing it in the over-parameterized regime. All weights are initialized from a Gaussian distribution $\mathcal{N}(0, \frac{1}{m})$. Training is performed using full-batch Adam with a learning rate of $\eta = 0.02$, and momentum parameters $\beta_1 = 0.9$, $\beta_2 = 0.999$.

**Setup for Fig. D8 and Fig. 1(b).** We trained a convolutional neural network on the CIFAR-10 dataset. For computational tractability in computing Hessian eigenvalues, we restricted the training set to 50 randomly sampled images. The network contains approximately $500,000$ parameters and is trained using Mean Squared Error (MSE) loss with one-hot encoded labels. Optimization is performed using full-batch Adam with a learning rate of $\eta = 0.001$ and default momentum parameters $\beta_1 = 0.9$, $\beta_2 = 0.999$.

**Setup for Fig. D19(a,b). We trained a ViT on the CIFAR-10 dataset. The ViT consists of $4$ layers and $8$ heads. The embedding dimension is $64$. The network is trained using Mean Squared Error (MSE) loss with one-hot encoded labels. Optimization is performed using full-batch Adam with a learning rate of $\eta = 0.001$ and default momentum parameters $\beta_1 = 0.9$, $\beta_2 = 0.999$.**

**Setup for Fig. D19(c,d). We trained a ResNet on the CIFAR-10 dataset. The network is trained using Mean Squared Error (MSE) loss with one-hot encoded labels. Optimization is performed using full-batch Adam with a learning rate of $\eta = 0.001$ and default momentum parameters $\beta_1 = 0.9$, $\beta_2 = 0.999$.**

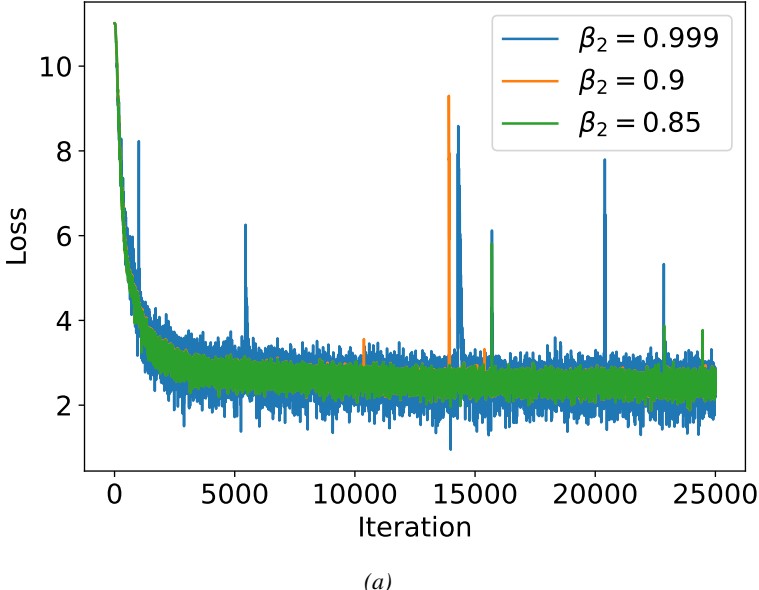

*(a)*

*Figure D12.* The raw loss of the Fig. 8(a).

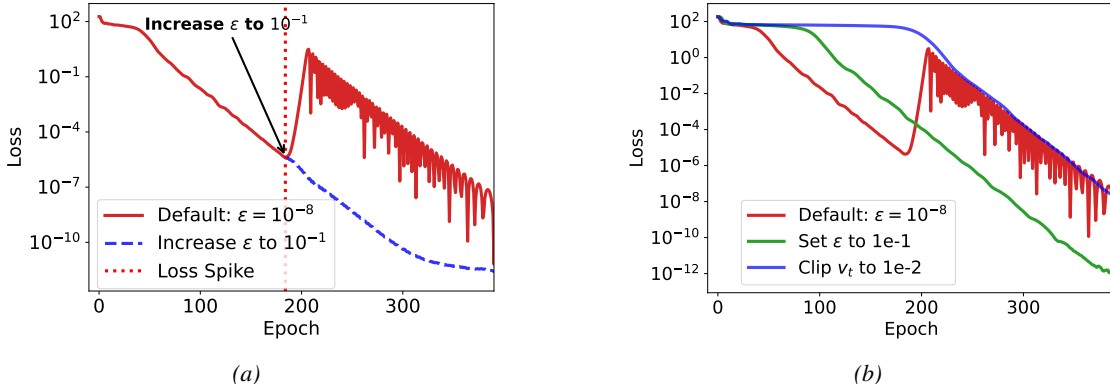

*(a)*          *(b)*

*Figure D13.* The training loss with the same experiment settings as Fig. 6. (a) The only difference of the blue solid line is that we change the $\varepsilon$ in Adam to 0.1 at epoch 184 where the loss in the original training process begin to spike. (b) The green solid line is the training loss that we change the $\varepsilon$ to 0.1 at the beginning of the training. The blue solid line is the training loss that we clip the $v_t$ in Adam to 0.01.

**Setup for Fig. 7 and Fig. 1(d).** We implemented an 8-layer standard Transformer with approximately 10 million parameters. The model is trained on a synthetic dataset designed to learn compositional rules from sequences (Zhang et al., 2025b), consisting of $900,000$ sequences. Training uses a batch size of 2048 and follows the next-token prediction paradigm with cross-entropy loss. The learning rate follows a linear warm-up stage followed by cosine decay. Optimization is performed using Adam with $\beta_1 = 0.9$ and $\beta_2 = 0.999$.

**Setup for Fig. 8 and Fig. D12** We implemented a LLaMA structure Transformer with 187M non-embedding parameters and trained on 100B data split from SlimPajama. The detailed hyperparameters are shown in Table 1.

**Setup for Fig. D2, Fig. D14** and Fig. 1(c). We further evaluate our theoretical insights using 4-layer (Fig. D2, Fig. D14) and 12-layer ((Fig. D2, Fig. 1(c))) standard Transformers trained on a synthetic classification task. The dataset is constructed to learn a specific anchor rule ($3x \rightarrow x$) from sequences (Zhang et al., 2025b), comprising $2,000$ sequences. The model is trained using cross-entropy loss. The learning rate follows a linear warm-up followed by cosine decay. Adam is used for optimization with $\beta_1 = 0.9$ and $\beta_2 = 0.999$.

| Hyperparameter | Value |
|---|---|
| Number of Layers | 16 |
| Hidden Size | 1280 |
| FFN Inner Hidden Size | 1280 |
| Attention Heads | 16 |
| Attention Head Size | 80 |
| Batch Size | 512 |
| Learning Rate Scheduler | 10% Warmup + Cosine Annealing |
| Adam $\beta_1$ | 0.9 |
| Adam $\beta_2$ | 0.999; 0.9; 0.85 |
| Adam $\epsilon$ | $10^{-8}$ |
| Gradient Clipping | 1.00 |

*Table 1.* Detailed Hyperparameters for the 187M Transformer.

**Setup for Fig. D22**    We trained two-layer fully connected neural network applied to a high-dimensional function approximation task. The target function is defined as $f^*(\boldsymbol{x}) = \boldsymbol{w}^{*\top}\boldsymbol{x} + \boldsymbol{x}^\top \mathrm{diag}(\boldsymbol{v}^*)\boldsymbol{x}$, where $\boldsymbol{w}^*, \boldsymbol{v}^* \in \mathbb{R}^{50}$ are the ground-truth parameters and $\boldsymbol{x} \in \mathbb{R}^{50}$ denotes the input features. A total of $n = 200$ data points are sampled, with inputs drawn from a standard Gaussian distribution. Gaussian noise with standard deviation $\varepsilon = 0.1$ is added to the outputs. The network has a hidden layer width of $m = 1000$, placing it in the over-parameterized regime. All weights are initialized from a Gaussian distribution $\mathcal{N}(0, \frac{1}{m})$. Training is performed using full-batch Adam with a learning rate of $\eta = 0.002$, momentum parameter $\beta_1 = 0.0$, and different variations of $\beta_2$.

### G.1. Practical feasibility of our monitoring approach.

For $\lambda_{\max}(\hat{H}_t)$: We do not need the full spectral information or all eigenvalues. To estimate the maximum eigenvalue, we employ the power iteration method, which requires only multiple Hessian-vector products. Specifically, starting from a random vector $\mathbf{v}_0$, power iteration performs:

$$\mathbf{v}_{k+1} = \frac{\hat{H}_t \mathbf{v}_k}{\|\hat{H}_t \mathbf{v}_k\|},$$

and the largest eigenvalue is approximated by $\mathbf{v}_k^\top \hat{H}_t \mathbf{v}_k$. This converges rapidly (typically 5-10 iterations) and each iteration costs only $O(n)$ via automatic differentiation, requiring no explicit Hessian construction. The total cost is $O(kn)$ where $k \ll n$ is the number of power iterations—entirely tractable even for large models.

For our predictor $\lambda_{\mathrm{grad}}(\hat{H}_t)$: The computational cost is even lower. By definition,

$$\lambda_{\mathrm{grad}}(\hat{H}_t) = \frac{g_t^\top \hat{H}_t g_t}{\|g_t\|^2},$$

which requires only a single Hessian-vector product $\hat{H}_t g_t$ in the gradient direction. This is precisely "a single projection", but this is not a limitation—it is exactly the relevant information for predicting loss spikes. We do not need full spectral information; we only need the curvature in the direction the optimizer is moving, which is captured by this single directional derivative.

## H. New Supplementary Experiments

**Compared to research on Edge of Stability (EoS).** Several papers on EoS have noted the close relationship between $\eta$ and $2/\lambda_{\max}(H)$ in modern deep learning as discussed in the main text. However, these phenomena are typically characterized as edge-of-stability behavior, which differs from the large, pronounced loss spikes we observe. The precise relationship between these instabilities and observed spikes remains unclear—instability may manifest as oscillations or as spikes, but the specific mechanism under which spikes occur is not well understood. As shown in our experiment (Figure D14), the system can remain in the EoS region for extended periods, but spikes occur specifically when the curvature in the gradient direction $\lambda_{\mathrm{grad}}(\hat{H}_t)$ exceeds $2/\eta$. Our work reveals how $\lambda_{\mathrm{grad}}(\hat{H}_t)$ increases, how larger $\beta_2$ leads to persistent instability and identifies that spikes occur precisely when the curvature in the gradient direction $\lambda_{\mathrm{grad}}(\hat{H}_t)$ exceeds $2/\eta$, rather than $\lambda_{\max}(H)$ as discussed in EoS literature. To our best knowledge, no prior work has explicitly identified these mechanisms.

**Why understanding quantitative mechanisms matters:** Loss spikes are notoriously difficult to study due to their strong correlations with numerous factors, leading to many seemingly plausible but ambiguous explanations without causal understanding. We emphasize that mechanistic understanding and quantitative prediction are crucial because they typically indicate causality.

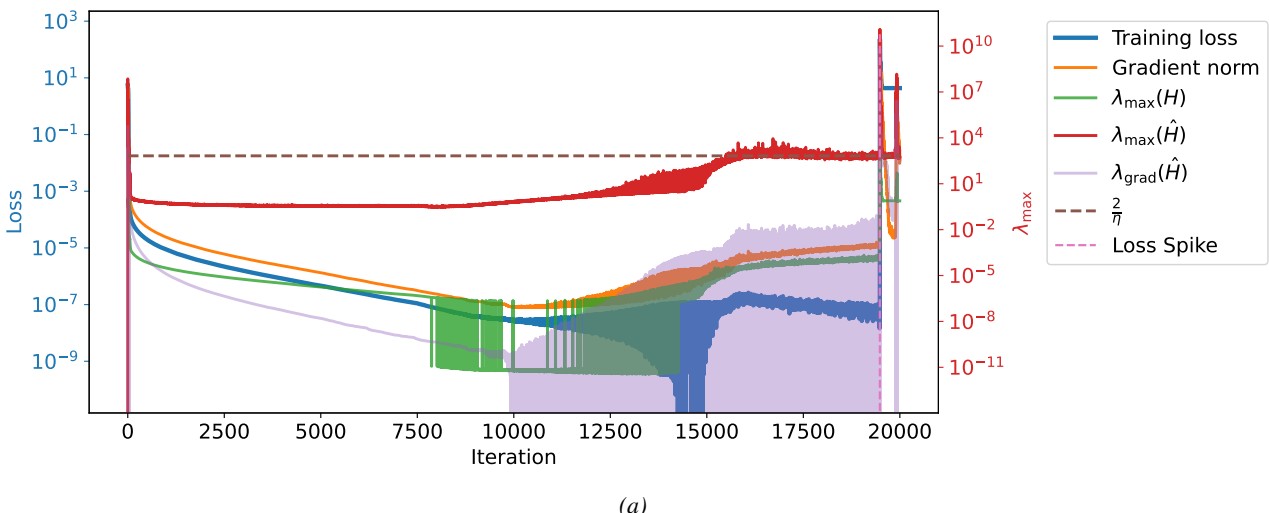

*(a)*

*Figure D14.* (a) Evolution of critical eigenvalues of a $3x \to x$ task (Zhang et al., 2025b): original Hessian maximum eigenvalue $\lambda_{\max}(\boldsymbol{H}_t)$, preconditioned Hessian maximum eigenvalue $\lambda_{\max}(\hat{\boldsymbol{H}}_t)$ and gradient-directional eigenvalue $\lambda_{\mathrm{grad}}(\hat{\boldsymbol{H}}_t)$ relative to $2/\eta$.

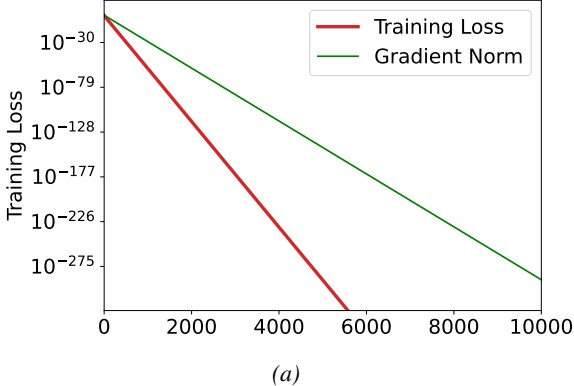

*(a)*

*Figure D15.* Adagrad optimization (Eq: (10))

on $f(\theta) = \frac{1}{2}\theta^2$. AdaGrad's second-moment estimate follows $v_t = v_{t-1} + g_t^2$, which is a strict accumulation. This ensures the effective learning rate $\eta/\sqrt{v_t}$ can only decrease monotonically over time, precluding the possibility of preconditioner decay that our theory identifies as the root cause of spikes.

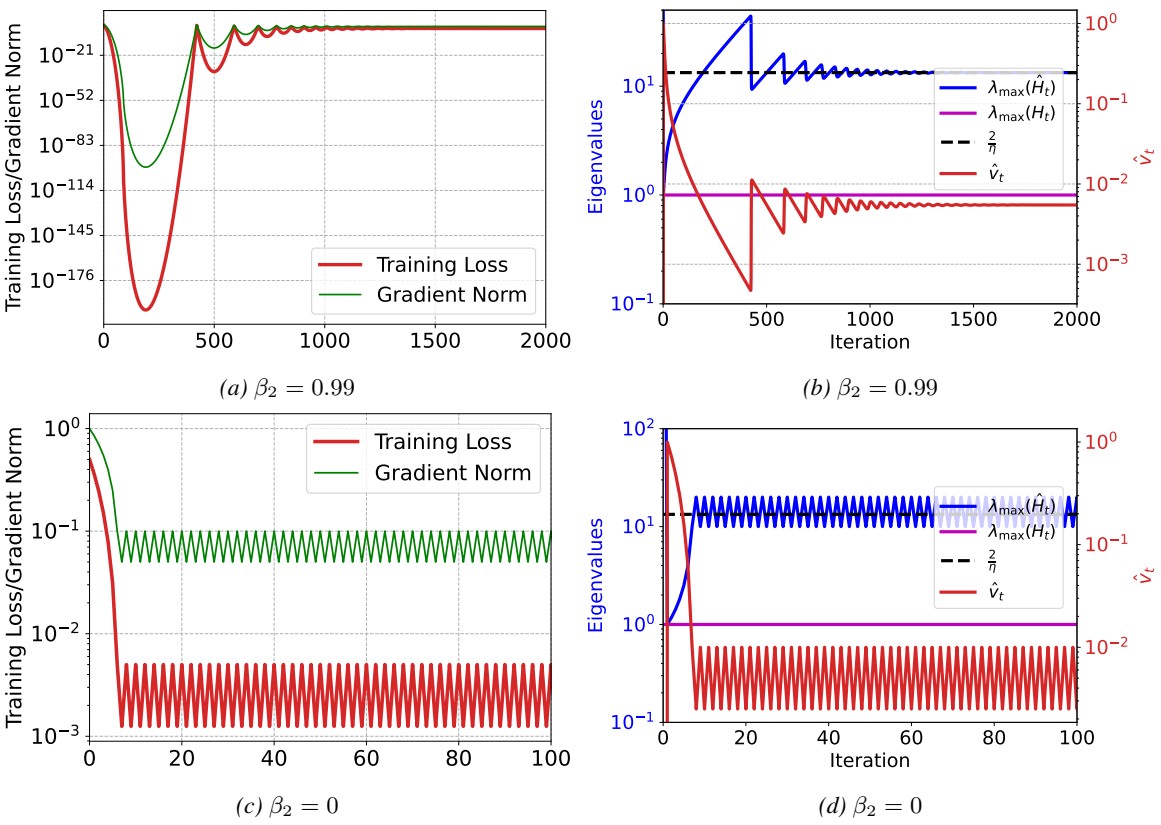

*(a)* $\beta_2 = 0.99$

*(b)* $\beta_2 = 0.99$

*(c)* $\beta_2 = 0$

*(d)* $\beta_2 = 0$

*Figure D16.* RMSProp optimization Eq: (11) ($\beta_1 = 0$ in Adam) on $f(\theta) = \frac{1}{2}\theta^2$ with $\beta_2 = 0.99$ and $0.00$. (a, c) Evolution of training loss and gradient norm. (b, d) Evolution of the second moment estimate $\hat{\boldsymbol{v}}_t$ and the maximum eigenvalue of the preconditioned Hessian.

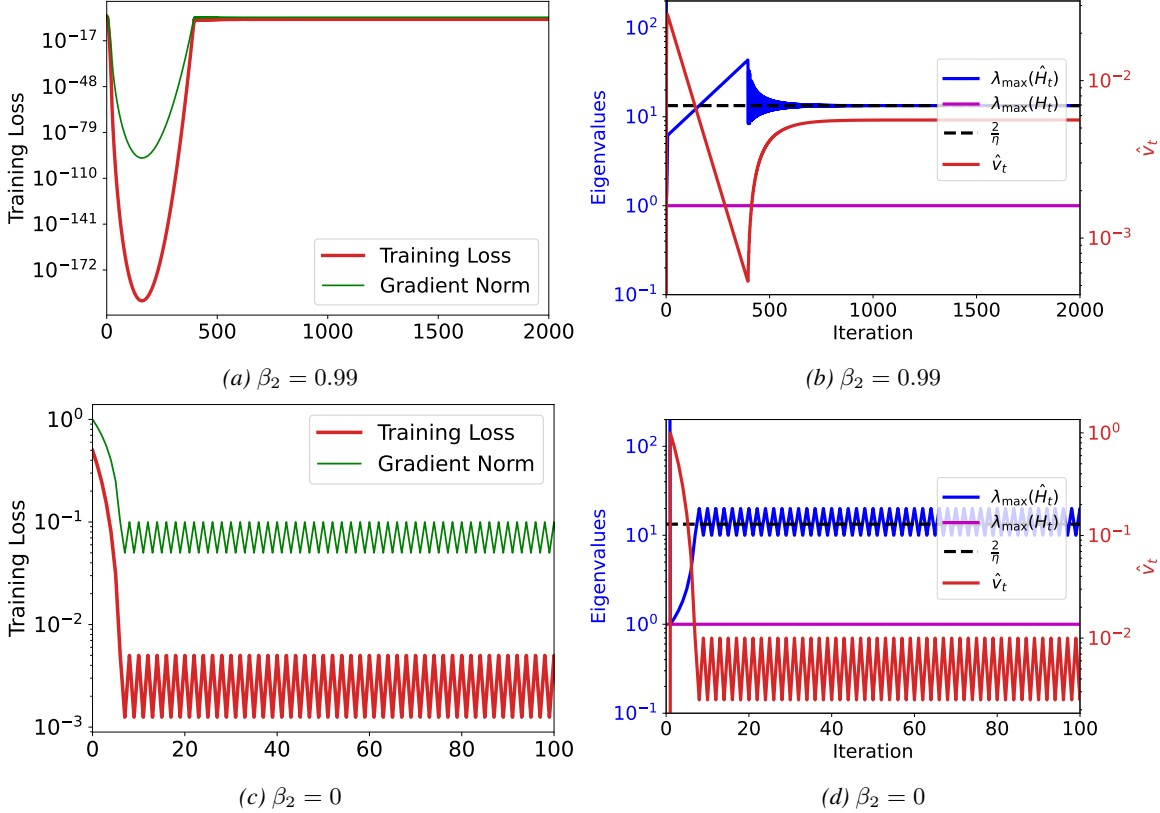

*Figure D17.* Adafactor optimization Eq: (12) on $f(\theta) = \frac{1}{2}\theta^2$ with $\beta = 0.99$ and $0$. For the Adafactor optimizer, we define $\hat{H}_t = \frac{\max(\varepsilon_2, RMS(\theta_{t-1})) * \rho_t}{(\sqrt{\hat{v}_t} + \varepsilon_1)\max(1, RMS(u_t)/d) * \eta}$. In our implementation, we set $\varepsilon_1 = 10^{-30}$ and disable the relative step size. (a, c) Evolution of training loss and gradient norm. (b, d) Evolution of the second moment estimate $\hat{v}_t$ and the maximum eigenvalue of the preconditioned Hessian.

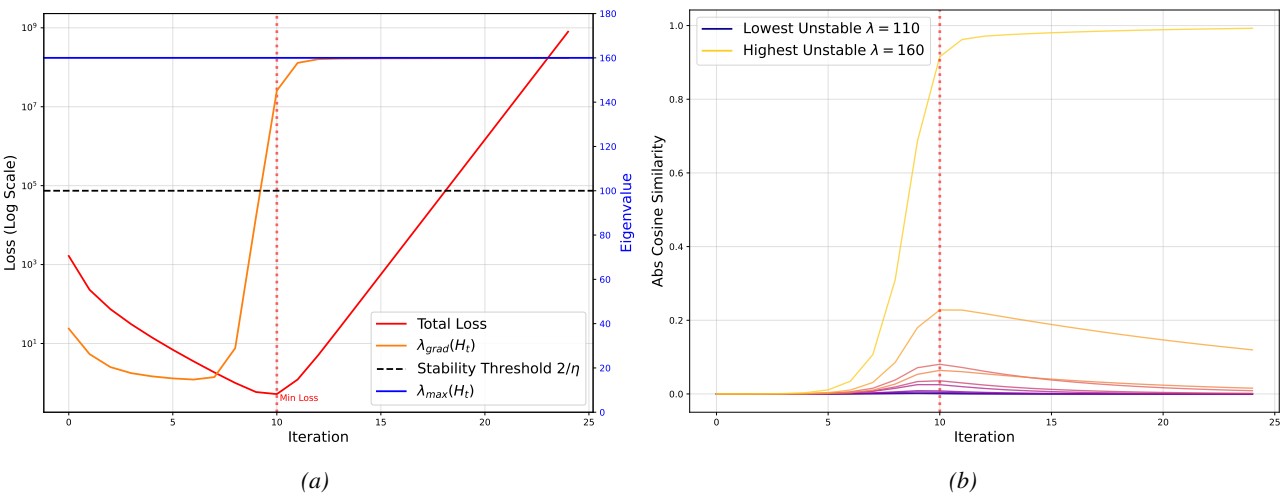

*Figure D18.* The optimization for a 100-dimensional quadratic function with GD. $\eta = 0.02$ and there are 90 stable direction that $\lambda < 100$ and 10 unstable direction that $\lambda > 100$. (a) Evolution of loss and critical eigenvalues: Hessian maximum eigenvalue $\lambda_{\max}(\boldsymbol{H}_t)$ and gradient-directional eigenvalue $\lambda_{\text{grad}}(\boldsymbol{H}_t)$ relative to $2/\eta$. (b) Cosine similarity between gradient direction and 10 unstable directions. When spikes occur, the gradient direction aligns predominantly with the most unstable eigendirection (i.e., the one corresponding to $\lambda_{\max}(\boldsymbol{H}_t)$), as this direction dominates the optimization dynamics.

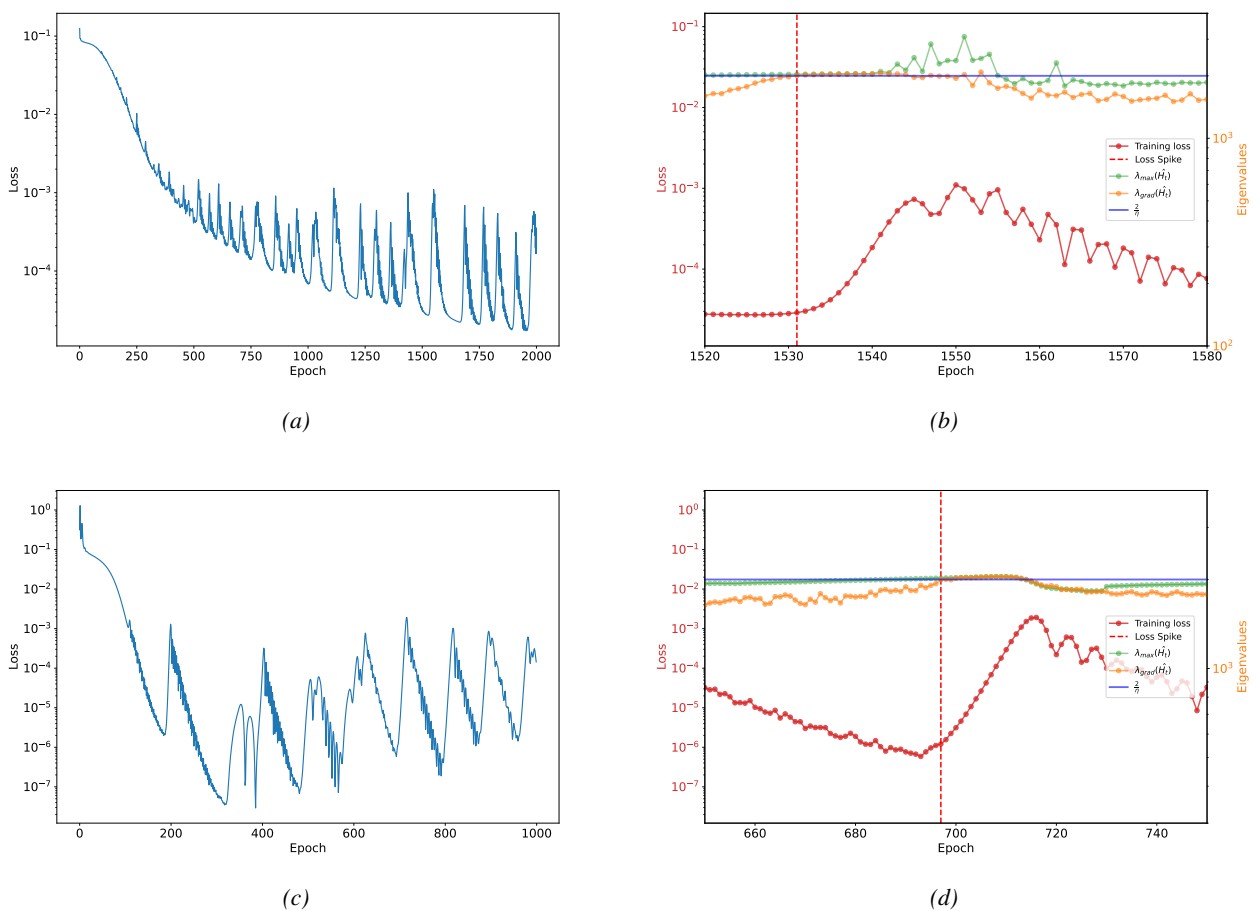

*Figure D19.* (a,c) The training loss of ViT and ResNet18 model on randomly selected 1000 CIFAR-10 images respectively. (b,d) Detailed inspection of loss spike intervals showing the maximum eigenvalues of the preconditioned Hessian $\lambda_{\max}(\hat{\boldsymbol{H}}_t)$, and gradient-directional eigenvalue $\lambda_{\text{grad}}(\boldsymbol{H}_t)$ relative to $2/\eta$.

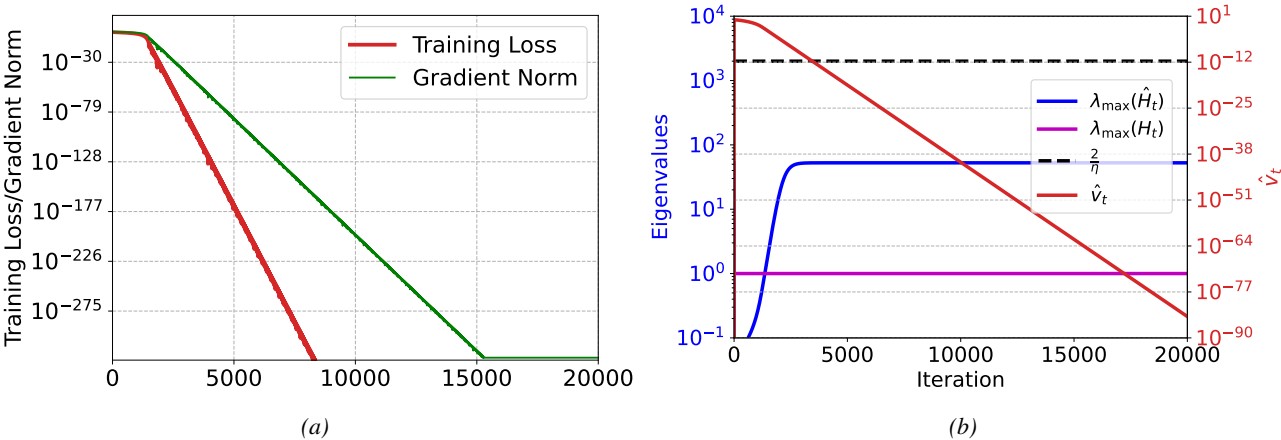

*Figure D20.* The loss spike in Figure 2 is not caused by rounding errors. Adam optimization on $f(\theta) = \frac{1}{2}\theta^2$ with a large $\epsilon = 10^{-3}$ values and learning rate 0.001. (a) Evolution of training loss and gradient norm. (b) Evolution of the second moment estimate $\hat{\boldsymbol{v}}_t$ and the maximum eigenvalue of the preconditioned Hessian. We increase Adam's $\epsilon$ parameter to $10^{-3}$ to ensure that $\lambda_{\text{grad}}(\hat{H}_t)$ can not exceed $2/\eta$, Adam can converge to loss values as low as $10^{-300}$.

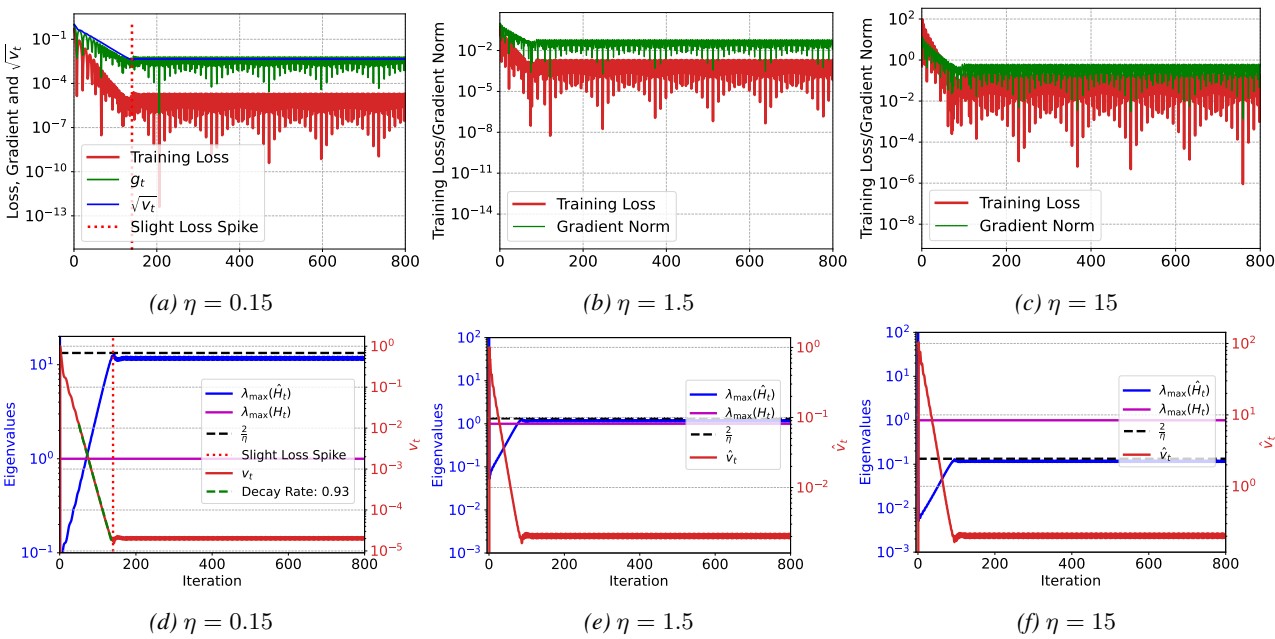

*Figure D21.* Stable loss decrease is still observed initially even with larger learning rates in the case of $\beta_2 = 0.9$. Our results show that when the learning rate is particularly large, $v_t$ grows rapidly in the early stages of optimization. This rapid growth of $v_t$ effectively reduces the preconditioned step size $\eta/\sqrt{v_t}$, which allows the loss to decrease stably at the beginning even under large nominal learning rates.

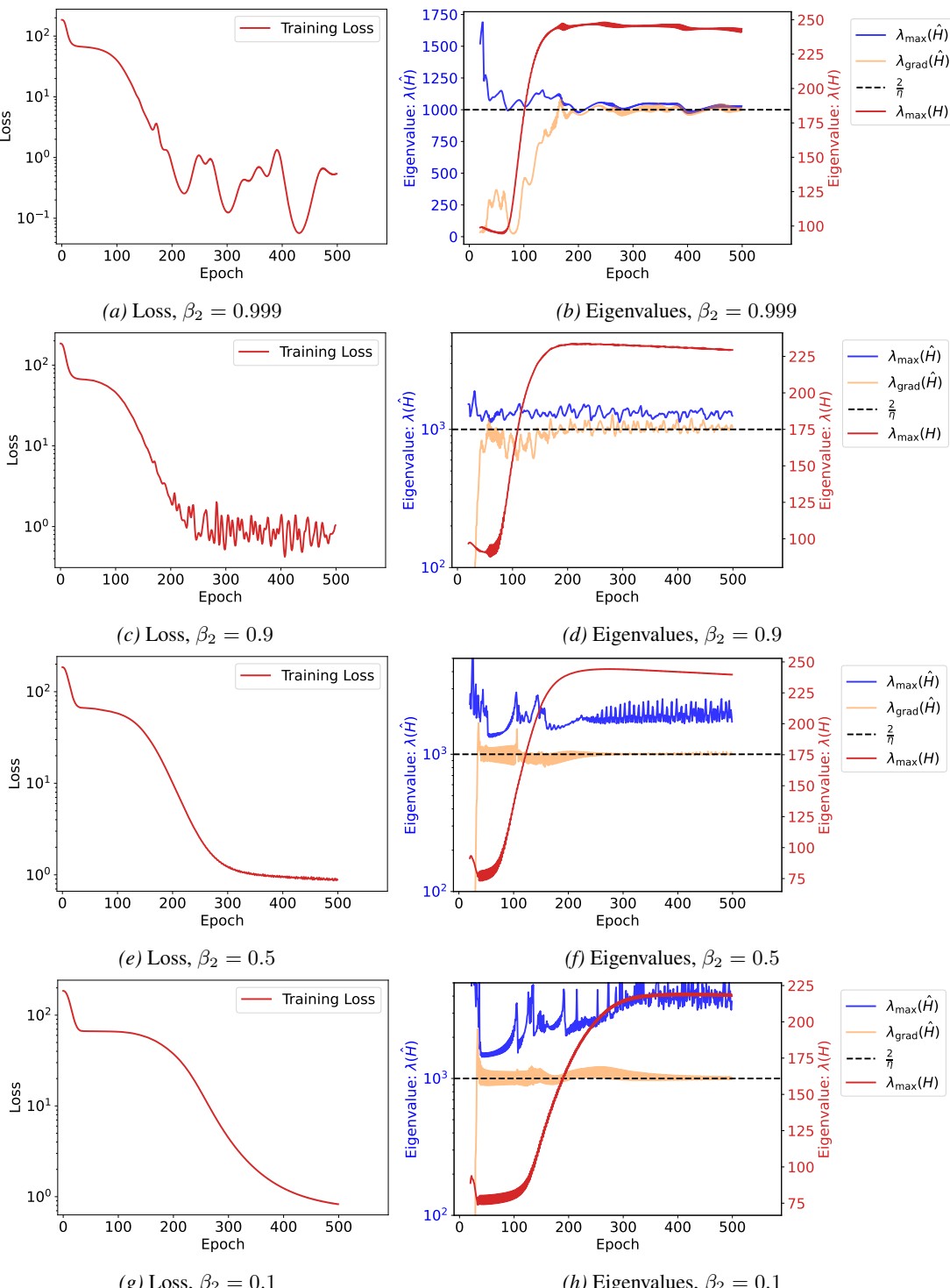

*Figure D22.* Training trajectories and eigenvalue evolution for varying $\beta_2$ values with $\beta_1 = 0$ (to isolate the effect of adaptive learning rate from momentum). Each row shows the loss curve and corresponding evolution of $\lambda_{\max}(\hat{H}_t)$ and $\lambda_{\text{grad}}(\hat{H}_t)$ for a different $\beta_2$ setting. Larger $\beta_2$ values produce more pronounced spikes in the loss, while smaller $\beta_2$ values lead to denser oscillations, mirroring the behavior observed for the quadratic function in Fig. 3. Notably, loss spikes and oscillations correlate with $\lambda_{\text{grad}}$ approaching $2/\eta$, rather than with $\lambda_{\max}(\hat{H}_t)$, providing empirical validation for the practical utility of our proposed $\lambda_{\text{grad}}$ metric.

