# OpenReview forum: "Adaptive Preconditioners Trigger Loss Spikes in Adam"
_ICML.cc/2026/Conference — ICML 2026 regular_

### Official Review · Reviewer_peNU · 2026-02-16

**Soundness:** 3
**Presentation:** 2
**Significance:** 4
**Originality:** 3
**Overall Recommendation:** 4
**Confidence:** 4

**Summary:**

This paper studies the mechanism of loss spikes using Adam. They demonstrates that failing to include timely information in the second moment in Adam will eventually lead to the spiking behavior. They also discuss the effect of $\beta_1,\beta_2$ in creating the decoupling and imply a natural fix of the spike that is to reduce $\beta_2$. Theoretical analysis is performed on a one-dimensional quadratic function to show 5 stages of the spiking behavior. Empirical verifications are also presented on practical language tasks.

**Compliance With Llm Reviewing Policy:**

Affirmed.

**Final Justification:**

This paper studies the mechanism of loss spikes using Adam, and provides a fix to loss spikes by changing parameters. My major concern of this paper is on the presentation side: the previous version slightly underestimated the two key points I mentioned above, and is thus not clear enough. The authors would like to revise the story accordingly, and answered my other questions about the theory (especially about $\beta_1$).

All my concerns have been addressed, and I will maintain my evaluation of this paper.

**Key Questions For Authors:**

- For stage 3 (Theorem 5.2, and also related discussions in the paper), I would expect more explicit technical explanation of why large $\beta_2$, or the decoupling, leads to the increase of 2nd order directional derivative (e.g. due to $\beta_2^t$ etc.). This is not explicitly shown in equation (7) or any theorems.
- In Remark 4.3, the authors comment that larger $\beta_1$ also affect the decoupling. Is this decoupling strictly related to spiking behavior or is this separate things? Because in equation (7), larger $\beta_1$ is actually preferred since it leads to smaller eigenvalues. The current paper is not clear on this point.
- The current presentation is a bit confusing to me, and the whole story could be made more clear. There are two key points that I feel the most important: 1) the mechanism of spikes due to the decoupling of $v_t$ and $g_t$; 2) a fix of spike by reducing $\beta_2$. It seems to me that the paper overemphasize the 2nd order directional derivative ($\lambda_{grad}$), which is important but no surprise (and consequently the insufficiency of landscape geometry, Hessian). The relationship between the mechanism and a natural fix of the spike (although already used in other paper) could be made more rigorous, technical, and more pronounced.

**Limitations:**

yes

**Strengths And Weaknesses:**

**Soundness**
The claims in the paper are well supported. The theoretical analysis is only performed on a one-dimensional quadratic case (I guess due to technical reasons), but the whole story is relatively complete.

**Presentation**
The story is rich but not clear enough. The emphasis in the paper could be adjusted. See questions.

**Significance** and **Originality**
The spiking phenomenon is very important in various tasks and this paper provides an new explanation and a natural solution to it.

---

> ### Author Rebuttal · Authors · 2026-03-28
>
> We sincerely thank the reviewer for high appreciation of the importance of our work. Below, we provide point-by-point clarifications addressing your key concerns. We will fully incorporate these suggestions in the revision to improve the overall narrative.
>
> **Q1: Technical Explanation of Decoupling and Curvature Growth in Stage 3.**
>
> This is a crucial distinction. We need to clarify two sequential physical processes: while the natural decay of gradients inevitably leads to curvature growth, the truly critical effect of “decoupling” is that it renders the adaptive mechanism blind, preventing it from promptly correcting this curvature increase.
>
> **(1) Natural growth of preconditioned curvature:** In Stage 3, regardless of whether decoupling occurs, as the local gradients $g_t$ of certain parameter blocks become increasingly small, $v_t$ decays according to $v_t = \beta_2 v_{t-1} + (1-\beta_2) g_t^2$. As revealed in Eq. (7), the decrease of the denominator $v_t$ necessarily leads to a continuous increase in the preconditioned Hessian curvature, until it exceeds the stability threshold.
>
> **(2) Decoupling disables the adaptive mechanism:** The key question is: *why does the adaptive mechanism fail to restore stability once the threshold is crossed?* The answer is decoupling, which introduces a multi-step lag and effectively disables adaptation. Without decoupling, $v_t$ would quickly recover, leading to self-correcting oscillations phenomenon (EoS, Fig. 3) rather than spikes. Thus, curvature growth is inevitable, but spikes arise specifically from the failure of adaptive correction.
>
> **Q2: Clarification of the Role of $\beta_1$ and Eq. (7) in Remark 4.3.**
>
> We appreciate the reviewer’s observation and clarify that two distinct effects are involved:
>
> **(1) The stabilizing effect of $\beta_1$ on instantaneous curvature:**  The reviewer correctly points out that larger $\beta_1$ reduces the factor $(1-\beta_1)/(1+\beta_1)$ in Eq. (7), directly lowering $\lambda_{\text{max}}(\hat{H}_t)$. Empirically, we also observe that moderately increasing $\beta_1$ can slightly reduce spike severity.
>
> **(2) Distinguishing two decoupling mechanisms:** Increasing $\beta_1$ leads to decoupling between the momentum $m_t$ and the current gradient $g_t$, whereas the primary cause of spikes is the decay of $v_t$ and the decoupling between $v_t$ and $g_t^2$, governed by $\beta_2$. In Stage 3, since $g_t$ is already very small, its squared term $g_t^2$ decays much faster than $g_t$ itself. As a result, $v_t$ is far more prone to severe decoupling than $m_t$, as clearly illustrated in Fig. 3(b).
>
> **(3) Commitment to revision:** To eliminate this ambiguity, we will rewrite Remark 4.3 in the revision to explicitly clarify:
>
> *"Although increasing $\beta_1$ reduces the preconditioned Hessian curvature, this effect is fundamentally limited: as $v_t$ can decay toward zero, the preconditioned curvature can grow unbounded and eventually dominate. Therefore, controlling spikes critically depends on regulating the behavior of $v_t$, rather than $\beta_1$. We empirically observe that $v_t$–$g_t^2$ decoupling is exacerbated by larger $\beta_2$, as higher $\beta_2$ increases the inertia of the second-moment estimator. In the no-inertia limit ($\beta_1, \beta_2 \to 0$), Adam reduces to SignGD, which is memoryless and thus immune to this decoupling effect—resulting in only oscillations rather than severe spikes."*
>
> **Q3: Narrative Restructuring and the Scientific Value of Quantitative Prediction.**
>
> We thank the reviewer for precisely identifying the two central contributions of our work. We fully agree with this suggestion and will revise the presentation accordingly, while also clarifying our motivation for retaining a detailed discussion of quantitative prediction.
>
> **(1) Emphasizing the core mechanism and remedy:** We will strengthen the two main threads you highlighted: (i) the fundamental mechanism of spikes caused by the decoupling between $v_t$ and $g_t^2$, and (ii) the principled mitigation strategy via reducing $\beta_2$. In the revision, we will revise the introduction section to lead with these two points more prominently.
>
> **(2) Why precise prediction ($\lambda_{\text{grad}}$) remains essential:** Loss spikes are notoriously difficult to study due to their entanglement with many **confounding factors**, which has led to numerous plausible but ambiguous explanations in the literature. Our emphasis on mechanistic understanding, together with precise quantitative prediction, is motivated by the fact that the ability to **predict spikes and validate them through controlled intervention** provides strong evidence of **causality** for the proposed mechanism. For this reason, accurate prediction remains an essential component of our framework.
>
> We thank you again for helping us improve the quality of our manuscript. Should any further questions remain, we are fully committed to providing additional clarification and discussion.

---

> > ### Author Rebuttal · Reviewer_peNU · 2026-04-01
> >
> > I would like to thank the authors for the detailed reply.
> >
> > My concerns have been fully addressed. Please consider revising the corresponding parts in the paper. I will keep my score.

---

> > > ### Author Response · Authors · 2026-04-08
> > >
> > > We sincerely appreciate the reviewer's positive score and constructive comments, as well as their engagement with our work.

---

### Official Review · Reviewer_A96N · 2026-03-12

**Soundness:** 3
**Presentation:** 2
**Significance:** 3
**Originality:** 3
**Overall Recommendation:** 4
**Confidence:** 4

**Summary:**

During training with the Adam optimizer, the loss sometimes jumps up suddenly before coming back down. These are called loss spikes. People assumed they happened because of the shape of the loss landscape — that some regions are harder to optimize. This paper shows that explanation is wrong, or at least incomplete. Adam can produce spikes even on a simple quadratic function, where the landscape never changes.
The real cause is inside Adam itself. Adam tracks a running average of squared gradients called the second moment. When gradients get small during training, this second moment decays. It decays so slowly (controlled by the parameter β₂) that it cannot react when gradients start rising again. During this gap, the effective step size grows too large and the optimizer becomes unstable. The loss spikes because the second moment is blind to what is happening.
The authors describe this as a five-stage process and introduce a predictor — the curvature in the gradient direction — that reliably signals when a spike is about to happen. They test this across small networks, CNNs, and large Transformers up to 187 million parameters. They also show that reducing β₂ makes spikes less frequent, because the second moment then reacts faster to changes in the gradient.

**Compliance With Llm Reviewing Policy:**

Affirmed.

**Final Justification:**

I thank the authors for the thorough and well-organized rebuttal. The key concerns have been adequately addressed.
The mechanism-driven justification for the three-step sustained predictor is convincing, and the controlled experiments (Figs. 2, 6b) provide reasonable evidence separating the adaptive mechanism from landscape geometry effects. The local quadratic approximation argument is acceptable given its consistency with the EoS literature. I also appreciate the commitment to moving the limitations discussion into the main text and sharpening the novelty framing relative to prior work.
Minor concerns remain — particularly the lack of systematic analysis of mitigation trade-offs and the limited evaluation of the spike taxonomy — but these are not blocking issues.
I maintain my score.

**Key Questions For Authors:**

1. How is the sustained spike predictor designed, and how sensitive is it? The sustained predictor is defined as the minimum of three consecutive eigenvalue measurements. This choice appears arbitrary — why three steps, and why the minimum?

2. Can the decoupling mechanism be distinguished from the landscape geometry explanation in real training? The paper argues that Adam's internal dynamics, not landscape sharpness, are the primary cause of spikes. However, in practice these two factors are not independent — a sharp loss landscape will produce large gradients, which interact directly with the second moment dynamics. Can the authors provide an experiment where the two causes are cleanly separated, showing that decoupling occurs and causes a spike even when the loss landscape is provably flat?

3. Why does the quadratic analysis transfer to non-convex settings? The core theory is derived for a 1D quadratic function, yet the authors apply its conclusions to deep Transformers. The paper shows empirically that the predictor works, but does not explain theoretically why it should. Is there a principled reason the local quadratic approximation remains valid long enough for the decoupling mechanism to operate in non-convex, high-dimensional settings?

**Limitations:**

The authors do discuss limitations in Appendix C, acknowledging that the theory is derived from simplified models and that in more complex scenarios, both landscape geometry and preconditioner dynamics likely interact to jointly produce spikes. They also acknowledge that disentangling these contributions in large-scale models remains an open challenge, and that computational cost is a constraint for scaling the Hessian-based analysis to larger models. This is a reasonable and honest self-assessment. However, the limitations discussion is placed in the appendix rather than the main paper, which reduces its visibility. A brief limitations paragraph in the main text — particularly addressing the full-batch assumption and the theory-to-practice gap — would make the paper more transparent and complete.

**Strengths And Weaknesses:**

Strengths
The paper is technically rigorous and well-grounded. The central claim, which argues that Adam's second-moment estimator decouples from instantaneous squared gradients and causes sustained violations of the stability threshold, is supported by a coherent combination of linear stability analysis, a formal five-stage theorem for quadratic optimization, and a proof that decaying learning rate schedules cannot prevent instability when the second moment decay rate is large. The authors are appropriately honest about the gap between their simplified model and complex real-world settings, and the empirical validation spans a wide range of architectures including FNNs, CNNs, ViTs, ResNets, and Transformers up to 187M parameters, making the generalization claim credible rather than merely asserted.
The paper is clearly written and the narrative flows naturally from motivation through theory to experiments. The five-stage progression and the sharp contrast between decoupling causing spikes and coupling producing Edge-of-Stability oscillations are communicated effectively through well-chosen figures. The proposed gradient-directional curvature predictor is both theoretically motivated and computationally cheap, requiring only a single Hessian-vector product, making it practically deployable for training monitoring.
The contribution is also meaningfully original. While the preconditioned Hessian perspective exists in prior work, this paper is the first to precisely identify why the preconditioned eigenvalue sustains stability threshold violations, pointing specifically to the exponential moving average structure of the second moment estimator under small-update regimes. The practical remedy of reducing the second moment decay rate, validated on a 187M-parameter LLaMA model and already corroborated by trends in recent large-scale training recipes, further underscores the real-world relevance of the analysis.
Weaknesses
The core theory is derived for a one-dimensional quadratic, and the theoretical gap to high-dimensional non-convex settings remains substantial. The decoupling argument relies on parameter updates being small relative to the second moment estimator, but the paper does not characterize when this condition holds during practical training or provide any bounds on its duration. As a result, while the empirical generalization is encouraging, it is difficult to assess the full scope of the mechanism's applicability from a theoretical standpoint.
The sustained spike predictor, defined as the minimum of the gradient-directional curvature over three consecutive steps, is introduced without theoretical justification and evaluated on a limited number of spike events from a single Transformer run. Its claimed perfect correspondence should therefore be interpreted with caution. More broadly, the mitigation strategies explored such as reducing the second moment decay rate, increasing the stability constant, and clipping the second moment estimator are relatively straightforward, and their trade-offs with convergence speed and final model quality are not systematically analyzed.
The taxonomy of spike types covering neutral, benign, malignant, and catastrophic spikes is an interesting conceptual contribution, but the paper offers no principled or computationally feasible criterion for distinguishing these types prospectively during training. Without such a criterion, practitioners cannot reliably act on this taxonomy in real training scenarios. Additionally, the connection between second moment decay and instability is partially anticipated by prior concurrent work, and the paper could more explicitly articulate what is genuinely novel relative to those existing observations.

---

> ### Author Rebuttal · Authors · 2026-03-27
>
> We thank the reviewer for the thorough and positive assessment. We provide point-by-point clarifications below.
>
> **Q1 & W2: On the Design and Sensitivity of the "Sustained Spike Predictor".**
>
> The predictor is designed to filter transient EoS oscillations and capture only sustained instabilities that lead to spikes.
>
> **(1) Mechanism-driven design (not arbitrary):** The design is grounded in the underlying physics. As shown in Section 4.2, a spike requires sustained instability—a single-step violation of the threshold $2/\eta$ is insufficient, since the gradient must grow through repeated unstable updates. This is empirically validated in Figs. D9 and D10, where transient threshold crossings do not lead to spikes.
>
> **(2) Conservative “minimum” criterion:** In stochastic mini-batch training, noise can cause brief threshold violations that self-correct within 1–2 steps. Taking the minimum over consecutive steps ensures that all measurements exceed the threshold, effectively filtering out transient fluctuations and reducing false positives.
>
> **(3) Justification for window size (3 steps):** The three-step window is the simplest non-trivial choice to operationalize “sustained.” Since typical EoS oscillations revert within 1–2 steps, requiring three consecutive violations reliably indicates that the optimizer has exited the normal oscillatory regime.
>
> **Q2: Decoupling vs. Loss Landscape Geometry.**
>
> **(1) Joint effect of geometry and decoupling:** The preconditioned Hessian $\hat{H}$ integrates both the landscape Hessian $H$ and the adaptive preconditioner $v_t$. An increase in $H$ or a decrease in $v_t$ will both enlarge the eigenvalues of $\hat{H}$. The key question is which factor more realistically triggers spikes. Empirically, it is well observed that replacing Adam with GD on the *same* loss landscape dramatically reduces spike frequency—the geometry remains unchanged, yet spikes largely disappear.
>
> **(2) Controlled evidence under fixed curvature:** We further isolate this effect via controlled experiments where curvature is fixed. In Fig. 2 (quadratic), curvature is constant; in Fig. 6(b) (neural networks), the raw curvature (red) remains stable while the preconditioned curvature (blue) increases sharply, leading to spikes. This demonstrates that spikes arise from the adaptive mechanism rather than changes in landscape geometry.
>
> **Q3 & W1: Validity of Local Quadratic Approximation in Non-Convex Networks.**
>
> The key is to perform dynamic local quadratic approximations at each iterate, rather than relying on a fixed point.
>
> **(1) A generic local condition (beyond quadratic models):** Spike formation is inherently temporally local. For any twice continuously differentiable loss, second-order Taylor expansion governs the local dynamics. As shown in Proposition 4.1, $\lambda_{\max}(H_t) < 2/\eta$ ensures monotonic decrease in this regime, regardless of global non-convexity. Thus, any spike must correspond to a local violation of this condition.
>
> **(2) Gradient alignment with the dominant eigendirection:** As the instability develops, the gradient progressively aligns with the maximum eigendirection (Fig. 6d, orange curve). This alignment means the 1D quadratic analysis captures the dominant mode of the dynamics, even in high-dimensional non-convex settings.
>
> **(3) Consistency with prior observations:** This perspective is also consistent with the EoS literature, which empirically observes the effectiveness of instantaneous quadratic approximations in neural networks.
>
> **W3: Trade-offs between mitigation strategies and convergence speed.**
>
> Reducing $\beta_2$ accelerates $v_t$’s response to gradients and, in extreme cases (e.g., $\beta_2=0$, SignGD), may harm convergence. However, our results show that *moderate* reductions do **not** affect convergence speed (e.g., Fig. 7(d), Fig. 8(a)). This is consistent with practice: $\beta_2$ is often reduced from 0.999 to 0.95 or 0.9 for stable large-scale training (e.g., LLaMA), and the recent literature [1] further suggests setting $\beta_2=\beta_1$.
>
> [1] In Search of Adam’s Secret Sauce. NeurIPS 2025.
>
> **W4: The taxonomy of spikes and prospective criteria.**
>
> We agree this remains an open challenge and will state it explicitly as a limitation. In practice, even changing the random seed can lead to different spike categories. Developing a prospective criterion (e.g., based on the pre-spike trajectory of $\lambda_{\text{grad}}(\hat{H}_t)$) is a natural direction for future work.
>
> **W5: The connection to prior work could be more explicitly articulated.**
>
> While some prior work has observed that reducing second-moment decay improves stability, it remains largely **correlational**. Our contribution is to identify the precise **causal mechanism**, explaining when and why spikes occur, moving beyond empirical heuristics. We will sharpen this in the main text.
>
> **Limitation Movement.**
>
> We agree and will include a dedicated Limitations paragraph in Section 7 of the main text.

---

### Official Review · Reviewer_ULhG · 2026-03-12

**Soundness:** 2
**Presentation:** 3
**Significance:** 1
**Originality:** 2
**Overall Recommendation:** 5
**Confidence:** 3

**Summary:**

The paper investigates the internal dynamics of an Adam optimizer leading to a loss spike. The authors identified a set of stages that happen before spikes, showing a phase in which the gradient starts growing starting from very small values, to large ones, and the velocity continue to decrease as the growth in the gradient is not sufficient to counter the exponential decay ( $\beta_2$ ) of Adam.

**Compliance With Llm Reviewing Policy:**

Affirmed.

**Final Justification:**

Thank you for your very precise answers, which helped mitigate my concerns. I am raising my score as a consequence.

**Key Questions For Authors:**

- What evidence is there that this is not one of many possible internal dynamics that leads to spikes?

- How unique are these dynamics to Adam? Do they happen also in more memory friendly optimizers like adafactor for example?

**Limitations:**

yes

**Strengths And Weaknesses:**

**Strengths:**
- The paper is easy to follow and well organized.

**Weaknesses:**
- *Significance & Novelty:* The authors mention a few papers reaching the same conclusion. Tuning the optimizer’s hyper-params is standard practice
- *Experiments:* The experimentation section is not comprehensive. A lot of the experiments are done on simple synthetic data which might be biasing the results toward specific dynamics. I think more experiments on various architectures and various real world data is needed.
- Example, figure 3, the loss and gradient magnitude are shrinking exponentially for many steps and then growing exponentially for many steps - In such a setting there is going to be lag with v catching up, but is this setting realistic in the real world - especially at late stages of training?
- The numeric values are very small to the point where it might be introducing numerical issues.

---

> ### Author Rebuttal · Authors · 2026-03-25
>
> We sincerely thank the reviewer for the critical feedback. We address each concern below.
>
> **W1: Significance & Novelty**
>
> We agree hyperparameter tuning is standard practice, but our core contribution is the causal, mechanistic explanation of *when* and *why* spikes occur—something prior work does not provide.
>
> **(1) The limitation of current practices:** The industry currently relies on heuristic hyperparameter tuning or complex normalization techniques to mitigate spikes. These approaches address the "how" but leave the exact "when" and "why" unanswered. Several works study AEoS but do not distinguish severe spikes from ordinary EoS oscillations.
>
> **(2) Uncovering the root mathematical mechanism:** The scientific value of our work lies in pinpointing the exact intrinsic adaptivity failure: the **decoupling** of $v_t$ and $g_t^2$. We demonstrate that this decoupling causes the adaptive mechanism to lag and effectively fail when the **preconditioned instability** occurs.
>
> **(3) Broader impact for optimizer design:** Uncovering this mechanism goes beyond justifying known tricks (e.g., reducing $\beta_2$); it motivates a rethinking of how to design more robust adaptive optimizers, offering valuable insight to the ML community.
>
> **W2: Comprehensiveness of Experiments and Synthetic Data Bias**
>
> We clarify that our validation spans multiple scales and real-world data:
>
> **(1) Large-scale real data:** Fig.8 trains a 187M LLaMA-structured Transformer on 100B tokens from SlimPajama under standard stochastic mini-batch training, observing the same gradient shrinking and preconditioned instability.
>
> **(2) Diverse architectures on real data:** We validate across FNN (Fig.6), CNN(Fig.D7,D8), ViT (Fig.D17a,b), and ResNet-18 (Fig.D17c,d) on CIFAR-10. We also add new FNN experiments on MNIST data with results available at https://ibb.co/KxnfBrpW.
>
> **(3) On the 10M Transformer synthetic data (Fig.7):** The data consists of compositional reasoning sequences, not selectively chosen, but a standard reasoning task for small-scale Transformers used in recent top-tier work([1],[2]). We will clarify this setting in the main text.
>
> [1] Initialization is Critical to Whether Transformers... in arXiv:2405.05409, NeurIPS 2024.
>
> [2] Complexity Control Facilitates Reasoning... in arXiv:2501.08537, TPAMI 2025.
>
> **W3: Realism of Fig. 3 dynamics**
>
> The decoupling mechanism in Fig. 3 is not a synthetic artifact—it is directly observed in real networks. In Figs.7(a), D7(d), and D8(d), $v_t$ decays at rate $\approx \beta\_2$ in specific parameter blocks of Transformers and CNNs, confirmed by fitting. This occurs when those layers learn effective representations and their local gradients shrink. Our 187M LLaMA experiments (Fig. 8c) show sudden gradient diminishment in certain layers immediately before spikes. The key insight is **not exponential decay per se**, but the resulting decoupling ($v_t = \beta_2 v_{t-1}$), which we empirically verify across all tested architectures.
>
> **W4: Numerical issues**
>
> Spikes are not floating-point artifacts. Our small-scale experiments use IEEE-754 double precision (values representable to ~$10\^{-308}$). Crucially, Fig. D18 shows that artificially increasing $\epsilon$ to prevent $\lambda\_{\text{grad}}(\hat{H}\_t)$ from exceeding $2/\eta$ allows Adam to converge stably to losses as low as $10\^{-300}$ with no spikes—proving the mechanism is mathematical, not numerical.
>
> **Q1: Is this one of many possible mechanisms?**
>
> We do not claim exclusivity, but provide three layers of evidence that this is a dominant, causally verifiable mechanism:
>
> **(1) Quantitative prediction:** $\lambda_{\text{grad}}>2/\eta$ achieves near-perfect correspondence with spike onset across all tested architectures (FNN, CNN, ViT, ResNet, 10M and 187M Transformers).
>
> **(2) Targeted intervention:** Increasing $\epsilon$ to enforce $\lambda_{\text{grad}}<2/\eta$ averts the spike (Fig. 6(f)). Reducing $\beta_2$ strengthens $v_t$-$g_t^2$ coupling and consistently reduces spike frequency (Figs. 7d, 8a). AdaGrad (Fig. D14), which uses cumulative rather than exponential averaging so $v_t$ can only grow, produces no spikes at all—directly confirming the role of $v_t$ decay.
>
> **(3) Isolation from external factors:** Spikes occur on perfectly clean data and simple quadratic models, ruling out data anomalies as the root cause.
>
> **Q2:. Do these dynamics occur in Adafactor?**
>
> Yes. This susceptibility stems from the EMA of squared gradients in the preconditioner. RMSprop exhibits spikes due to preconditioner decay (Fig.D15), while AdaGrad does not (Fig.D14). We add new **Adafactor** experiments confirming similar decay and spikes. The preconditioner for Adafactor is $\hat{H}\_t = \frac{\max(\varepsilon_2,RMS(\theta_{t-1}))\rho_t}{(\sqrt{v_t}+\varepsilon_1)\max(1,RMS(u_t)/d)\eta}$, where $v_t$ is reconstructed from the factored second moment. Results for Adafactor under varying $\beta_2$ are provided at https://ibb.co/fVs56xtM.

---

> > ### Author Rebuttal · Reviewer_ULhG · 2026-04-01
> >
> > Thank you for your very precise answers, which helped mitigate my concerns. I am raising my score as a consequence.

---

> > > ### Author Response · Authors · 2026-04-08
> > >
> > > We sincerely thank the reviewer for re-evaluating our work and raising their score. All corresponding modifications have been carefully incorporated into the revised manuscript.

---

### Official Review · Reviewer_vha7 · 2026-03-13

**Soundness:** 4
**Presentation:** 4
**Significance:** 3
**Originality:** 4
**Overall Recommendation:** 5
**Confidence:** 3

**Summary:**

This paper shows that the toss spikes during neural network training with the Adam occur when Adam’s second moment decouples from the true squared gradients. The paper identifies an interesting 5 stage mechanisms for the spikes, which is interesting

**Compliance With Llm Reviewing Policy:**

Affirmed.

**Final Justification:**

I think this is a good work with a good message and nicely done experiments etc

**Key Questions For Authors:**

NA

**Limitations:**

I think the limitations can be better discussed

**Strengths And Weaknesses:**

Strength: I think this is a very good work!

The message is clear and novel, and with real implications. The problem of the spikes of adaptive optimizers is relevant and interesting for both theory and practice.

 The clear identification of the 5-stage process is interesting and insightful. This could have both empirical and scientific values and could inspire more work to do this kind of work. In some sense, this constitutes a case study of training, which has been more often rejected by the community in the past, but I think case studies should be more common in the future in order to advance the science AI

I do not have much to complain. Do not take the shortness of this review to mean that I did not read it in detail, but that I think the paper is quite well written in every aspect

Weakness: nothing major, I think the authors could explain the limitations a lot better (such as the simplicity of the model etc..)

---

> ### Author Rebuttal · Authors · 2026-03-27
>
> We sincerely thank the reviewer for the thorough reading and highly positive evaluation. We are glad the five-stage mechanistic characterization resonated, and we agree that detailed case studies of this kind are valuable for advancing the science of optimization.
>
> **Weakness: Need for a better discussion of limitations (e.g., the simplicity of the model).**
>
> We fully agree. We will move the core limitations discussion from Appendix C into the main text as a dedicated paragraph in Section 7 (Conclusion and Discussion). The revised paragraph will address three dimensions of limitation that we believe are most important for readers to be aware of:
>
> *"Limitations. (1) Our rigorous theoretical analysis is established for the 1D quadratic case, and the transfer to high-dimensional non-convex settings rests on empirical validation rather than formal proof. In practical large-scale scenarios, loss landscape geometry and adaptive preconditioner dynamics likely interact jointly to produce spikes, and cleanly disentangling these contributions remains an open challenge. (2) Additionally, most of our eigenvalue-tracking experiments use full-batch gradients for tractability; the stochastic mini-batch setting introduces noise that complicates the precise timing of spike prediction, as we partially address with the sustained predictor in Section 6.2. (3) Finally, scaling the Hessian-vector-product-based eigenvalue analysis to models significantly beyond 200M parameters remains computationally demanding, and developing more efficient approximation methods is an important direction for future work."*
>
> We believe this explicit addition provides a much more balanced view of our contributions and clearly outlines the boundaries of our analysis. We thank you again for helping us improve the completeness of our manuscript.

---

> > ### Author Rebuttal · Reviewer_vha7 · 2026-03-31
> >
> > Thanks for the rebuttal. I will keep my original score.
> >
> > I also want to say that I disagree with Reviewer ULhG's criticisms, which I find entirely generic (the theory part) and can be used against any theory paper (the experiment part)

---

> > > ### Author Response · Authors · 2026-04-08
> > >
> > > We thank the reviewer for their positive score and constructive feedback, as well as their continued engagement with our manuscript.

---

### Decision · Program_Chairs · 2026-04-30

**Decision:**

Accept (regular)

**Comment:**

### **Summary**
This paper investigates the phenomenon of "loss spikes" during neural network training with the Adam optimizer. The authors propose a mechanistic explanation grounded in the "decoupling" between Adam's second moment estimator ($v_t$) and the instantaneous squared gradients ($g_t^2$). Through a 5-stage characterization based on linear stability analysis of quadratic functions, the authors argue that this decoupling leads to a sustained violation of the stability threshold by the maximum eigenvalue of the preconditioned Hessian, thereby triggering these loss spikes. During the review process, all four reviewers appreciated the empirical narrative, the detailed case studies, and unanimously recommended acceptance.

Despite the unanimous positive scores from the reviewers, a rigorous examination of the paper's theoretical foundations (particularly Section 4 and Appendix D) reveals mathematical inconsistencies. The concerns are based on the following three mathematical points, and are partially addressed in the rebuttal. We hope the authors will update their paper for the final version accordingly.

**1. Invalidity of the Global Quadratic Approximation:**
The core derivations (Equations 4 and 5) rely on a local second-order Taylor expansion around a fixed point $\theta_0$. While linear stability analysis is standard for studying local convergence near a stationary point, applying a fixed Hessian $H(\theta_0)$ to approximate the entire trajectory $\{\theta_1, \theta_2, \dots, \theta_t\}$ of a loss spike is mathematically insufficient. Loss spikes represent macroscopic, large-displacement events in the parameter space (such as escaping local basins). Consequently, the parameters do not reside in a sufficiently small neighborhood to justify discarding higher-order terms globally.

**2. Incompatibility with Non-Convex Landscapes (Proposition 4.2):**
The stability proof for Proposition 4.2 implicitly requires the Hessian to be positive definite. The asymptotic stability condition $1 - \alpha_i + \beta_1 > 0$ mathematically reduces to $\eta(1-\beta_1)\lambda_i > 0$. Given that $\eta > 0$ and $\beta_1 \in [0, 1)$, this necessitates $\lambda_i > 0$. While the authors restrict their claim to "positive-curvature eigendirections," ignoring negative curvature directions limits the analysis for deep learning landscapes. In highly non-convex neural networks, negative eigenvalues (saddle points) are primary drivers of instability.

**3. Mathematical Discrepancy in Deriving Equation (7):**
Even if the optimization is restricted to a simple quadratic function, the conclusions of Proposition 4.2 cannot be mathematically generalized to Equation (7). Incorporating Adam's actual update rule transforms the recurrence relation into:

$$
\delta\theta _{t+1} \approx \left[ \left( 1 + \beta _1 \frac{\sqrt{\hat{v} _{t-1}}+\epsilon}{\sqrt{\hat{v} _{t}}+\epsilon} \right) I - \eta (1-\beta _1) \mathrm{Diag}\left(\frac{1}{\sqrt{\hat{v} _{t}}+\epsilon}\right) H \right] \delta\theta _{t} - \beta _1 \frac{\delta\theta _{t-1}}{\sqrt{\hat{v} _{t-1}}+\epsilon} - \eta(1-\beta _1)\frac{\nabla L(\theta _0)}{\sqrt{\hat{v} _{t}}+\epsilon}
$$

This expansion demonstrates that the system becomes time-varying. This does not align with the fundamental requirement in the proof of Proposition 4.2, which assumes a Linear Time-Invariant (LTI) system where the coefficients $\alpha_i$, $\beta_1$, and $c_i$ are constant.

Furthermore, the preconditioned Hessian $\hat{H}$ in Adam is intrinsically asymmetric. Therefore, it cannot be orthogonally decomposed into the form $\hat{H} = Q \Sigma Q^T$ (where $Q$ is an orthogonal matrix and $\Sigma$ is a diagonal matrix). If such a decomposition existed, it would mathematically necessitate that $\hat{H}^T = (Q \Sigma Q^T)^T = Q \Sigma^T Q^T = Q \Sigma Q^T = \hat{H}$, improperly implying that $\hat{H}$ is symmetric.

### **Conclusion**
While the AC fully acknowledges the interesting empirical observations regarding Adam's optimization dynamics, and deeply appreciates the positive scores from all four reviewers based on the paper's experimental narrative, the AC has significant concerns regarding the theoretical framework. The proposed mechanism, which relies heavily on stationary, time-invariant, and positive-definite quadratic assumptions, unfortunately cannot be seamlessly generalized to the time-varying, non-convex dynamics of Adam using the current mathematical derivations. The AC believes the empirical findings are highly valuable to the community, but the theoretical proofs require a substantial and rigorous revision. We encourage the authors to refine the mathematical formulations for the final version.